# REINFORCE Converges to Optimal Policies with Any Learning Rate

**Samuel Robertson**[1][*] **Thang D. Chu**[1][*] **Bo Dai**[2][3] **Dale Schuurmans**[1][2]
**Csaba Szepesvári**[1][2] **Jincheng Mei**[2]
[1]University of Alberta   [2]Google DeepMind   [3]Georgia Institute of Technology

{smrobert,thang}@ualberta.ca
{bodai,schuurmans,szepi,jcmei}@google.com

## Abstract

We prove that the classic REINFORCE stochastic policy gradient (SPG) method converges to globally optimal policies in finite-horizon Markov Decision Processes (MDPs) with *any* constant learning rate. To avoid the need for small or decaying learning rates, we introduce two key innovations in the stochastic bandit setting, which we then extend to MDPs. **First**, we identify a new exploration property: the online SPG method samples every action infinitely often, improving on previous results that only guaranteed at least two actions would be sampled infinitely often. This means SPG inherently achieves asymptotic exploration without modification. **Second**, we eliminate the assumption of unique mean reward values, a condition that previous convergence analyses in the bandit setting relied on, but that does not translate to MDPs. Our results deepen the theoretical understanding of SPG in both bandit problems and MDPs, with a focus on how it handles the exploration-exploitation trade-off when standard analysis techniques for optimization and stochastic approximation methods cannot be applied, as is the case with large constant learning rates.

## 1 Introduction

Policy gradient (PG) methods constitute one of the most popular classes of algorithms for reinforcement learning (RL). In the PG paradigm, a learner acts according to a parameterized policy; the expected return is directly optimized by computing its gradient with respect to the policy parameters and performing stochastic gradient ascent. PG methods have played a key role in the advancements of deep RL (Lillicrap et al., 2019; Schulman et al., 2017a,b): combined with deep neural networks, PG algorithms have shown strong empirical performance across many domains, including robotics Akkaya et al. (2019), games Vinyals et al. (2019), and large language model training (Rafailov et al., 2024; Ouyang et al., 2022).

Despite PG methods' conceptual simplicity and rich set of practical applications, known theoretical guarantees on their performance come with restrictive assumptions. In particular, convergence proofs either require oracle access to the exact gradient (Liu et al., 2024; Agarwal et al., 2020), which is akin to demanding that the reward function and dynamics of the environment are known to the learner, or they impose harsh constraints on the learning rate used for stochastic gradient ascent (Mei et al., 2024b; Klein et al., 2024). Both of these assumptions are violated in typical applications. In the stochastic setting, where the rewards and transition probabilities are unknown and must be estimated from interaction with the environment, convergence of the classic REINFORCE algorithm (Williams,

---

[*]Equal contributions

39th Conference on Neural Information Processing Systems (NeurIPS 2025).

1992) has only been shown under the assumption that the learning rate is either sufficiently small (Klein et al., 2024) or decaying (Zhang et al., 2020a). In this work we study REINFORCE with the standard softmax parameterization, and narrow the gap between theory and practice by providing the first proof in the stochastic setting that REINFORCE will globally converge to an optimal policy in tabular finite-horizon Markov Decision Processes (MDPs) with *any* constant learning rate. Along the way we show new results about the stochastic gradient bandit algorithm (Sutton and Barto, 2018; Mei et al., 2024a), which is the special case induced by applying REINFORCE to a bandit problem. Specifically, we show that the stochastic gradient bandit algorithm automatically achieves sufficient exploration for global convergence with an arbitrary constant learning rate; in doing so we remove a key assumption in prior work, and thus resolve an open problem posed by Mei et al. (2024a). Our results in the bandit setting extend to a more general "nonstationary bandit problem", where the reward function is allowed to drift mildly across timesteps. This extension is then embedded into the RL setting where, with some additional arguments, we derive the convergence of REINFORCE. In summary, the main contributions of this work are threefold:

i) We show that the stochastic gradient bandit algorithm will select every arm infinitely often (i.o.) in any bandit problem and with any learning rate. We find it surprising that this strong property emerges from such a simple algorithm, without any explicit hacks to encourage exploration. We obtain a counterpart result in the RL setting, but the bandit case is independently interesting, and also critical for our second contribution:

ii) In the bandit setting we remove the central assumption of Mei et al. (2024a), that no two arms have the same expected reward, and prove that the stochastic gradient bandit algorithm still converges to an optimal policy. For bandits this assumption is impossible to verify without access to the true reward function (at which point the bandit problem is already solved), but more importantly it renders the extension to RL virtually impossible.

iii) In RL we provide the first proof that REINFORCE converges with large learning rates in the stochastic setting. This requires the first two contributions: the exploration result is applied directly to RL, and the bandit result is extended to a nonstationary bandit problem that can be embedded into an MDP.

Positioning our work, to our best knowledge, we note that existing convergence results for stochastic policy gradient (SPG) methods typically suffer from one of the following drawbacks: **(i)** they rely on decaying learning rate schedules for convergence guarantees (Zhang et al., 2020a; Ding et al., 2022, 2024; Mei et al., 2023), a requirement not aligned with the constant rates commonly used in practice; **(ii)** they study constant learning rates (Mei et al., 2024b; Klein et al., 2024), but provide guarantees only for rates considered impractically small; or **(iii)** they are restricted to the simplest bandit settings by assumptions on the reward structure (Mei et al., 2024a), preventing extensions to RL. Filling this gap, our work provides rigorous convergence guarantees for SPG using practical learning rates in MDPs, without requiring uniqueness of the optimal policy.

## 2 Related Work

In the exact gradient setting, where the rewards and transition probabilities of the MDP are known, Agarwal et al. (2020) exploited the notion of gradient dominance (Polyak, 1963) to show that PG algorithms with the tabular softmax parameterization can attain asymptotic convergence towards a globally optimal policy. Mei et al. (2020b) leveraged Łojasiewicz-like inequalities to show a $O(1/t)$ convergence rate, where $t$ is the number of update iterations. Moreover, Zhang et al. (2020b) exploited the "hidden convexity" property of MDPs to establish a global optimality result. In summary, these works exploited a variety of conditions for global convergence results of vanilla PG algorithms in the exact gradient setting.

There have also been several techniques developed to improve the convergence rate of PG algorithms, including entropy regularization (Mei et al., 2020b), normalization (Mei et al., 2022), natural gradients (Cen et al., 2021; Lan, 2022; Khodadadian et al., 2021; Xiao, 2022), and accelerated learning rates (Chen et al., 2024). Some of these techniques have known convergence rates in the exact setting (Liu et al., 2024), but the results are unlikely to carry over to the stochastic setting. Indeed, all of these techniques except accelerated learning rates have been shown not to globally converge in the stochastic setting due to "over-committal" behavior in the update rule (Mei et al., 2021), and

accelerated learning rates have no convergence guarantee; the noise in the gradient causes these methods to commit overzealously to suboptimal policies.

PG algorithms for the stochastic setting, or SPG algorithms, have also received theoretical attention. The common approach is to pair decaying or sufficiently small learning rates with SPG in order to control the noise in the gradient estimates. In particular, Zhang et al. (2020a) showed that REINFORCE Williams (1992) with a $O(1/\sqrt{t})$ learning rate converges to an $\epsilon$-optimal policy at a $\tilde{O}(\epsilon^{-2})$ rate. Ding et al. (2024) demonstrated that SPG algorithms with softmax parameterizations, entropy regularization, and a $O(1/t)$ learning rate enjoy a $\tilde{O}(\epsilon^{-2})$ sample complexity. In addition to controlling the learning rates, variance reduction techniques have been developed to reduce the effect of noisy estimates (Masiha et al., 2023; Fatkhullin et al., 2023). The main drawbacks of these approaches are that they introduce decaying learning rates (Zhang et al., 2020a; Ding et al., 2024; Mei et al., 2023), incur expensive subroutines (Masiha et al., 2023), or assume prior knowledge of the true state-value function (Mei et al., 2023).

Recently, Mei et al. (2024b) analyzed softmax SPG in the bandit setting and proved that it converges to the optimal policy with a sufficiently small constant learning rate. However, their learning rate depends on prior knowledge of the reward gap between the optimal and next-best arm, which is typically unknown. Klein et al. (2024) proved global convergence for SPG algorithms with constant learning rates in finite-horizon MDPs, but their technique strongly requires the learning rate to be sufficiently small in order to induce a smoothness property of the objective function. The most relevant work to ours is that of Mei et al. (2024a), who recently showed that SPG algorithms with arbitrarily large learning rates converge to an optimal policy in the bandit setting. The major caveat is the assumption that there are no ties in the mean reward among arms–even among suboptimal ones. This strong assumption raises concerns about convergence in a bandit problem with any equally good arms (e.g. any bandit problem with its worst arm duplicated), and also hinders the extension to RL, since the analogous assumption would be that no state has ties in $Q$-values under *any policy*.

## 3 Challenges of Non-Unique Solutions

### 3.1 Non-Uniqueness of Policies in RL

In standard optimization, it is well known that gradient-based algorithms can exhibit non-convergence of their parameters (or iterates) when multiple optimal solutions exist (Absil et al., 2005). To avoid this challenge, existing results for the SPG algorithm in the $K$-armed bandit setting (Mei et al., 2024b,a) rely on the following assumption, which implies the uniqueness of the globally optimal policy.

**Assumption 3.1** (True mean reward has no ties). For all $a, b \in [K]$, if $a \neq b$, then $r(a) \neq r(b)$.

In Assumption 3.1 $[K] := \{1, \ldots, K\}$ denotes the set of $K$ arms and $r(a)$ is the true mean reward for arm $a \in [K]$. Assumption 3.1 implies that there is a unique optimal arm, which we denote $a^* := \arg\max_{a \in [K]} r(a)$. This results in a unique one-hot globally optimal policy $\pi^*$ with $\pi^*(a^*) = 1$ and $\pi^*(a) = 0$ for all $a \neq a^*$.

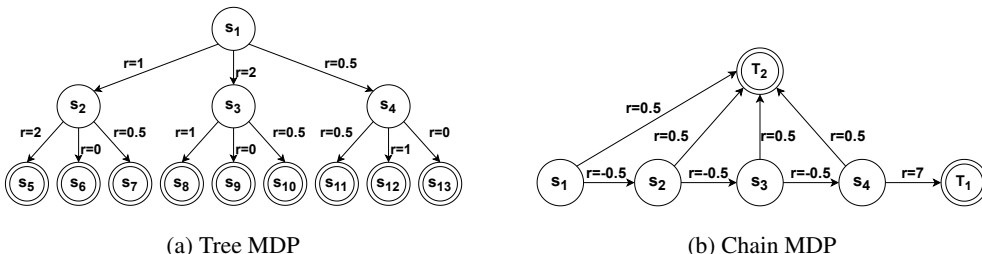

(a) Tree MDP                    (b) Chain MDP

Figure 1: Classical examples of finite-horizon MDPs. Transitions are labelled with (deterministic) rewards, and double circles indicate terminal states.

However, extending to the RL setting presents the challenge of multiple optimal policies, a scenario which is not prevented by the straightforward extension of Assumption 3.1 to each state, since

Assumption 3.1 only constrains immediate rewards. In contrast to bandits, RL involves sequential decisions, and different action sequences (trajectories) can yield the same (maximal) cumulative reward. This situation is common in tasks like navigation with alternative optimal paths; consider the tree-structured MDP shown in Fig. 1a (state space $\mathcal{S} = \{s_1, \ldots, s_{13}\}$, action space $\mathcal{A} = \{a_1, a_2, a_3\}$). Here, both $s_1 \to s_2 \to s_5$ and $s_1 \to s_3 \to s_8$ are optimal paths with total reward 3. Because previous bandit convergence analyses (Mei et al., 2024b,a) critically rely on the assumption of a unique optimal policy, they cannot be directly applied to RL problems exhibiting such non-uniqueness.

## 3.2 SPG Policy Non-Convergence in the Presence of Ties

On the other hand, Mei et al. (2024b, Remark 5.3) conjectured that the SPG algorithm could still achieve convergence even without Assumption 3.1. Their conjecture was based on the idea that SPG has a "self-reinforcing" property, causing the probability of **only one** arm to eventually become dominant and converge to 1, thus resulting in a stationary one-hot optimal policy as $t \to \infty$. That is, $\pi_t(a^*) \to 1$ for **only one** optimal arm as $t \to \infty$, almost surely, **even when multiple optimal arms exist**. If this behavioral property holds, the latter part of the convergence proof can utilize the contradiction-based arguments presented in (Mei et al., 2024b, Theorem 5.1, Claim 2).

Our first major finding, supported by both empirical evidence and theoretical analysis, is that the aforementioned conjecture is incorrect: SPG-like algorithms do not necessarily converge to a **single** policy in the presence of multiple solutions. To demonstrate this, we designed a bandit experiment with two optimal arms (mean 0.2) and one suboptimal arm (mean $-0.1$). Fig. 2a shows typical runs of the stochastic gradient bandit algorithm (Algorithm 1) with initial parameters $\theta_0 := (0, 0, 0)$ and learning rates $\eta \in \{1, 10\}$ for $10^5$ iterations, revealing that, while the total probability of optimal arms converges to 1 ($\sum_{a \in \mathcal{A}^*} \pi_t(a) \to 1$), the probabilities of individual optimal arms (1 and 2) display non-stationary behavior. We observed analogous behavior in a similar experiment on a tree-structured MDP using REINFORCE (Williams, 1992) with $\eta \in \{0.1, 0.5\}$, as shown in Fig. 2b, where optimal action probabilities from state $s_1$ fail to converge to a unique action. Moreover, we prove the following theorem, which supports the phenomena observed in simulation.

**Proposition 3.2** (Non-Stationary Convergence). *Consider a $K$-armed bandit with all arms being equally good, i.e. $\mathcal{A}^* = [K]$, and at least one arm has a nonzero probability of generating a nonzero reward. Running Algorithm 1 with any $\eta > 0$ leads to*

$$\liminf_t \pi_t(a) < 1 \text{ a.s,} \tag{1}$$

*for all $a \in [K]$. Therefore $(\pi_t)_{t \geq 0}$ does not converge to any one-hot policy.*

*Proof sketch* 1. We first argue that the sequence of parameters $(\theta_t)_{t \geq 0}$ does not converge to a fixed finite vector in $\mathbb{R}^K$. Since all arms are equally good, $(\theta_t)_{t \geq 0}$ is a martingale, and we can use the fact that it doesn't converge to a finite value and standard martingale results to show that it will undergo unbounded oscillations, implying the result.

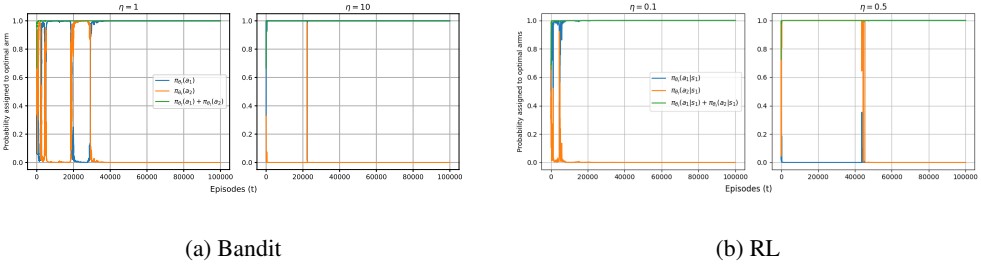

(a) Bandit                                      (b) RL

Figure 2: Fig. 2a shows that the total probability of optimal arms converges to 1, but the probabilities of individual optimal arms are non-stationary (i.e. $\pi_t(a_1)$ oscillates). Fig. 2b shows similar non-stationary behavior for optimal actions in an RL setting with multiple optimal trajectories.

## 3.3 Limitations of Standard Analysis

Global convergence for SPG algorithms is typically established through a two-stage proof: **(i)** establish convergence to a stationary point, and **(ii)** demonstrate (often by contradiction) that the

attained stationary point is globally optimal. This methodology originates from seminal work on PG in the exact gradient setting (Agarwal et al., 2020, Theorem 5.1).

Our findings demonstrate two key points: **(i)** ties in trajectory or policy value can exist in RL settings regardless of assumptions on immediate rewards; and **(ii)** the typical two-stage proof strategy for arguing global convergence of SPG cannot be directly extended to RL. However, as shown in the next section and suggested by the above simulations, convergence results **can** be obtained even with ties, but this requires new analysis. This is primarily because our approach needs to carefully reason about the per-timestep expected progress in distinguishing optimal from suboptimal actions despite the presence of these ties. Specifically, we prove that the learned policy eventually converges to assign all probability mass to the optimal set (a form of "generalized one-hot policy"), i.e. $\sum_{a \in \mathcal{A}^*} \pi_t(a) \to 1$ as $t \to \infty$.

# 4 An Illustrative Bandit Setting

This section presents new insights into the exploration properties of the SPG algorithm. We first analyze the simplest bandit setting for illustration and then extend the results to RL.

## 4.1 Stochastic Gradient Bandit

We consider a stochastic multi-armed bandit problem with $K \geq 2$ arms and rewards bounded in $[-R, R]$ (where $R > 0$). At each iteration $t \geq 1$, the learner selects an arm $a_t \in [K] := \{1, \ldots, K\}$ and observes a reward $r_t$ sampled from a fixed distribution $P_{a_t} \in \mathcal{M}_1([-R, R])$.[1] The true mean reward for arm $a \in [K]$ is $r(a) := \int_{-R}^{R} x P_a(dx)$. The set of optimal arms is denoted $\mathcal{A}^* := \arg\max_{a \in [K]} r(a)$.

The learner aims to find a policy $\pi \in \mathcal{M}_1([K])$ that maximizes expected reward. We use the softmax parameterization over $\mathbb{R}^K$: for $\theta \in \mathbb{R}^K$ and $a \in [K]$,

$$\pi_\theta(a) := \frac{\exp(\theta(a))}{\sum_{b \in [K]} \exp(\theta(b))} \,. \tag{2}$$

The optimization problem the learner is solving thus has the objective

$$\max_{\theta \in \mathbb{R}^K} \pi_\theta^\top r \,. \tag{3}$$

We study the stochastic gradient bandit algorithm (Algorithm 1), which performs stochastic gradient ascent on Eq. (3) (Sutton and Barto, 2018; Mei et al., 2024b). Given $\theta_0$ and learning rate $\eta > 0$ the algorithm iteratively updates parameters using the information it receives from single interactions. The stream of parameters generated will be referred to as $(\theta_t)_{t \geq 0}$, and we will use $\pi_t := \pi_t$ for the policy used to select $a_{t+1}$.

---

**Algorithm 1** Stochastic gradient bandit algorithm

---
1: **input** $\theta_0 \in \mathbb{R}^K, \eta > 0$
2: **for** $t \geq 0$ **do**
3:     Select $a_{t+1} \sim \pi_t$, and observe $r_{t+1} \sim P_{a_{t+1}}$.
4:     $\theta_{t+1}(a_{t+1}) \leftarrow \theta_t(a_{t+1}) + \eta(1 - \pi_t(a_{t+1}))r_{t+1}$.
5:     **for** $a \in [K], \ a \neq a_{t+1}$ **do**
6:         $\theta_{t+1}(a) \leftarrow \theta_t(a) - \eta\pi_t(a)r_{t+1}$.
7:     **end for**
8: **end for**

---

## 4.2 A Novel Exploration Lemma

We detail the reason why existing results (Mei et al., 2024a) do not generalize, even to bandit settings with reward ties. Mei et al. (2024a, Lemma 2) establishes an exploration property for SPG, showing

---

[1]Where $\mathcal{M}_1(S)$ denotes the collection of probability distributions over the set $S$.

that at least two **distinct** arms are sampled i.o. Their subsequent convergence proof (Mei et al., 2024a, Theorem 2) relies on the argument that at least one of these i.o. sampled actions must be optimal. However, in the presence of reward ties, it is possible for two actions to share the same reward value (the sub-optimal action's interval from (Mei et al., 2024a, Eq. (15)) no longer exists). Consequently, the arguments that construct a contradiction to show "at least one of these i.o. sampled actions must be optimal" are no longer valid.

Given the failure of existing approaches with reward ties, new analytical results are required for convergence proofs, even in the bandit setting. Our second key finding is a generalized exploration property for SPG: we establish that despite reward ties, **every arm** is sampled i.o. To formalize this, we define $N_t(a)$ as the number of times action $a \in [K]$ has been sampled up to iteration $t \geq 1$, i.e.

$$N_t(a) := \sum_{s=1}^{t} \mathbb{I}\{a_s = a\}. \tag{4}$$

The asymptotic count is $N_\infty(a) := \lim_{t \to \infty} N_t(a)$, which is either finite or infinite. If $N_\infty(a) < \infty$, action $a$ is only sampled finitely many times; if $N_\infty(a) = \infty$, action $a$ is sampled i.o.

**Lemma 4.1** (Bandit Exploration). *Using Algorithm 1 with any constant learning rate $\eta \in \Theta(1)$, every arm is almost surely played infinitely often. That is, $\forall a \in [K] : N_\infty(a) = \infty$ almost surely.*

*Proof sketch* 2. For any arm $a' \in [K]$ such that $N_\infty(a') < \infty$, the Extended Borel-Cantelli (Breiman, 1992) Lemma implies $\sum_{t=0}^{\infty} \pi_t(a') < \infty$. Since such an arm is sampled only finitely many times, its parameter $\theta_t(a')$ remains bounded, $\sup_t |\theta_t(a')| < \infty$, and its probability converges to zero: $\lim_{t \to \infty} \pi_t(a') = 0$. Without loss of generality, let $a \in [K]$ be an arm with $N_\infty(a) < \infty$. The condition $\lim_{t \to \infty} \pi_t(a) = 0$ requires that some parameter grows unboundedly, i.e. $\lim_{t \to \infty} \max_{a' \in [K]} \theta_t(a') = \infty$. To preserve the total probability mass, this necessitates that some parameter must diverge to negative infinity: $\lim_{t \to \infty} \min_{a' \in [K]} \theta_t(a') = -\infty$. Thus, there exists at least one arm $b \in [K]$ such that $\liminf_{t \to \infty} \theta_t(b) = -\infty$. Furthermore, since the sum of probabilities for finitely sampled arms is finite, any arm $b$ with $\liminf_{t \to \infty} \theta_t(b) = -\infty$ must be sampled infinitely often ($N_\infty(b) = \infty$).

We use these properties of arms $a$ (finitely sampled, bounded parameter) and $b$ (infinitely sampled, parameter unbounded below) to construct a proof by contradiction. The fact that arm $b$ is sampled infinitely often despite its parameter repeatedly dropping to arbitrarily low values implies that $\theta_t(b)$ must periodically increase to become larger than $\theta_t(a)$ (and other bounded parameters) infinitely often. Consider the event $C_t := \{\theta_t(b) < \theta_t(a), a_t = b\}$. We first show that if $\theta_t(b) \leq \theta_t(a)$ and the parameter update causes $\theta_{t+1}(b) > \theta_{t+1}(a)$, this implies $a_t = b$. We then prove that the event $C_t$ occurs only a finite number of times. However, for arm $b$ to be sampled infinitely often ($N_\infty(b) = \infty$) while $\liminf \theta_t(b) = -\infty$ and $\theta_t(a)$ is bounded, it must be sampled infinitely often during periods when $\theta_t(b) < \theta_t(a)$. This contradicts the finding that $C_t$ occurs only finitely often, proving our initial assumption ($N_\infty(a) < \infty$ for some arm $a$) is false.

## 4.3 Convergence Without the Assumption of Unique Rewards

Our new result about the exploration of SPG in the bandit setting, Lemma 4.1, allows us to remove an assumption necessary for the results of prior work (Mei et al., 2024a,b), namely that there are no ties in the true mean rewards of the arms (Assumption 3.1). However, this requires new analysis beyond the exploration proof. In this section we sketch out the steps used to show our central result in the bandit setting: that Algorithm 1 converges almost surely regardless of the learning rate.

**Theorem 4.2** (Convergence in Bandits). *In the bandit setting of Section 4.1 without Assumption 3.1, Algorithm 1 with any $\eta \in \Theta(1)$ almost surely converges to playing optimal arms,*

$$\lim_{t \to \infty} \sum_{a \in \mathcal{A}^*} \pi_t(a) = 1 \text{ a.s.} \tag{5}$$

The proof of this theorem breaks into two propositions, the first of which being that the sum of parameters of optimal arms tends to infinity (excluding the trivial case where all arms are equally good and Section 4.1 holds vacuously).

**Proposition 4.3** (Infinite Optimal Parameters). *If $\mathcal{A}^* \neq [K]$ then $\lim_{t \to \infty} \sum_{a \in \mathcal{A}^*} \theta_t(a) = \infty$ a.s.*

The second proposition states that all finite arms individually have their parameters diverge to negative infinity.

**Proposition 4.4** (Negative Infinite Suboptimal Parameters). *For every suboptimal arm $b \in [K] \setminus \mathcal{A}^*$, $\lim_{t \to \infty} \theta_t(b) = -\infty$ a.s.*

Equipped with these two propositions, the proof of Theorem 4.2 becomes straightforward enough that we need not resort to a proof sketch:

*Proof of Theorem 4.2.* If $\mathcal{A}^* = [K]$ then $\sum_{a \in \mathcal{A}^*} \pi_t(a) = 1$ for all $t \geq 0$ and the result holds vacuously. Henceforth suppose $\mathcal{A}^* \neq [K]$. We have that $\lim_{t \to \infty} \sum_{a \in \mathcal{A}^*} \pi_t(a) = 1 - \lim_{t \to \infty} \sum_{b \in [K] \setminus \mathcal{A}^*} \pi_t(b)$, so it suffices to show that, for all $b \in [K] \setminus \mathcal{A}^*$, $\lim_{t \to \infty} \pi_t(b) = 0$. To this end fix $b \in [K] \setminus \mathcal{A}^*$. We have the following bound from expanding the definition of $\pi_t$:

$$\lim_{t \to \infty} \pi_t(b) = \lim_{t \to \infty} \frac{\exp(\theta_t(b))}{\sum_{a \in [K]} \exp(\theta_t(a))} \qquad \text{(Eq. (2)) (6)}$$

$$\leq \lim_{t \to \infty} \frac{\exp(\theta_t(b))}{\sum_{a \in \mathcal{A}^*} \theta_t(a)} \qquad (\exp(x) \geq x \ , \ \mathcal{A}^* \subset [K]) \ (7)$$

$$= \frac{\lim_{t \to \infty} \exp(\theta_t(b))}{\lim_{t \to \infty} \sum_{a \in \mathcal{A}^*} \theta_t(a)} \ . \qquad (8)$$

Proposition 4.4 implies that the upper limit in Eq. (8) approaches 0 and Proposition 4.3 implies that the lower limit goes to infinity. Thus $\lim_{t \to \infty} \pi_t(b) = 0$, concluding the proof. □

The proofs of Propositions 4.3 and 4.4 are long and technical, and we refer the reader to the appendix for the details.

## 5 Reinforcement Learning

The results in RL depend on the results of Section 4, but in order to apply them we will need to port them to a slightly generalized bandit problem. We describe the necessary modifications in the following subsection, before proceeding to MDPs.

### 5.1 Nonstationary Bandit Setting

We still consider a $K$-armed bandit, with $K \geq 2$ and rewards in $[-R, R]$. The interaction between the learner and the environment is much the same as in Section 4.1, with the exception that now the reward distributions are allowed to change across timesteps. That is, we change out the distribution $P_a \in \mathcal{M}_1([-R, R])$ of rewards given that arm $a$ is played with a sequence of such distributions $(P_a^t)_{t \geq 1}$, and the reward at each iteration $t \geq 1$ is sampled from $P_{a_t}^t \in \mathcal{M}_1([-R, R])$; we also allow the expected rewards given that an arm is played to vary over time, so $r(a)$ becomes $(r^t(a))_{t \geq 1}$, and we have $\mathbb{E}[r_t | a_t = a] = r^t(a)$.

However, we constrain the setting in two ways. First, we suppose that there exists a filtration $(\mathcal{F}_t)_{t \geq 0}$ such that $P^t, r^t$ are $\mathcal{F}_{t-1}$-measurable and $a_t, r_t$ are $\mathcal{F}_t$-measurable. Intuitively, $\mathcal{F}_t$ contains the information available to the learner at iteration $t$, and this assumption means that the reward distributions (and thus their means) may only depend on the arms played and rewards observed up to the current timestep, as well as additional sources of randomness that are independent of the future. The second constraint on the environment is that we assume the existence of a "true" mean reward vector $r \in [-R, R]^K$, and suppose that there exists some random timestep $\tau$ such that, for all $t \geq \tau$ and all $a \in [K]$, $|r(a) - r^t(a)| \leq \Delta/3$, where $\Delta := \min_{a,b \in [K] \, : \, r(a) \neq r(b)} |r(a) - r(b)|$ is the minimum nonzero gap in the "true" mean reward between any two arms. This says that eventually the expected reward of playing arm $a$ will settle down to a neighbourhood of $r(a)$, and in particular that the arms in $\mathcal{A}^* := \arg\max_{a \in [K]} r(a)$ have the highest expected reward after iteration $\tau$. Given these modifications to the bandit setting, we can extend the results of Section 4 with minimal changes. The algorithm stays exactly the same, with the only modification to Algorithm 1 being that, at line 3, $r_{t+1} \sim P_{a_{t+1}}$ becomes $r_{t+1} \sim P_{a_{t+1}}^{t+1}$.

After extending all the bandit results to the nonstationary bandit setting, we can finally apply them for a result in RL.

## 5.2 Reinforcement Learning Setting

We consider a finite-horizon MDP, defined by the tuple $\mathcal{M} = (\mathcal{H}, \mathcal{S}, \mathcal{A}, \{r_h\}_{h=0}^{H-1}, \{P_h\}_{h=0}^{H-1}, \rho)$, where $\mathcal{H} = \{0, 1, \ldots, H-1\}$ is the index set of timesteps in an episode; $\mathcal{S} = \mathcal{S}_0 \cup \ldots \cup \mathcal{S}_{H-1}$ and $\mathcal{A} = \mathcal{A}_0 \cup \ldots \cup \mathcal{A}_{H-1}$ are finite state and action spaces, respectively, with $\mathcal{S}_h$ ($\mathcal{A}_h = \cup_{s \in \mathcal{S}_h} \mathcal{A}_s$) being the sets of possible states and (actions) at step $h \in \mathcal{H}$, and $\mathcal{A}_s$ is the set of possible actions from state s; $r_h : \mathcal{S}_h \times \mathcal{A}_h \to [-R, R]$ is a reward function that is bounded by $R > 0$; $P_h : \mathcal{S}_h \times \mathcal{A}_h \to \mathcal{M}_1(\mathcal{S}_{h+1})$ is the transition function; and $\rho : \mathcal{S}_0 \to \mathcal{M}_1(\mathcal{S}_0)$ is the initial state distribution. We denote $\pi := (\pi^h)_{h=0}^{H-1}$ as a time-dependent policy where $\pi^h : \mathcal{S}_h \to \mathcal{M}_1(\mathcal{A}_h)$ is the policy in the horizon $h$. An episode proceeds under the following protocol. At the beginning of the episode, the learner selects a non-stationary policy $\pi$. The episode then evolves through $s_0 \sim \rho$ and $a_h \sim \pi^h(\cdot|s_h), s_{h+1} \sim p_h(\cdot|s_h, a_h), r_h = r_h(a_h, s_h)$ for all $h \in \mathcal{H}$. We define the trajectory $\tau := (s_0, a_0, r_0, s_1, \ldots, s_{H-1}, a_{H-1}, r_{H-1})$. Therefore, the probability of a given trajectory $\tau$ is

$$\Pr(\tau) = \rho(s_0)\pi^0(a_0|s_0)p_0(s_1|s_0, a_0)\ldots\pi_{H-1}(a_{H-1}|s_{H-1}) \tag{9}$$

We also define the value functions and action-value functions for $h \in \mathcal{H}$

$$V_h^\pi(s) := \mathbb{E}^\pi\left[\sum_{h'=h}^{H-1} r_{h'} \middle| s_h = s\right] \tag{10}$$

$$Q_h^\pi(s, a) := \mathbb{E}^\pi\left[\sum_{h'=h}^{H-1} r_{h'} \middle| s_h = s, a_h = a\right] \tag{11}$$

The goal is to find a time-dependent policy $\pi^*$ that maximizes the state-value function at time 0, i.e. $V_0^\pi(\rho) := \mathbb{E}_{s \sim \rho}[V_0^\pi(s)]$:

$$\pi^* \in \arg\max_\pi V_0^\pi(\rho). \tag{12}$$

We also define optimal state and state-action value function, $V_h^* := V_h^{\pi^*}$ and $Q_h^* := Q_h^{\pi^*}$. In this paper, we focus on softmax parameterized policies. Specifically, we parameterize each $\pi^h$ by $\theta^h$ for all $h \in \mathcal{H}$ by

$$\pi_{\theta^h}^h(a|s) := \frac{\exp(\theta^h(s, a))}{\sum_{a'} \exp(\theta^h(s, a'))} \tag{13}$$

where $\theta^h \in \mathbb{R}^{\mathbb{A}_h}$ with $\mathbb{A}_h := \sum_{s \in \mathcal{S}_h} |\mathcal{A}_s|$ for all $h \in \mathcal{H}$. To improve the readability, we will sometimes write $\pi_t$ in place of $\pi_{\theta_t}$ and $\pi_t^h$ in place of $\pi_{\theta_t^h}^h$. The true gradient of the Eq. (12) is

$$\frac{\partial V_0^\pi(s)}{\partial \theta_t^h(s, a)} = \mathbb{E}^\pi\left[\frac{\partial}{\partial \theta_t^h(s, a)} \log \pi_t(a_h|s_h) \sum_{h=0}^{H-1} r_h\right] = \mathbb{E}^\pi\left[(\mathbb{I}[a_h = a] - \pi_t^h(a|s)) \sum_{h=0}^{H-1} r_h\right] \tag{14}$$

where $\mathbb{I}[a_h = a]$ is the indicator function of whether action $a$ is played in the horizon $h$, for all $s \in \mathcal{S}_h, a \in \mathcal{A}_h, h \in \mathcal{H}$. Since we are in the stochastic setting, we will use REINFORCE estimator to estimate the gradient and update the parameters

$$\frac{\hat{\partial} V_0^\pi(s)}{\partial \theta_h(s, a)} = \left(\sum_{h'=h}^{H-1} r_{h'}\right)\left(\mathbb{I}[a_h = a] - \pi_t^h(a|s)\right) \tag{15}$$

The REINFORCE algorithm is shown in Algorithm 2.

---

**Algorithm 2** REINFORCE

---

1: **for** each episode **do**
2:     Sample a trajectory $\tau$ using $\rho, \{\pi_{\theta_h}\}_{h=1}^{H-1}, \{P_h\}_{h=1}^{H-1}$
3:     **for** all $a \in |\mathcal{A}|, s \in |\mathcal{S}|$ **do**
4:         Use Eq. (15) to update $\theta(s, a)$
5:     **end for**
6: **end for**

---

We first show an exploration result, the counterpart to the exploration result shown above in the bandit setting, before sketching the proof of our main theorem in RL.

## 5.3  RL Exploration Lemma

**Lemma 5.1.** *Running REINFORCE with any $\eta \in \Theta(1)$ in a finite-horizon MDP $\mathcal{M}$, for all $h \in \mathcal{H}$, for all reachable $s \in \mathcal{S}_h$ and for all $a \in \mathcal{A}_s$ we have, almost surely, that every reachable state action pair will be visited i.o, i.e $N_\infty(s,a) = \infty$.*

*Proof sketch* 3. First, we show that for all horizon $h \in \mathcal{H}$, if $s \in \mathcal{S}_h$ is reachable and played i.o, then all actions $a \in \mathcal{S}$ are also played i.o by Lemma 4.1. Next, we use induction to show that for all horizon $h \in \mathcal{H}$, if $s \in \mathcal{S}_h$ is reachable visited i.o, then $s' \in \mathcal{S}_{h+1}$ is also visited i.o. Therefore, for all $h \in \mathcal{H}$, all reachable state-action pairs $(s,a) \in \mathcal{S} \times \mathcal{A}$ will be played i.o.

## 5.4  Convergence in finite-horizon MDP

**Theorem 5.2** (Convergence in RL). *For the MDP defined as above, using REINFORCE with constant learning rate $\eta \in \Theta(1)$, we have, almost surely, for all $s \in \mathcal{S}_0$, $V_0^{\pi_t}(s) \to V_0^*(s)$ as $t \to \infty$*

*Proof sketch* 4. We show the convergence theorem using the backward induction. Suppose for all horizon $h \in \{h', \ldots, H-1\}$, we have $\sum_{a \in \mathcal{A}_s^*} \pi_t^h(a|s) \to 1$ for all $s \in \mathcal{S}_h$, we want to show that $\sum_{a \in \mathcal{A}_s^*} \pi_t^{h-1}(a|s) \to 1$ for all $s \in \mathcal{S}_{h-1}$. Since $\sum_{a \in \mathcal{A}_s^*} \pi_t^h(a|s) \to 1$ for all $s \in \mathcal{S}_h$, where $h \in \{h', \ldots, H-1\}$, we know that there exists time step $\tau$ s.t $V_h^*(s) - V_h^{\pi_t}(s) \le \frac{\delta}{3}$, where $\delta$ is the minimum non-zero gap between two Q-values. For all $a \in \mathcal{A}_{s_{h-1}}$, there exists a minimum gap of $\frac{\delta}{3}$ in the Q-value. Therefore, applying the bandit convergence result, we know that $\sum_{a \in \mathcal{A}_s^*} \pi_t^{h-1}(a|s) \to 1$ as $t \to \infty$ for all $s \in \mathcal{S}_{h-1}$. Recursively, we know that for all $s \in \mathcal{S}_0$, $\sum_{a \in \mathcal{A}_s^*} \pi_{\theta_t}^0(a|s) \to 1$ as $t \to \infty$.

We also provide the statement and proof of convergence rate in the appendix (Theorem E.3).

## 5.5  Simulations

We conduct several experiments to illustrate the convergence behavior of REINFORCE algorithm in the finite-horizon setting. Experiments are performed using a chain MDP (Fig. 1b) with state space $\mathcal{S} = \{s_0, \ldots, s_3, T_1, T_2\}$, where $T_1$ and $T_2$ are terminal states, and action space $\mathcal{A} = \{a_0, a_1\}$. Taking action $a_0$ in any state yields a mean reward of $0.5$ and transitions to a terminal state $T_1$. Taking action $a_1$ in state $s_i$ ($i \in \{0, 1, 2\}$) yields a mean reward of $-0.5$ and transitions to state $s_{i+1}$. In state $s_3$, action $a_1$ yields a mean reward of $7$ and transitions to a terminal state $T_2$. The policy is parameterized using a softmax function, and parameters are initialized to $\mathbf{0} \in \mathbb{R}^{|\mathcal{S}| \times |\mathcal{A}|}$. For each learning rate $\eta$, the REINFORCE algorithm is run for $10^5$ episodes across 30 seeds. Performance is evaluated by measuring the average suboptimality gap from the initial state distribution $\rho$, defined as $V_0^*(\rho) - V_0^{\pi_t}(\rho)$, over the 30 seeds. Our first experiment (Fig. 3a) demonstrates the benefits of using a large learning rate. Previous convergence analysis of REINFORCE (Theorem 4.1 Klein et al., 2024) relies on small constant learning rates, which can significantly impede practical training speed. For instance, the analysis in Klein et al. (2024) guarantees convergence with $\eta = \frac{1}{5H^2R\sqrt{T}}$, where $T$ is the number of training episodes. In our environment ($H = 4$, $R = 7$, $T = 10^5$), this corresponds to an extremely small learning rate $\eta \approx 10^{-7}$. Therefore, we evaluated REINFORCE algorithm with larger learning rates $\eta \in \{0.00001, 0.001, 0.1\}$. Fig. 3a shows that the suboptimality gap remains nearly constant for $\eta = 0.00001$, whereas it decreases substantially faster as $\eta$ increases from $0.001$ to $0.1$. This demonstrates the practical benefit of employing larger learning rates for accelerated convergence, supported by our theoretical guarantees. We further explore the effect of even larger learning rates $\eta \in \{0.5, 1, 2\}$, presented in Fig. 3b. These rates, while potentially accelerating learning if updates are favorable, generally slow down convergence compared to the moderately large rates. The suboptimality curves exhibit more abrupt changes and show less consistent improvement over episodes. The large shaded regions indicate significantly higher variance with these very large learning rates. This suggests that large steps can easily push parameters away from optimal configurations, leading to prolonged exploration of suboptimal regions until a corrective update is sampled. Finally, Fig. 4 illustrates the evolution of the learned policy for optimal actions at each horizon. For all learning rates, we observe, on average, that the probability of selecting optimal actions converges first for the last horizon, then for the second-to-last, and so on, proceeding backward through the horizon. This backward convergence pattern in policy probabilities is consistent with our proof strategy for the convergence of the REINFORCE algorithm, which relies on a backward

induction approach. We also extend our experiments to demonstrate the relationship between the algorithm's performance and different learning rates. Details on the experimental setups and results can be found in the section Appendix F. Overall, we consistently find a "bowl-shaped" relationship between the learning rate and performance, meaning both exessively small and exessively large learning rates lead to high suboptimality, while middling values achieve the smallest suboptimality. The specific shape and optimal point of this bowl vary significantly with the environment's structure.

*Remark* 5.3. It is worth noting that not all environments exhibit this specific backward convergence pattern in learning optimal policy.

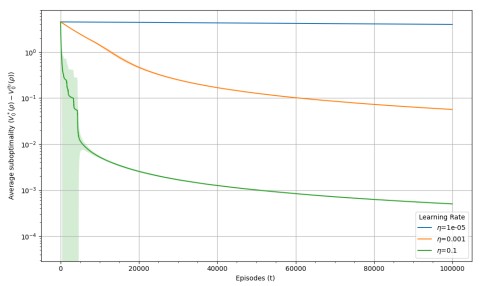 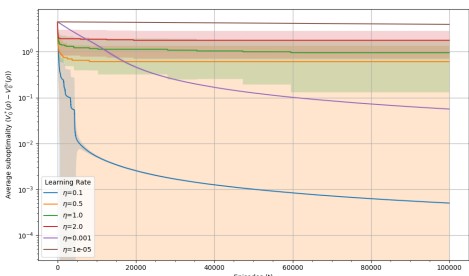

(a) Benefits of using larger learning rates.   (b) Drawbacks of using exessively large learning rates.

Figure 3: Fig. 3a shows that using a larger learning rate can improve the performance of REINFORCE, while Fig. 3b shows that excessively large learning rates have substantial variance, which can slow down the convergence rate.

# 6 Conclusions and Future work

This work enhances our understanding of the convergence properties of the widely used REINFORCE algorithm. Our novel proof offers deeper insights into the exploration effects of stochastic gradient methods and raises new research questions. Notably, recent findings by Mei et al. (2024a) indicate a convergence rate of $O(\log(t)/t)$ for stochastic gradient bandit algorithms. As demonstrated in Fig. 3b, REINFORCE with excessively large learning rates exhibits high variance, impeding convergence. Future work could explore optimal learning rate schedules to harness the initial benefits of larger rates while subsequently mitigating variance.

# 7 Acknowledgement and Disclosure of funding

The authors would like to thank Jeffrey Rosenthal for pointing out relevant results and materials regarding non-convergence results. Csaba Szepesvári and Dale Schuurmans gratefully acknowledge funding from the Canada CIFAR AI Chairs Program, Amii and NSERC.

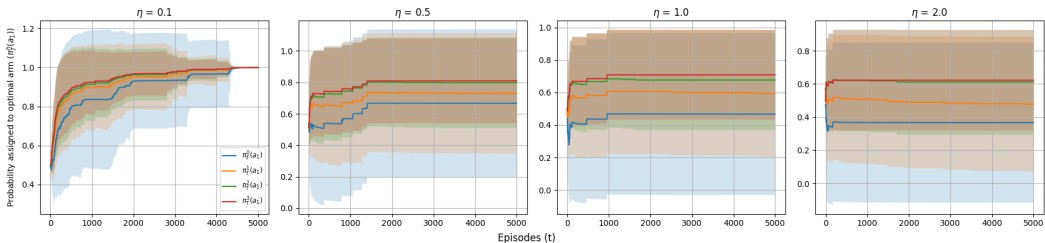

Figure 4: These figures show the convergence rate of the optimal policy in each horizon for different learning rates. In particular, we observe that the optimal policy of the last horizon will converge first, then the second-to-last one until the first horizon. This observation aligns with our analysis.

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

☞ In this appendix we will deal repeatedly with almost sure events, i.e. events that occur with probability 1. We typically mention this throughout the proofs, except for in one important case where "a.s." is omitted to reduce clutter: whenever statements involving conditional expectations (by extension conditional probabilities, variances) do not have an explicit probabilistic quantification, they are understood to hold almost surely. Of course, this is the only possible interpretation for such statements, since conditional expectations are only defined up to a set of measure 0.

## A    Technical Tools

We begin with some fundamental results from probability theory. The first is a generalization of the Borel-Cantelli Lemma.

**Lemma A.1** (Extended Borel-Cantelli Lemma, Corollary 5.29 of Breiman (1992)). *Given a filtration* $(\mathcal{F}_t)_{t\geq 0}$ *and a sequence of events* $(A_t)_{t\geq 0}$ *with* $A_t \in \mathcal{F}_t$ *for all* $t \geq 0$,

$$\sum_{t \geq 0} \mathbb{I}[A_t] = \infty \overset{a.s.}{\Longleftrightarrow} \sum_{t \geq 0} \mathbb{P}(A_t|\mathcal{F}_{t-1}) = \infty \,. \tag{16}$$

*That is,* $(A_t)_{t \geq 0}$ *occurs i.o. if and only if* $\sum_{t \geq 0} \mathbb{P}(A_t|\mathcal{F}_{t-1})$ *is infinite, up to a set of measure zero.*

Our analysis relies critically and repeatedly on the celebrated inequality of Freedman. The version we will use is similar to the one stated by Mei et al. (2024a,b). Since we require a general filtration, we include the original statement by Freedman below in Lemma A.2, followed by the statement and derivation of the form most convenient to us in Lemma A.3. Whenever we mention "Freedman's inequality" elsewhere in this work it shall refer to the latter.

**Lemma A.2** ((Original) Freedman's Inequality, Theorem 1.6 of Freedman (1975)). *Given a filtered probability space with filtration* $(\mathcal{F}_t)_{t\geq 0}$, *an adapted sequence of random variables* $(X_t)_{t\geq 1}$, *and constants* $a, b > 0$, *if* $\forall t \geq 1 : \mathbb{E}[X_t|\mathcal{F}_{t-1}] = 0$ *and* $|X_t| \leq 1$ *then*

$$\mathbb{P}\Big(\exists t \geq 1 \,:\, \sum_{i \in [t]} X_i \geq a \,,\, \sum_{i \in [t]} \mathrm{Var}[X_i|\mathcal{F}_{i-1}] \leq b\Big) \leq \exp\left(\frac{-a^2}{2(a+b)}\right). \tag{17}$$

**Lemma A.3** (Freedman's Inequality). *Let* $(X_t)_{t\geq 1}$ *be a random sequence adapted to the filtration* $(\mathcal{F}_t)_{t\geq 0}$, $B \geq 0$ *be a constant such that* $\forall t \geq 0 : |X_t| \leq B$, *and denote* $V_t := \sum_{i \in [t]} \mathrm{Var}[X_i|\mathcal{F}_{i-1}]$. *For any* $\delta \in (0, 1]$, *it holds with probability* $1 - \delta$ *that*

$$\forall t \geq 1 \,:\, \Big| \sum_{i \in [t]} X_i - \mathbb{E}[X_i|\mathcal{F}_{i-1}]\Big| \leq 20\sqrt{V_t + 4B^2 + 1}\log\left(\frac{V_t + 2}{\delta}\right). \tag{18}$$

*Remark* A.4. The derivation of Lemma A.3 closely follows the proof of Theorem C.3 of Mei et al. (2024b). We aimed for a simple bound rather than a tight one.

*Proof.* Fix $\epsilon \in (0, 1)$, and let $S_t := \sum_{i \in [t]} X_i$ and $V_t := \sum_{i \in [t]} \mathrm{Var}[X_i|\mathcal{F}_{i-1}]$. First we will suppose that $\mathbb{E}[X_t|\mathcal{F}_{t-1}] = 0$ and $|X_t| \leq 1$ for all $t \geq 1$, and show that

$$\mathbb{P}\Big(\exists t \geq 1 \,:\, S_t \geq 10\sqrt{V_t + 1}\log\left(\frac{V_t + 2}{\epsilon}\right)\Big) \leq \epsilon. \tag{19}$$

For $x \geq 1$ let $g(x) := 3\log((x+2)^2/\epsilon)$, and we have

$$g(x) + \sqrt{g(x)x} \tag{20}$$

$$\leq 6\log((x+2)/\epsilon) + \sqrt{3x}\sqrt{\log((x+2)^2/\epsilon)} \qquad ((x+2)^2/\epsilon \leq (x+2)^2/\epsilon^2) \tag{21}$$

$$\leq 6\log((x+2)/\epsilon) + \sqrt{3x}\log((x+2)^2/\epsilon) \qquad (\log((x+2)^2/\epsilon) \geq \log(4) \geq 1) \tag{22}$$

$$\leq 6\log((x+2)/\epsilon) + 2\sqrt{3x}\log((x+2)/\epsilon) \qquad ((x+2)^2/\epsilon \leq (x+2)^2/\epsilon^2) \tag{23}$$

$$\leq 6\log((x+2)/\epsilon) + 4\sqrt{x+1}\log((x+2)/\epsilon) \tag{24}$$

$$\leq 10\sqrt{x+1}\log((x+2)/\epsilon). \tag{25}$$

Setting $x := V_t$ in Eq. (25) yields

$$\mathbb{P}\Big(\exists t \geq 1 \,:\, S_t \geq 10\sqrt{V_t+1}\log\Big(\frac{V_t+2}{\epsilon}\Big)\Big) \tag{26}$$

$$\leq \mathbb{P}\big(\exists t \geq 1 \,:\, S_t \geq g(V_t) + \sqrt{g(V_t)V_t}\big) \tag{27}$$

$$= \sum_{i\geq 0} \mathbb{P}\big(\exists t \geq 1 \,:\, S_t \geq g(V_t) + \sqrt{g(V_t)V_t}\,,\, i \leq V_t < i+1\big) \tag{28}$$

$$= \sum_{i\geq 0} \mathbb{P}\big(\exists t \geq 1 \,:\, S_t \geq g(i) + \sqrt{g(i)i}\,,\, V_t \leq i+1\big) \qquad (g\,,\, \sqrt{\cdot}\ \text{are increasing}) \tag{29}$$

$$\leq \sum_{i\geq 0} \exp\Big(-\frac{\big(g(i)+\sqrt{g(i)i}\big)^2}{2\big(g(i)+\sqrt{g(i)i}+i+1\big)}\Big)\,.$$
$$\text{(Lemma A.2 with } a := g(i) + \sqrt{g(i)i} \text{ and } b := i+1) \tag{30}$$

To control the term appearing in the $\exp$ above, we will use the following inequality, which holds for $u \geq 2$ and $i \geq 0$:

$$\frac{\big(u+\sqrt{ui}\big)^2}{2\big(u+\sqrt{ui}+i+1\big)} = \frac{u\big(u+2\sqrt{ui}+i\big)}{2\big(u+\sqrt{ui}+i+1\big)} \tag{31}$$

$$= \frac{u}{3} \cdot \frac{2u+6\sqrt{ui}+3i+u}{2u+2\sqrt{ui}+2i+2} \tag{32}$$

$$\geq u/3\,. \qquad (u \geq 2) \tag{33}$$

Since $g(i) \geq 3\log(4) \geq 2$, we can combine the above two displays by setting $u := g(i)$ and conclude

$$\mathbb{P}\Big(\exists t \geq 1 \,:\, S_t \geq 10\sqrt{V_t+1}\log\Big(\frac{V_t+2}{\epsilon}\Big)\Big) \leq \sum_{i\geq 0} \exp(-g(i)/3) \tag{34}$$

$$= \epsilon \sum_{i\geq 0} \frac{1}{(i+2)^2} \tag{35}$$

$$= \epsilon \sum_{i\geq 2} i^{-2} \tag{36}$$

$$= \epsilon(\pi^2/6 - 1) \qquad (\textstyle\sum_{i\geq 1} i^{-2} = \pi^2/6) \tag{37}$$

$$\leq \epsilon\,. \tag{38}$$

We are finished showing Eq. (19). We can apply this result to both $(X_t)_{t\geq 0}$ and $(-X_t)_{t\geq 0}$, setting $\epsilon := \delta/2$ in each application, whence a union bound guarantees that, with probability at least $1 - \delta$,

$$\forall t \geq 1 \,:\, \sum_{i\in[t]} |X_i| < 10\sqrt{V_t+1}\log\Big(\frac{V_t+2}{\delta/2}\Big) \tag{39}$$

$$\leq 10\sqrt{V_t+1}\Big(\log\Big(\frac{V_t+2}{\delta}\Big) + \log(2)\Big) \tag{40}$$

$$\leq 20\sqrt{V_t+1}\log\Big(\frac{V_t+2}{\delta}\Big)\,. \tag{41}$$

Given a random sequence $(X_t)_{t\geq 1}$ that satisfies $|X_t| \leq B$ for some $B \geq 1$, we can apply Eq. (41) to the sequence $(X_t/B)_{t\geq 1}$:

$$\forall t \geq 1 \,:\, \sum_{i\in[t]} |X_i/B| < 20\sqrt{V_t/B^2+1}\log\Big(\frac{V_t/B^2+2}{\delta}\Big) \tag{42}$$

$$\leq 20\sqrt{V_t/B^2+1}\log\Big(\frac{V_t+2}{\delta}\Big) \qquad (V_t/B^2 \leq V_t) \tag{43}$$

$$\Rightarrow \sum_{i\in[t]} |X_i| \leq 20\sqrt{V_t+B^2}\log\Big(\frac{V_t+2}{\delta}\Big)\,, \tag{44}$$

with probability $1 - \delta$. Combining Eq. (41), which holds for $|X_t| \leq 1$, and Eq. (44), which holds for $|X_t| \leq B$ where $B \geq 1$, and upper bounding $\max(B^2, 1) \leq B^2 + 1$, we can remove the requirement that $B \geq 1$ and conclude that, if $\mathbb{E}[X_t | \mathcal{F}_{t-1}] = 0$ and $|X_t| \leq B$ for all $t \geq 1$, with probability $1 - \delta$

$$\forall t \geq 1 : \sum_{i \in [t]} |X_i| \leq 20\sqrt{V_t + B^2 + 1} \log\left(\frac{V_t + 2}{\delta}\right). \tag{45}$$

To remove the assumption that $\forall t \geq 1 : \mathbb{E}[X_t | \mathcal{F}_{t-1}] = 0$ and get the desired result, we apply Eq. (45) to $(X_t - \mathbb{E}[X_t | \mathcal{F}_{t-1}])_{t \geq 1}$, noting that if $|X_t| \leq B$ then $|X_t - \mathbb{E}[X_t | \mathcal{F}_{t-1}]| \leq 2B$. □

The following result applies Freedman's Inequality to a sequence of bounded, and eventually (conditionally) self-bounded, random variables. It says that if the conditional expectations are not summable then the variables themselves will not be summable. We expect that the result is folklore, but cannot find a reference.

**Lemma A.5** (Freedman Divergence Trick). *Let $(X_t)_{t \geq 1}$ be a random sequence adapted to the filtration $(\mathcal{F}_t)_{t \geq 0}$ and $B \geq 0$ be a constant such that $\forall t \geq 0 : |X_t| \leq B$. Suppose $\sum_{t \geq 1} \mathbb{E}[X_t | \mathcal{F}_{t-1}] = \infty$ and, for some random (a.s. finite) index $\tau \geq 1$ and constant $C \geq 0$, for all $t \geq \tau$, $\mathrm{Var}[X_t | \mathcal{F}_{t-1}] \leq C\mathbb{E}[X_t | \mathcal{F}_{t-1}]$. Then $\sum_{t \geq 1} X_t = \infty$ a.s.*

*Remark* A.6. Note that the result does not require $\tau$ to be a stopping time.

*Proof.* For $t \geq 0$ let $S_t := \sum_{i \in [t]} X_i$, $\overline{S}_t := \sum_{i \in [t]} \mathbb{E}[X_i | \mathcal{F}_{i-1}]$, and $V_t := \sum_{i \in [t]} \mathrm{Var}[X_i | \mathcal{F}_{i-1}]$. For any $\delta \in (0, 1]$, we can apply Freedman's Inequality (Lemma A.3) to $(X_t)_{t \geq 1}$. This gives that, with probability $1 - \delta$, for any $t \geq \tau$,

$$S_t \geq \overline{S}_t - 20\sqrt{V_t + 4B^2 + 1} \log\left(\frac{V_t + 2}{\delta}\right) \tag{46}$$

$$= \overline{S}_t - \overline{S}_\tau + \overline{S}_\tau - 20\sqrt{V_t - V_\tau + V_\tau + 4B^2 + 1} \log\left(\frac{V_t - V_\tau + V_\tau + 2}{\delta}\right) \tag{47}$$

$$\geq \overline{S}_t - \overline{S}_\tau - \tau B - 20\sqrt{V_t - V_\tau + (4 + \tau)B^2 + 1} \log\left(\frac{V_t - V_\tau + \tau B^2 + 2}{\delta}\right) \tag{(|X_t| \leq B) (48)}$$

$$\geq \overline{S}_t - \overline{S}_\tau - \tau B - 20\sqrt{C(\overline{S}_t - \overline{S}_\tau) + (4 + \tau)B^2 + 1} \log\left(\frac{C(\overline{S}_t - \overline{S}_\tau) + \tau B^2 + 2}{\delta}\right). \tag{(\mathrm{Var}[X_i | \mathcal{F}_{i-1}] \leq C\mathbb{E}[X_i | \mathcal{F}_{i-1}] \text{ for } t \geq i \geq \tau) (49)}$$

By assumption $\lim_t \overline{S}_t = \infty$, so $\lim_t \overline{S}_t - S_\tau = \infty$ as well. Clearly, the subtrahend in the display above is $o(\overline{S}_t - \overline{S}_\tau)$. Hence, taking the limit of $t \to \infty$, we have $\lim_t S_t = \infty$ with probability $1 - \delta$. Since $\delta$ was arbitrary, this also holds with probability one (by taking $\delta \to 0$). □

Finally, we will need a classic result of Doob.

**Lemma A.7** (Doob's Martingale Convergence Theorem (Doob, 2012)). *Given a random sequence $(X_t)_{t \geq 1}$ adapted to the filtration $(\mathcal{F}_t)_{t \geq 0}$, if $\forall t \geq 1 : \mathbb{E}[X_t | \mathcal{F}_{t-1}] \leq X_{t-1}$ and $\sup_{t \geq 0} \mathbb{E}[-\min(X_t, 0)] < \infty$, then $(X_t)_{t \geq 1}$ converges a.s. In particular, $X_t \to X$ a.s. as $t \to \infty$, where $X := \limsup_t X_t$ and $\mathbb{E}[|X|] < \infty$.*

# B  Bandits

In this section all results are stated in the bandit setting described in Section 4.1. The first subsection establishes the results needed for the proof of convergence in bandits, Theorem 4.2. The second subsection proves that convergence is not always to a one-hot policy, Proposition 3.2

## B.1  Convergence

We begin with a simple but crucial property of Algorithm 1, which follows from a symmetry of the update rule.

**Lemma B.1** (Conservation of mass). *For all $t \geq 0$, $\sum_{a \in [K]} \theta_t(a) = \sum_{a \in [K]} \theta_0(a)$.*

*Proof.* Proceeding by induction, the base is tautological; recalling that $a_t$ is the arm played at time $t$, we have

$$\sum_{a \in [K]} \theta_{t+1}(a) = \theta_{t+1}(a_t) + \sum_{a \in [K] \setminus \{a_t\}} \theta_{t+1}(a) \tag{50}$$

$$= \theta_t(a_t) + \eta(1 - \pi_t(a_t)) r_t(a_t) + \sum_{a \in [K] \setminus \{a_t\}} [\theta_t(a) - \eta \pi_t(a) r_t(a_t)] \tag{51}$$

$$= \eta r_t(a_t) + \sum_{a \in [K]} [\theta_t(a) - \eta \pi_t(a) r_t(a_t)] \tag{52}$$

$$= \eta r_t(a_t) + \sum_{a \in [K]} \theta_t(a) - \eta r_t(a_t) \sum_{a \in [K]} \pi_t(a) \tag{53}$$

$$= \sum_{a \in [K]} \theta_t(a). \qquad (\textstyle\sum_{a \in [K]} \pi_t(a) = 1) \tag{54}$$

$\square$

The rest of the proofs in this section will refer to the filtration $(\mathcal{F}_t)_{t \geq 0}$ defined by $\mathcal{F}_t := \sigma((a_i, r_i)_{i < t})$, and we adopt the shorthands $\mathbb{E}_t[\cdot] := \mathbb{E}[\cdot | \mathcal{F}_t]$ and $\mathrm{Var}_t[\cdot] := \mathrm{Var}[\cdot | \mathcal{F}_t]$. The following result is a stronger version of Lemma 2 of Mei et al. (2024a), and it guarantees that Algorithm 1 explores enough to keep trying all arms forever regardless of the observations.

**Lemma 4.1** (Bandit Exploration). *Using Algorithm 1 with any constant learning rate $\eta \in \Theta(1)$, every arm is almost surely played infinitely often. That is, $\forall a \in [K] : N_\infty(a) = \infty$ almost surely.*

*Proof.* The first step is to show that, for any arm $b \in [K]$, if $|\{t \geq 0 : a_t = b\}| < \infty$ then $\sup_t |\theta_t(b)| < \infty$ a.s. Picking $b \in [K]$ and setting $m := \sup(\{0\} \cup \{t \geq 0 : a_t = b\})$, without assuming $|\{t \geq 0 : a_t = b\}| < \infty$ we have the bound

$$\sup_t |\theta_t(b)| \leq |\theta_0(b)| + \sup_t \sum_{i \in [t]} |\theta_i(b) - \theta_{i-1}(b)| \qquad \text{(triangle inequality)} \tag{55}$$

$$\leq |\theta_0(b)| + \sum_{i \geq 1} |\theta_i(b) - \theta_{i-1}(b)| \qquad (\textstyle\sum_{i > t} |\theta_i(b) - \theta_{i-1}(b)| > 0) \tag{56}$$

$$= |\theta_0(b)| + \sum_{i \in [m]} |\theta_i(b) - \theta_{i-1}(b)| + \sum_{i > m} |\theta_i(b) - \theta_{i-1}(b)| \tag{57}$$

$$\leq |\theta_0(b)| + \sum_{i \in [m]} \eta R + \sum_{i > m} \eta R \pi_{\theta_i}(b) \qquad \text{(update rule of Algorithm 1)} \tag{58}$$

$$\leq |\theta_0(b)| + \eta R \big( m + \sum_{t \geq 0} \pi_t(b) \big) \qquad (\textstyle\sum_{i=0}^{m} \eta R \pi_{\theta_i}(b) > 0) \tag{59}$$

$$=: \alpha(b). \tag{60}$$

Also, the Extended Borel-Cantelli Lemma (Lemma A.1) applied to $(\mathcal{F}_t)_{t \geq 0}$ with the event sequence $A_t := \{a_t = b\}$ implies

$$\sum_{t \geq 0} \mathbb{I}[a_t = b] < \infty \overset{\text{a.s.}}{\Longleftrightarrow} \sum_{t \geq 0} \pi_t(b) < \infty. \tag{61}$$

If $|\{t \geq 0 : a_t = b\}| < \infty$ then $m < \infty$ and $\sum_{t \geq 0} \mathbb{I}[a_t = b] < \infty$, and the latter inequality together with Eq. (61) implies $\sum_{t \geq 0} \pi_t(b) < \infty$ a.s, thus Eq. (60) yields $\sup_t |\theta_t(b)| \leq \alpha(b) < \infty$ a.s. A union bound over $b \in [K]$ implies that almost surely

$$\forall b \in [K] : |\{t \geq 0 : a_t = b\}| < \infty \implies \alpha(b) < \infty. \tag{62}$$

We are ready to fix an arm $a \in [K]$ and show that the event $\mathcal{E} := \big\{ |\{t \geq 0 : a_t = a\}| < \infty \big\}$ has probability 0. For the remainder of the proof until the almost the very end we will work under the

assumption that $\mathcal{E}$ occurs. On $\mathcal{E}$ we have $\alpha(a) < \infty$ a.s, which implies $\sum_{t \geq 0} \pi_t(a) < \infty$ a.s, which in turn implies $\lim_t \pi_t(a) = 0$ a.s. The definition of $\pi_t(a)$ gives us

$$\lim_t \pi_t(a) = \lim_t \frac{\exp(\theta_t(a))}{\sum_{b \in [K]} \exp(\theta_t(b))} \qquad \text{(Eq. (2)) (63)}$$

$$\geq \lim_t \frac{\exp(-\alpha(a))}{\sum_{b \in [K]} \exp(\theta_t(b))} \qquad (\alpha(a) \geq |\theta_t(a)|) \text{ (64)}$$

$$\geq \lim_t \frac{\exp(-\alpha(a))}{K \exp(\max_{b \in [K]} \theta_t(b))}, \quad (\sum_{b \in [K]} \exp(\theta_t(b)) \leq K \exp(\max_{b \in [K]} \theta_t(b))) \text{ (65)}$$

so from $\lim_t \pi_t(a) = 0$ a.s. we get $\lim_t \max_{b \in [K]} \theta_t(b) = \infty$ a.s. Then conservation of mass (Lemma B.1) implies that $\lim_t \min_{b \in [K]} \theta_t(b) = -\infty$ a.s. By Eq. (62) all arms that are selected only finitely often have parameters bounded away from $-\infty$ a.s, so there is a.s. an arm $b$ that is played i.o. with $\liminf_t \theta_t(b) = -\infty$. We will refer to such an arm as $b$ for the remainder of the proof. However, because $b$ is played i.o, another application of the Extended Borel-Cantelli Lemma (to $(\mathcal{F}_t)_{t \geq 0}$ with events $A_t := \{a_t = b\}$) yields $\sum_{t \geq 0} \pi_t(b) = \infty$ a.s. Since $\sum_{t \geq 0} \pi_t(a) < \infty$ a.s, we have that $\pi_t(b) > \pi_t(a)$, and equivalently $\theta_t(b) > \theta_t(a)$, for infinitely many $t \geq 0$ a.s. In summary, $\theta_t(b)$ oscillates from being arbitrarily low to being larger than $\theta_t(a) \geq -\alpha(a)$.[2]

We will now argue that, for sufficiently large $t$, if $\theta_t(b) \leq \theta_t(a)$ but $\theta_{t+1}(b) > \theta_{t+1}(a)$ then $a_t = b$. Let $T$ be the minimum timestep such that, for all $t \geq T$,

$$\max_{c \in [K]} \theta_t(c) \geq \log(\eta R) + \alpha(a), \quad \text{and} \quad a_t \neq a. \qquad (66)$$

Since we are working on the event $\mathcal{E}$ we have $a_t = a$ for only finitely many $t$, $\log(\eta R) + \alpha(a) < \infty$ a.s, and $\lim_t \max_{c \in [K]} \theta_t(c) = \infty$ a.s; taken together, these observations imply that $T < \infty$ exists a.s.

For $t \geq T$, suppose $\theta_t(b) \leq \theta_t(a)$ and $a_t \neq b$, and we must show

$$\theta_{t+1}(b) \leq \theta_{t+1}(a) \qquad (67)$$
$$\iff \quad \theta_t(b) - \eta \pi_t(b) r_t(a_t) \leq \theta_t(a) - \eta \pi_t(a) r_t(a_t) \qquad (a_t \notin \{a, b\}) \text{ (68)}$$
$$\iff \quad \eta r_t(a_t)(\pi_t(a) - \pi_t(b)) \leq \theta_t(a) - \theta_t(b). \qquad (69)$$

Since $\theta_t(b) \leq \theta_t(a)$ we have $\pi_t(b) \leq \pi_t(a)$, and standard inequalities yield

$$\eta r_t(a_t)(\pi_t(a) - \pi_t(b)) \leq \eta R(\pi_t(a) - \pi_t(b)) \qquad (0 \leq \pi_t(a) - \pi_t(b)) \text{ (70)}$$

$$= \eta R \frac{\exp(\theta_t(a)) - \exp(\theta_t(b))}{\sum_{c \in [K]} \exp(\theta_t(c))} \qquad (71)$$

$$\leq \eta R \frac{\exp(\theta_t(a)) - \exp(\theta_t(b))}{\exp(\max_{c \in [K]} \theta_t(c))}$$
$$\qquad\qquad (\sum_{c \in [K]} \exp(\theta_t(c)) \geq \exp(\max_{c \in [K]} \theta_t(c))) \text{ (72)}$$

$$\leq \frac{\exp(\theta_t(a)) - \exp(\theta_t(b))}{\exp(\theta_t(a))} \qquad (\text{Eq. (66)}, \alpha(a) \geq \theta_t(a)) \text{ (73)}$$

$$= 1 - \exp(\theta_t(b) - \theta_t(a)) \qquad (74)$$
$$\leq 1 - (1 + \theta_t(b) - \theta_t(a)). \qquad (\exp(x) \geq 1 + x) \text{ (75)}$$

Thus Eq. (69) holds and we have established that, for all $t \geq T$, if $\theta_t(b) \leq \theta_t(a)$ and $\theta_{t+1}(b) > \theta_{t+1}(a)$ then $a_t = b$. Since $\theta_t(b)$ fluctuates from below $\theta_t(a)$ to above it i.o, we have that the events in the sequence $(B_t)_{t \geq 0}$ defined by $B_t := \{\theta_t(b) \leq \theta_t(a), a_t = b\} \in \mathcal{F}_{t+1}$ occur i.o. a.s. Applying the Extended Borel-Cantelli Lemma to $(\mathcal{F}_t)_{t \geq 0}$ and $(B_t)_{t \geq 0}$ implies that $\sum_{t \geq 0} \mathbb{P}(B_t | \mathcal{F}_t) = \infty$ a.s. However,

$$\mathbb{P}(B_t | \mathcal{F}_t) = \mathbb{I}[\theta_t(b) \leq \theta_t(a)] \pi_t(b) \leq \pi_t(a), \qquad (76)$$

so $\sum_{t \geq 0} \mathbb{P}(B_t | \mathcal{F}_t) \leq \sum_{t \geq 0} \pi_t(a) < \infty$ a.s.

At this point we have that, on event $\mathcal{E}$, both $\sum_{t \geq 0} \mathbb{P}(B_t | \mathcal{F}_t) < \infty$ a.s. and $\sum_{t \geq 0} \mathbb{P}(B_t | \mathcal{F}_t) = \infty$ a.s. Since these events are mutually exclusive they both occur with probability 0, and since they are jointly exhaustive we have $\mathbb{P}(\mathcal{E}) = 0$. $\qquad \square$

---

[2]It is easy to see that $\theta_t(b)$ must also become arbitrarily large i.o, but this is not necessary for the proof.

The proof of our main result in the bandit setting, that $\lim_{t\to\infty} \sum_{a\in\mathcal{A}^*} \pi_t(a) = 1$, is broken into two propositions: the first guarantees that $\lim_{t\to\infty} \sum_{a\in\mathcal{A}^*} \theta_t(a) = \infty$, in particular that as the time steps get large at least one $a \in \mathcal{A}^*$ will have an arbitrarily large parameter[3]; the second proposition says that $\lim_{t\to\infty} \theta_t(b) = -\infty$ for all $b \in [K] \setminus \mathcal{A}^*$. Taken together, the propositions imply that eventually some (potentially time step dependent) optimal arm dominates every suboptimal arm, establishing Convergence in Bandits (Theorem 4.2). We now turn to proving the two propositions.

The subsequent proofs will go a little smoother with some extra notation; we define $\Delta := \min_{a,b\in[K]:r(a)\neq r(b)} |r(a) - r(b)|$ to be the minimum nonzero gap between expected rewards of arms and $r(\mathcal{A}^*) := \max_{a\in[K]} r(a)$ to be the maximum attainable expected reward. Finally, we overload $\pi_t(\cdot)$ to take sets as input, i.e, given $\mathcal{S} \subset [K]$ we let $\pi_t(\mathcal{S}) := \sum_{a\in\mathcal{S}} \pi_t(a)$ to be the probability that an arm in $\mathcal{S}$ is selected. With these abbreviations in hand, the first proposition is as follows.

**Proposition 4.3** (Infinite Optimal Parameters). *If $\mathcal{A}^* \neq [K]$ then $\lim_{t\to\infty} \sum_{a\in\mathcal{A}^*} \theta_t(a) = \infty$ a.s.*

*Proof.* For $t \geq 0$, let $X_t := \sum_{a\in\mathcal{A}^*} \theta_{t+1}(a) - \theta_t(a)$, such that $\sum_{i=0}^t X_i = \sum_{a\in\mathcal{A}^*} \theta_{t+1}(a) - \theta_0(a)$. By the update rule of Algorithm 1, note also that

$$X_t = \eta \sum_{a\in\mathcal{A}^*} (\mathbb{I}[a_t = a] - \pi_t(a))r_t . \tag{77}$$

The conditional expectation of $X_t$ given $\mathcal{F}_t$ can be lower bounded by

$$\mathbb{E}_t[X_t] = \sum_{a\in[K]} \mathbb{E}_t[\mathbb{I}[a_t = a]X_t] \qquad\qquad (\textstyle\sum_{a\in[K]}\mathbb{I}[a_t=a]=1) \tag{78}$$

$$= \sum_{a\in\mathcal{A}^*} \mathbb{E}_t[\mathbb{I}[a_t = a]\eta(1 - \pi_t(\mathcal{A}^*))r_t] + \sum_{b\in[K]\setminus\mathcal{A}^*} \mathbb{E}_t[\mathbb{I}[a_t = b]\eta(-\pi_t(\mathcal{A}^*))r_t]$$
$$\text{(Eq. (77)) (79)}$$

$$= \eta(1 - \pi_t(\mathcal{A}^*)) \sum_{a\in\mathcal{A}^*} \mathbb{E}_t[\mathbb{I}[a_t = a]r_t] - \eta\pi_t(\mathcal{A}^*) \sum_{b\in[K]\setminus\mathcal{A}^*} \mathbb{E}_t[\mathbb{I}[a_t = b]r_t]$$
$$(\pi_t \text{ is } \mathcal{F}_t\text{-measurable) (80)}$$

$$= \eta(1 - \pi_t(\mathcal{A}^*)) \sum_{a\in\mathcal{A}^*} \pi_t(a)r(a) - \eta\pi_t(\mathcal{A}^*) \sum_{b\in[K]\setminus\mathcal{A}^*} \pi_t(b)r(b)$$
$$(\mathbb{E}_t[\mathbb{I}[a_t = \cdot]r_t] = \pi_t(\cdot)r(\cdot)) (81)$$

$$\geq \eta(1 - \pi_t(\mathcal{A}^*)) \sum_{a\in\mathcal{A}^*} \pi_t(a)r(\mathcal{A}^*) - \eta\pi_t(\mathcal{A}^*) \sum_{b\in[K]\setminus\mathcal{A}^*} \pi_t(b)(r(\mathcal{A}^*) - \Delta)$$
$$(r(a) = r(\mathcal{A}^*) , \ r(b) \leq r(\mathcal{A}^*) - \Delta) (82)$$

$$= \eta\pi_t(\mathcal{A}^*)(1 - \pi_t(\mathcal{A}^*))\big(r(\mathcal{A}^*) - (r(\mathcal{A}^*) - \Delta)\big)$$
$$(\textstyle\sum_{a\in\mathcal{A}^*}\pi_t(a) = \pi_t(\mathcal{A}^*), \sum_{b\in[K]\setminus\mathcal{A}^*}\pi_t(b) = 1 - \pi_t(\mathcal{A}^*)) (83)$$

$$= \eta\pi_t(\mathcal{A}^*)(1 - \pi_t(\mathcal{A}^*))\Delta , \tag{84}$$

---

[3]Excluding the trivial case where $\mathcal{A}^* = [K]$, i.e. all arms are equally good.

and the conditional variance can be upper bounded by

$$\text{Var}_t[X_t] \leq \mathbb{E}_t[X_t^2] \tag{85}$$

$$= \sum_{a \in [K]} \mathbb{E}_t[\mathbb{I}[a_t = a] X_t^2] \qquad (\textstyle\sum_{a \in [K]} \mathbb{I}[a_t = a] = 1) \tag{86}$$

$$= \sum_{a \in \mathcal{A}^*} \mathbb{E}_t[\mathbb{I}[a_t = a]\big(\eta(1 - \pi_t(\mathcal{A}^*))r_t\big)^2] + \sum_{b \in [K] \setminus \mathcal{A}^*} \mathbb{E}_t[\mathbb{I}[a_t = b]\big(-\eta\pi_t(\mathcal{A}^*)r_t\big)^2]$$
$$\text{(Eq. (77))} \tag{87}$$

$$\leq \eta^2(1 - \pi_t(\mathcal{A}^*))^2 R^2 \sum_{a \in \mathcal{A}^*} \mathbb{E}_t[\mathbb{I}[a_t = a]] + \eta^2 \pi_t(\mathcal{A}^*)^2 R^2 \sum_{b \in [K] \setminus \mathcal{A}^*} \mathbb{E}_t[\mathbb{I}[a_t = b]]$$
$$(r_t^2 \leq R^2) \tag{88}$$

$$= \eta^2(1 - \pi_t(\mathcal{A}^*))^2 R^2 \sum_{a \in \mathcal{A}^*} \pi_t(a) + \eta^2 \pi_t(\mathcal{A}^*)^2 R^2 \sum_{b \in [K] \setminus \mathcal{A}^*} \pi_t(b)$$
$$(\mathbb{E}_t[\mathbb{I}[a_t = \cdot]] = \pi_t(\cdot)) \tag{89}$$

$$= \eta^2 R^2 \Big((1 - \pi_t(\mathcal{A}^*))^2 \pi_t(\mathcal{A}^*) + \pi_t(\mathcal{A}^*)^2 (1 - \pi_t(\mathcal{A}^*))\Big)$$
$$(\textstyle\sum_{a \in \mathcal{A}^*} \pi_t(a) = \pi_t(\mathcal{A}^*), \ \sum_{b \in [K] \setminus \mathcal{A}^*} \pi_t(b) = 1 - \pi_t(\mathcal{A}^*)) \tag{90}$$

$$= \eta^2 R^2 \pi_t(\mathcal{A}^*)(1 - \pi_t(\mathcal{A}^*)). \tag{91}$$

Thus for all $t \geq 0$ we have $\text{Var}_t[X_t] \leq \eta R^2 \Delta^{-1} \mathbb{E}_t[X_t]$, $|X_t| \leq \eta R$, and $X_t$ is $\mathcal{F}_{t+1}$-measurable. Setting $b := \eta R$, $\tau := 0$, and $c := \eta R^2 \Delta^{-1}$, we need only to prove that $\sum_{t \geq 0} \mathbb{E}_t[X_t] = \infty$, at which point we can apply the Freedman Divergence Trick (Lemma A.5) to conclude

$$\lim_t \sum_{a \in \mathcal{A}^*} \theta_{t+1}(a) - \theta_0(a) = \sum_{t \geq 0} X_t = \infty \text{ a.s,} \tag{92}$$

$$\implies \lim_t \sum_{a \in \mathcal{A}^*} \theta_t(a) = \infty \text{ a.s.} \qquad (\textstyle\sum_{a \in \mathcal{A}^*} \theta_0(a) < \infty) \tag{93}$$

Thus in the remainder of the proof we turn our attention to showing $\sum_{t \geq 0} \mathbb{E}_t[X_t] = \infty$. Applying Eq. (84) and $\eta\Delta > 0$, we need only show that

$$\sum_{t \geq 0} \pi_t(\mathcal{A}^*)(1 - \pi_t(\mathcal{A}^*)) = \infty. \tag{94}$$

Lemma 4.1 together with $\emptyset \neq \mathcal{A}^* \neq [K]$ implies that

$$\sum_{t \geq 0} \mathbb{I}[a_t \in \mathcal{A}^*] = \sum_{t \geq 0} \mathbb{I}[a_t \notin \mathcal{A}^*] = \infty \text{ a.s.} \tag{95}$$

Since $\mathbb{P}(a_t \in \mathcal{A}^*|\mathcal{F}_t) = \pi_t(\mathcal{A}^*)$ and $\mathbb{P}(a_t \notin \mathcal{A}^*|\mathcal{F}_t) = 1 - \pi_t(\mathcal{A}^*)$, the Extended Borel-Cantelli Lemma (Lemma A.1) applied to Eq. (95) furnishes $\sum_{t \geq 0} \pi_t(\mathcal{A}^*) = \sum_{t \geq 0}(1 - \pi_t(\mathcal{A}^*)) = \infty$ a.s. We now break into cases to show that Eq. (94) holds regardless of the behavior of $\pi_t(\mathcal{A}^*)$.

If $\pi_t(\mathcal{A}^*) \geq 1/2$ only finitely often then we can set $u := \max\{t \geq 0 : \pi_t(\mathcal{A}^*) \geq 1/2\}$ for

$$\sum_{t \geq 0} \pi_t(\mathcal{A}^*)(1 - \pi_t(\mathcal{A}^*)) \geq \sum_{t > u} \frac{\pi_t(\mathcal{A}^*)}{2} = \infty. \tag{96}$$

Similarly, if $\pi_t(\mathcal{A}^*) < 1/2$ only finitely often then $u := \max\{t \geq 0 : \pi_t(\mathcal{A}^*) < 1/2\}$ gives us

$$\sum_{t \geq 0} \pi_t(\mathcal{A}^*)(1 - \pi_t(\mathcal{A}^*)) \geq \sum_{t > u} \frac{1 - \pi_t(\mathcal{A}^*)}{2} = \infty. \tag{97}$$

We can narrow our focus to the case where $\pi_t(\mathcal{A}^*)$ is both above and below $1/2$ i.o. In particular, there must be infinitely many $t \geq 0$ such that $\pi_t(\mathcal{A}^*) < 1/2$ but $\pi_{\theta_{t+1}}(\mathcal{A}^*) \geq 1/2$, and for such $t$ we have

$$\pi_{\theta_{t+1}}(\mathcal{A}^*) = \frac{\sum_{a \in \mathcal{A}^*} \exp(\theta_{t+1}(a))}{\sum_{a \in \mathcal{A}^*} \exp(\theta_{t+1}(a)) + \sum_{b \in [K] \setminus \mathcal{A}^*} \exp(\theta_{t+1}(b))}. \qquad \text{(Eq. (2))} \tag{98}$$

The above equation is of the form $x/(x+y)$, where $x := \sum_{a\in\mathcal{A}^*}\exp(\theta_{t+1}(a))$ and $y := \sum_{b\in[K]\setminus\mathcal{A}^*}\exp(\theta_{t+1}(b))$. Since $x/(x+y)$ is increasing in $x$ and decreasing in $y$ for $x, y > 0$, and $|\theta_{t+1}(c) - \theta_t(c)| \le \eta R$ for all $c \in [K]$, we can maximize the right hand side for the upper bound

$$\pi_{\theta_{t+1}}(\mathcal{A}^*) \le \frac{\sum_{a\in\mathcal{A}^*}\exp(\theta_t(a) + \eta R)}{\sum_{a\in\mathcal{A}^*}\exp(\theta_t(a) + \eta R) + \sum_{b\in[K]\setminus\mathcal{A}^*}\exp(\theta_t(b) - \eta R)}\,. \tag{99}$$

Also, $\pi_t(\mathcal{A}^*) < 1/2$ yields $\sum_{a\in\mathcal{A}^*}\exp(\theta_t(a)) < \sum_{b\in[K]\setminus\mathcal{A}^*}\exp(\theta_t(b))$, so

$$\frac{\sum_{a\in\mathcal{A}^*}\exp(\theta_t(a) + \eta R)}{\sum_{a\in\mathcal{A}^*}\exp(\theta_t(a) + \eta R) + \sum_{b\in[K]\setminus\mathcal{A}^*}\exp(\theta_t(b) - \eta R)} \tag{100}$$

$$= \frac{\exp(\eta R)\sum_{a\in\mathcal{A}^*}\exp(\theta_t(a))}{\exp(\eta R)\sum_{a\in\mathcal{A}^*}\exp(\theta_t(a)) + \exp(-\eta R)\sum_{b\in[K]\setminus\mathcal{A}^*}\exp(\theta_t(b))} \tag{101}$$

$$< \frac{\exp(\eta R)\sum_{a\in\mathcal{A}^*}\exp(\theta_t(a))}{(\exp(\eta R) + \exp(-\eta R))\sum_{a\in\mathcal{A}^*}\exp(\theta_t(a))}$$
$$\left(\sum_{a\in\mathcal{A}^*}\exp(\theta_t(a)) < \sum_{b\in[K]\setminus\mathcal{A}^*}\exp(\theta_t(b))\right) \tag{102}$$

$$= \frac{\exp(\eta R)}{\exp(\eta R) + \exp(-\eta R)} = \frac{\exp(2\eta R)}{\exp(2\eta R) + 1}\,. \tag{103}$$

Connecting the above displays, there are infinitely many $t \ge 0$ with $\pi_t(\mathcal{A}^*) < 1/2$ and $\pi_{\theta_{t+1}}(\mathcal{A}^*) \ge 1/2$, and for such $t$ we have $1 - \pi_{\theta_{t+1}}(\mathcal{A}^*) > 1 - \exp(2\eta R)/(\exp(2\eta R) + 1) = (\exp(2\eta R) + 1)^{-1}$. Therefore $\pi_{\theta_{t+1}}(\mathcal{A}^*)(1 - \pi_{\theta_{t+1}}(\mathcal{A}^*)) \ge (2\exp(2\eta R) + 2)^{-1}$ i.o, establishing Eq. (94). $\qquad\square$

The second proposition has a more complicated proof, due to the technical difficulty added by having multiple suboptimal arms with the same expected value. Controlling the suboptimal arms will be much more convenient with the following extra notation. Letting $n := |\{r(a) : a \in [K]\}|$ be the size of the range of the expected reward vector $r$, we partition the arms into $(\Phi_i)_{i\in[n]}$, where $\Phi_i := \arg\min_{a\in[K]\setminus\cup_{j<i}\Phi_j} r(a)$. Thus $\Phi_1$ is the set of arms with minimal expected reward, $\Phi_2$ is the set of arms with the second lowest expected reward, and so forth, culminating with $\Phi_n = \mathcal{A}^*$. Given $i \in [n]$, we will use the shorthands $\Phi_i^- := \cup_{j<i}\Phi_j$ and $\Phi_i^+ := \cup_{j>i}\Phi_j$. Note that $\Phi_i^-$ and $\Phi_i^+$ are the sets of arms with lower, respectively higher, expected reward than the arms in $\Phi_i$. First we will conjure up a couple bound that hold for the increments of suboptimal parameters.

**Lemma B.2** (Bounds on the Expectation and Variance of Increments for Suboptimal Arms). *For any $i \in [n-1]$, for any $b \in \Phi_i$, we have the bounds:*

$$\mathbb{E}_t[\theta_{t+1}(b) - \theta_t(b)] \le \eta\pi_t(b)\Big((1 - \pi_t(\Phi_i))r(b) - (r(b) + \Delta)\pi_t(\Phi_i^+) + R\pi_t(\Phi_i^-)\Big), \tag{104}$$

$$\mathrm{Var}_t[\theta_{t+1}(b) - \theta_t(b)] \le \eta^2 R^2 \pi_t(b)(1 - \pi_t(b))\,. \tag{105}$$

*Proof.*

$$\mathbb{E}_t[\theta_{t+1}(b) - \theta_t(b)]$$

$$= \sum_{a\in[K]} \mathbb{E}_t[\mathbb{I}[a_t = a](\theta_{t+1}(b) - \theta_t(b))] \qquad (\textstyle\sum_{a\in[K]} \mathbb{I}[a_t = a] = 1) \ (106)$$

$$= \mathbb{E}_t[\mathbb{I}[a_t = b]\eta(1 - \pi_t(b))r_t] + \sum_{a\in[K]\setminus\{b\}} \mathbb{E}_t[\mathbb{I}[a_t = a]\eta(-\pi_t(b))r_t]$$
$$\text{(update rule of Algorithm 1) (107)}$$

$$= \eta(1 - \pi_t(b))\mathbb{E}_t[\mathbb{I}[a_t = b]r_t] - \eta\pi_t(b)\sum_{a\in[K]\setminus\{b\}} \mathbb{E}_t[\mathbb{I}[a_t = a]r_t]$$
$$(\pi_t \text{ is } \mathcal{F}_t\text{-measurable}) \ (108)$$

$$= \eta(1 - \pi_t(b))\pi_t(b)r(b) - \eta\pi_t(b)\sum_{a\in[K]\setminus\{b\}} \pi_t(a)r(a) \quad (\mathbb{E}_t[\mathbb{I}[a_t = \cdot]r_t] = \pi_t(\cdot)r(\cdot)) \ (109)$$

$$= \eta\pi_t(b)\Big((1 - \pi_t(b))r(b) - \sum_{a\in\Phi_i\setminus\{b\}} \pi_t(a)r(b) - \sum_{c\in\Phi_i^+} \pi_t(c)r(c) - \sum_{d\in\Phi_i^-} \pi_t(d)r(d)\Big)$$
$$(110)$$

$$\leq \eta\pi_t(b)\Big((1 - \pi_t(b))r(b) - r(b)\sum_{a\in\Phi_i\setminus\{b\}} \pi_t(a) - (r(b) + \Delta)\sum_{c\in\Phi_i^+} \pi_t(c) + R\sum_{d\in\Phi_i^-} \pi_t(d)\Big)$$
$$(r(c) \geq r(b) + \Delta \, , \, r(d) \geq -R) \ (111)$$

$$= \eta\pi_t(b)\Big((1 - \pi_t(\Phi_i))r(b) - (r(b) + \Delta)\pi_t(\Phi_i^+) + R\pi_t(\Phi_i^-)\Big). \qquad (112)$$

$$\mathrm{Var}_t[\theta_{t+1}(b) - \theta_t(b)] \leq \mathbb{E}_t[(\theta_{t+1}(b) - \theta_t(b))^2] \qquad (113)$$

$$= \sum_{a\in[K]} \mathbb{E}_t[\mathbb{I}[a_t = a](\theta_{t+1}(b) - \theta_t(b))^2] \qquad (\textstyle\sum_{a\in[K]} \mathbb{I}[a_t = a] = 1) \ (114)$$

$$\leq \mathbb{E}_t[\mathbb{I}[a_t = b]\eta^2 R^2(1 - \pi_t(b))^2] + \sum_{a\in[K]\setminus\{b\}} \mathbb{E}_t[\mathbb{I}[a_t = a]\eta^2 R^2\pi_t(b)^2]$$
$$\text{(update rule of Algorithm 1) (115)}$$

$$\leq \eta^2 R^2(1 - \pi_t(b))^2\pi_t(b) + \eta^2 R^2\pi_t(b)^2(1 - \pi_t(b))$$
$$(\mathbb{E}_t[\mathbb{I}[a_t = \cdot]] = \pi_t(\cdot)) \ (116)$$

$$= \eta^2 R^2\pi_t(b)(1 - \pi_t(b)). \qquad (117)$$

$$\square$$

The next proposition will be applied inductively to control the relationship between the expectation and variance of arbitrary suboptimal arms.

**Lemma B.3.** *For constants $C$, $C' \geq 0$ and $i \in [n-1]$, if all $b \in \Phi_i^-$ satisfy $\lim_{t\to\infty} \theta_t(b) = -\infty$ a.s. then* a.s. *exists a finite timestep $\tau$ such that, for all $c \in \Phi_{i+1}^-$, $\sum_{a\in\mathcal{A}^*} \theta_\tau(a) \geq C + C'\theta_\tau(c)$.*

*Proof.* Without loss of generality suppose $C' \geq 1$. Throughout the proof we will use the following two constants, which depend on $C$ and $C'$:

$$U_1 := C'(K\eta R + K\log(8RK/\Delta + K)) + C, \text{ and} \qquad (118)$$
$$U_2 := \eta R + \log(8RK/\Delta + 1). \qquad (119)$$

Fix $\epsilon \in (0, 1]$, and define

$$D := \sup_{x\geq 0} -x + 20\sqrt{1 + 4K^2\eta^2 R^2 + 4\eta R^2 x/\Delta}\log\left(\frac{2 + 4\eta R^2 x/\Delta}{\epsilon/K}\right), \qquad (120)$$

noting that $D < \infty$. Proposition 4.3 says that $\lim_t \sum_{a\in\mathcal{A}^*} \theta_t(a) = \infty$ a.s, and we have by assumption that $\lim_t \theta_t(b) = -\infty$ a.s. for all $b \in \Phi_i^-$. Together these observations guarantee an a.s. finite

timestep $\mu$ such that

$$\sum_{a \in \mathcal{A}^*} \theta_\mu(a) \geq U_1 + D \geq -D \geq \max_{b \in \Phi_i^-} \theta_\mu(b) \,. \tag{121}$$

Consider the collection of sequences $\{(X_t^b)_{t > \mu} : b \in \Phi_i^-\}$ defined by $X_t^b := \theta_t(b) - \theta_{t-1}(b)$, and the sequence $(Y_t)_{t > \mu}$ defined by $Y_t := \sum_{a \in \mathcal{A}^*} \theta_t(a) - \theta_{t-1}(a)$. For each $b \in \Phi_i^-$, $(X_t^b)_{t > \mu}$ satisfies the requirements of Freedman's Inequality (Lemma A.3) with $B := \eta R$; also, $(Y_t)_{t > \mu}$ does with $B := K \eta R$. Therefore we can apply Freedman's Inequality with $\delta := \epsilon/K$ to all of these sequences simultaneously and take a union bound to conclude that, with probability $1 - \epsilon(|\Phi_i^-| + 1)/K \leq 1 - \epsilon$,

$$\sum_{a \in \mathcal{A}^*} \theta_t(a) - \theta_\mu(a) \geq \sum_{\mu < k \leq t} \mathbb{E}_{k-1}[Y_k^b] - 20 \sqrt{1 + 4K^2 \eta^2 R^2 + \sum_{\mu < k \leq t} \mathrm{Var}_{k-1}[Y_k^b] \log \left( \frac{2 + \sum_{\mu < k \leq t} \mathrm{Var}_{k-1}[Y_k^b]}{\epsilon/K} \right)} \,, \tag{122}$$

$$\forall b \in \Phi_i^- : \theta_t(b) - \theta_\mu(b) \leq \sum_{\mu < k \leq t} \mathbb{E}_{k-1}[X_k^b] + 20 \sqrt{1 + 4\eta^2 R^2 + \sum_{\mu < k \leq t} \mathrm{Var}_{k-1}[X_k^b] \log \left( \frac{2 + \sum_{\mu < k \leq t} \mathrm{Var}_{k-1}[X_k^b]}{\epsilon/K} \right)} \,, \tag{123}$$

for all $t > \mu$. Let $\mathcal{E}$ denote the event that both Eqs. (122) and (123) hold at all such $t$. We will argue that, on $\mathcal{E}$,

$$\sum_{a \in \mathcal{A}^*} \theta_t(a) \geq U_1 \geq 0 \geq \max_{b \in \Phi_i^-} \theta_t(b) \tag{124}$$

for all $t \geq \mu$ by strong induction. Thus let $t \geq \mu$, and suppose that Eq. (124) holds with $k$ in place of $t$, for all $\mu \leq k < t$, noting that it holds for $k = \mu$ by the definition of $\mu$. Eqs. (84) and (91) together imply that

$$\mathrm{Var}_{k-1}[Y_k] \leq \frac{\eta R^2}{\Delta} \mathbb{E}_{k-1}[Y_t] \leq \frac{4\eta R^2}{\Delta} \mathbb{E}_{k-1}[Y_t] \,, \tag{125}$$

for $k > \mu$, so the assumption that event $\mathcal{E}$ holds implies

$$\sum_{a \in \mathcal{A}^*} \theta_t(a) \geq \sum_{a \in \mathcal{A}^*} \theta_\mu(a) - D \tag{126}$$

$$\geq U_1 + D - D = U_1 \,. \tag{Eq. (121)} \tag{127}$$

Now pick an arbitrary $b \in \Phi_i^-$. Without loss of generality, say $b \in \Phi_j$ for some $j < i$. For $\mu \leq k < t$ the inductive hypothesis implies that there exists $a \in \mathcal{A}^*$ such that $\theta_k(a) \geq U_1/K$. Thus

$$\pi_k(\Phi_j^+)/\pi_k(\Phi_j^-) \geq \pi_k(a)/\pi_k(\Phi_i^-) \qquad (\Phi_j^- \subset \Phi_i^- \,, a \in \Phi_j^+) \tag{128}$$

$$\geq \exp(U_1/K)/K \qquad (\max_{b \in \Phi_i^-} \theta_k(b) \leq 0) \tag{129}$$

$$\geq 1 + 4R/\Delta \,, \tag{130}$$

where the final inequality above follows from $U_1 \geq K \log(K + 4K/\Delta)$. Defining the constant $\gamma := (\Delta/2 + r(b) + R)/(\Delta + r(b) + R)$, we have $\gamma \leq (\Delta + 4R)/(2\Delta + 4R)$, which implies $\gamma/(1 - \gamma) \leq 1 + 4R/\Delta \leq \pi_k(\Phi_j^+)/\pi_k(\Phi_j^-)$. Therefore

$$\pi_k(\Phi_j^+) \geq \frac{\gamma}{1 - \gamma} \pi_k(\Phi_j^-) \tag{131}$$

$$\Rightarrow \pi_k(\Phi_j^+) \geq \gamma(\pi_k(\Phi_j^+) + \pi_k(\Phi_j^-)) \tag{132}$$

$$\Rightarrow \pi_k(\Phi_j^+)(\Delta + r(b) + R) \geq (\Delta/2 + r(b) + R)(1 - \pi_k(\Phi_j)) \tag{133}$$

$$\Rightarrow \pi_k(\Phi_j^+)(\Delta + r(b)) - R(1 - \pi_k(\Phi_j) - \pi_k(\Phi_j^+)) \geq (\Delta/2 + r(b))(1 - \pi_k(\Phi_j)) \tag{134}$$

$$\Rightarrow -\Delta(1 - \pi_k(\Phi_j))/2 \geq (1 - \pi_k(\Phi_j))r(b) - \pi_k(\Phi_j^+)(\Delta + r(b)) + R\pi_k(\Phi_j^-) \,. \tag{135}$$

Combining Eqs. (104) and (135) produces

$$\mathbb{E}_{k-1}[X_k^b] \leq -\frac{\eta \Delta}{2} \pi_k(b)(1 - \pi_k(\Phi_j)) \,. \tag{136}$$

From Eq. (130) we have $\pi_k(a) \geq \pi_k(\Phi_i^-) \geq \pi_k(\Phi_j)$, so $1 - \pi_k(\Phi_j) \geq 1/2$. Thus Eqs. (105) and (136) together provide

$$\mathbb{E}_{k-1}[X_k^b] \leq -\frac{\eta\Delta}{4}\pi_k(b) \tag{137}$$

$$\leq -\frac{\Delta}{4\eta R^2}\mathrm{Var}_{k-1}[X_k^b]. \tag{138}$$

Eqs. (123) and (138) imply

$$\theta_t(b) = \theta_t(b) - \theta_\mu(b) + \theta_\mu(b) \tag{139}$$
$$\leq \theta_\mu(b) + D \tag{140}$$
$$\leq -D + D = 0. \tag{141}$$

Since $b \in \Phi_i^-$ was arbitrary, this concludes the inductive argument. We have shown that, on $\mathcal{E}$, Eq. (124) holds.

Define the stopping time $\nu$ by

$$\nu := \min\left\{t \geq \mu \;:\; \left(\forall b \in \Phi_i \;:\; \theta_t(b) < U_2\right) \text{ or } \left(\sum_{a \in \mathcal{A}^*}\theta_t(a) < U_1\right) \text{ or } \left(\max_{b \in \Phi_i^-}\theta_t(b) > 0\right)\right\}, \tag{142}$$

and define $(Z_t)_{t \geq \mu}$ by $Z_t := \sum_{b \in \Phi_i}\max(\theta_{\min(t,\nu)}(b), 0)$. We will show that $(Z_t)_{t \geq \mu}$ is a super-martingale, i.e. for all $t \geq \mu$, $\mathbb{E}_t[Z_{t+1} - Z_t] \leq 0$. If $t \geq \nu$ then we have $\mathbb{E}_t[Z_{t+1} - Z_t] = 0$, so assume $t < \nu$. Let $\mathcal{B} := \{b \in \Phi_i \;:\; \theta_t(b) \geq \eta R\}$ and $\mathcal{C} := \{c \in \Phi_i \;:\; \theta_t(c) < \eta R\}$, so

$$\mathbb{E}_t[Z_{t+1} - Z_t] = \sum_{b \in \Phi_i}\mathbb{E}_t[\max(\theta_{t+1}(b), 0) - \max(\theta_t(b), 0)] \qquad (t < \nu) \tag{143}$$

$$= \sum_{b \in \mathcal{B}}\mathbb{E}_t[\max(\theta_{t+1}(b), 0) - \max(\theta_t(b), 0)] + \sum_{c \in \mathcal{C}}\mathbb{E}_t[\max(\theta_{t+1}(c), 0) - \max(\theta_t(c), 0)]. \tag{144}$$

The terms in the sum on the left of Eq. (144) can be bounded by

$$\mathbb{E}_t[\max(\theta_{t+1}(b), 0) - \max(\theta_t(b), 0)] = \mathbb{E}_t[\theta_{t+1}(b) - \theta_t(b)]$$
$$(|\theta_{t+1}(b) - \theta_t(b)| \leq \eta R, \; \theta_t(b) \geq 0 + \eta R) \tag{145}$$
$$\leq -\frac{\eta\Delta}{2}\pi_t(b)(1 - \pi_t(\Phi_i)), \tag{146}$$

where the last inequality above comes from Eq. (136) and the fact that $\nu > t$.[4] For the sum on the right of Eq. (144), we can bound the terms by

$$\mathbb{E}_t[\theta_{t+1}(c) - \theta_t(c)] = \mathbb{E}_t[\mathbb{I}[a_t = c](\theta_{t+1}(c) - \theta_t(c))] + \mathbb{E}_t[\mathbb{I}[a_t \neq c](\theta_{t+1}(c) - \theta_t(c))] \tag{147}$$
$$\leq \mathbb{E}_t[\mathbb{I}[a_t = c]R\eta] + \mathbb{E}_t[\mathbb{I}[a_t \neq c]\pi_t(c)R\eta]$$
$$\text{(update rule of Algorithm 1) (148)}$$
$$\leq 2\eta R\pi_t(c). \tag{149}$$

Combining Eqs. (144), (146) and (149) produces

$$\mathbb{E}_t[Z_{t+1} - Z_t] \leq -\frac{\eta\Delta}{2}(1 - \pi_t(\Phi_i))\sum_{b \in \mathcal{B}}\pi_t(b) + 2\eta R\sum_{c \in \mathcal{C}}\pi_t(c) \tag{150}$$

$$= -\frac{\eta\Delta}{2}(1 - \pi_t(\Phi_i))\pi_t(\mathcal{B}) + 2\eta R\pi_t(\mathcal{C}). \tag{151}$$

---

[4]Specifically, $\nu > t$ implies the inductive hypothesis that was used to prove Eq. (136), and $\Phi_j$ can be replaced with $\Phi_i$.

If $\pi_t(\mathcal{C}) = 0$ then the above is negative, so we may assume $\pi_t(\mathcal{C}) > 0$. Since $\nu > t$, there is some $b \in \Phi_i$ with $\theta_t(b) \geq U_2 \geq \eta R$, so

$$\pi_t(\mathcal{B})/\pi_t(\mathcal{C}) \geq \pi_t(b)/\pi_t(\mathcal{C}) \tag{152}$$

$$\geq \frac{\exp(\theta_t(b))}{\sum_{c \in \mathcal{C}} \exp(\theta_t(c))} \tag{153}$$

$$\geq \frac{\exp(U_2)}{n \exp(\eta R)} \qquad \text{(definitions of } \mathcal{B} \text{ and } \mathcal{C}) \tag{154}$$

$$= \exp(U_2 - \eta R)/K \tag{155}$$

$$\geq (8RK/\Delta)/K \qquad (U_2 \geq \eta R + \log(8RK/\Delta)) \tag{156}$$

$$= 8R/\Delta \,. \tag{157}$$

Also from $\nu > t$, we have that $\sum_{a \in \mathcal{A}^*} \theta_t(a) \geq U_1$, so at least one $a \in \mathcal{A}^*$ satisfies $\theta_t(a) \geq U_1/K$. Fixing such an $a$ gives

$$(1 - \pi_t(\Phi_i))/\pi_t(\mathcal{C}) \geq \pi_t(a)/\pi_t(\mathcal{C}) \tag{158}$$

$$\geq \exp(U_1/K - \eta R)/K \qquad \text{(like Eq. (155))} \tag{159}$$

$$\geq (8Rn/\Delta + K)/K \qquad (U_1 \geq K\eta R + K\log(8RK/\Delta + K)) \tag{160}$$

$$= 8R/\Delta + 1 \,. \tag{161}$$

We will break into two cases, first assuming that $\pi_t(\Phi_i) \leq 1/2$. In this case we can upper bound Eq. (151) by

$$-\frac{\eta\Delta}{2}(1 - \pi_t(\Phi_i))\pi_t(\mathcal{B}) + 2\eta R\pi_t(\mathcal{C}) \leq -\frac{\eta\Delta}{4}\pi_t(\mathcal{B}) + 2\eta R\pi_t(\mathcal{C}) \tag{162}$$

$$\leq -2\eta R\pi_t(\mathcal{C}) + 2\eta R\pi_t(\mathcal{C}) \qquad \text{(Eq. (157))} \tag{163}$$

$$= 0 \,. \tag{164}$$

On the other hand, if $\pi_t(\Phi_i) > 1/2$ then

$$1/2 < \pi_t(\mathcal{B}) + \pi_t(\mathcal{C}) \tag{165}$$

$$\leq \pi_t(\mathcal{B})(1 + \Delta/8R) \qquad \text{(Eq. (157))} \tag{166}$$

$$\Rightarrow \frac{2R}{4R + \Delta/2} \leq \pi_t(\mathcal{B}) \,. \tag{167}$$

Starting once more from the right hand side of Eq. (151), we have

$$-\frac{\eta\Delta}{2}(1 - \pi_t(\Phi_i))\pi_t(\mathcal{B}) + 2\eta R\pi_t(\mathcal{C}) \leq -\frac{\Delta}{2} \cdot \frac{2\eta R}{4R + \Delta/2}(1 - \pi_t(\Phi_i)) + 2\eta R\pi_t(\mathcal{C})$$
$$\text{(Eq. (167))} \tag{168}$$

$$\leq -2\eta R \cdot \frac{\Delta/2}{4R + \Delta/2}(8R/\Delta + 1)\pi_t(\mathcal{C}) + 2\eta R\pi_t(\mathcal{C})$$
$$\text{(Eq. (161))} \tag{169}$$

$$= 0 \,. \tag{170}$$

In concert, Eqs. (164) and (170) together with Eq. (151) imply that $\mathbb{E}_t[Z_{t+1} - Z_t] \leq 0$ when $\mu \leq t < \nu$. Therefore $(Z_t)_{t \geq \mu}$ is a submartingale, and it is clear from its definition that $Z_t$ is bounded below by 0 at all times. We can apply Lemma A.7 and conclude that $(Z_t)_{t \geq \mu}$ converges a.s. to a random variable $Z$ with $\mathbb{E}[|Z|] \leq \infty$.

We will again break into two cases, first assuming that $\nu = \infty$, i.e. the stopping time never stops. In this case $\lim_t Z_t = \lim_t \sum_{b \in \Phi_i} \max(\theta_t(b), 0)$, and this quantity will a.s. converge to a finite value; because each summand is nonnegative, this implies that all $b \in \Phi_i$ satisfy $\limsup_t \theta_t(b) < \infty$. From the assumption that $\forall c \in \Phi_i^- : \lim_t \theta_t(c) = -\infty$, we have that, for all $b \in \Phi_{i+1}^- = \Phi_i \cup \Phi_i^-$, $\limsup_t \theta_t(b) < \infty$. By Proposition 4.3, there a.s. exists a finite timestep $\tau$ such that $\sum_{a \in \mathcal{A}^*} \theta_\tau(a) \geq C + C' \max_{c \in \Phi_{i+1}^-} \limsup_t \theta_t(c) \geq C + C' \max_{c \in \Phi_{i+1}^-} \theta_\tau(c)$, as desired.

The other case is that $\nu < \infty$; this implies either the event $\mathcal{E}$ fails to occur (since $\mathcal{E}$ implies Eq. (124)), or $\forall b \in \Phi_i : \theta_\nu(b) < U_2$. On event $\mathcal{E}$, for all $b \in \Phi_{i+1}^-$,

$$C'\theta_\nu(b) + C \leq C'U_2 + C \leq U_1 \leq \sum_{a \in \mathcal{A}^*} \theta_\nu(a), \qquad (0 + C, \, C'U_2 + C \leq U_1) \tag{171}$$

and setting $\tau := \nu$ gives the desired result. Therefore, regardless of whether or not $\nu < \infty$, the only way we don't have the desired result is if $\mathcal{E}$ fails to occur, which happens with probability at most $\epsilon$. Since $\epsilon$ was arbitrary, it can be taken to 0, and the desired result will hold a.s. $\qquad\square$

Having shown the above lemma, we are ready to establish that the parameters of suboptimal arms diverge to $-\infty$.

**Proposition 4.4** (Negative Infinite Suboptimal Parameters). *For every suboptimal arm $b \in [K] \setminus \mathcal{A}^*$, $\lim_{t\to\infty} \theta_t(b) = -\infty$ a.s.*

*Proof.* Since $\Phi_n = \mathcal{A}^*$ and $\cup_{i\in[n]}\Phi_i = [K]$, the set of suboptimal arms is $\cup_{i\in[n-1]}\Phi_i$. Thus we will perform induction over $i \in [n-1]$, proving that all $b \in \Phi_i$ satisfy $\lim_t \theta_t(b) = -\infty$ a.s. from the inductive hypothesis that

$$\forall c \in \Phi_i^- \ : \ \lim_t \theta_t(c) = -\infty \text{ a.s.} \tag{172}$$

Note that Eq. (172) is vacuously satisfied for $i = 1$. Fix an arbitrary $\epsilon \in (0, 1]$, and define

$$D := \sup_{x \geq 0} -x + 20\sqrt{1 + 4K^2\eta^2 R^2 + 4\eta R^2 x/\Delta} \log\left(\frac{2 + 4\eta R^2 x/\Delta}{\epsilon/K}\right), \tag{173}$$

noting that $D < \infty$. Let $\tau$ be the first timestep such that

$$\sum_{a\in\mathcal{A}^*} \theta_\tau(a) \geq (K+1)D + K\log(K + 4K/\Delta) + K \max_{b\in\Phi_{i+1}^-} \theta_\tau(b), \tag{174}$$

and note that $\tau$ is a stopping time. Also, $\tau < \infty$ a.s. by applying Lemma B.3 (which is applicable due to the inductive hypothesis in Eq. (172)) with $C := (K+1)D + K\log(K + 4K/\Delta)$ and $C' := K$.

now we can apply freedman's lemma to both the suboptimal arm and optimal sum, and conclude that the suboptimal arm goes to $-\infty$ wp $1 - \delta$. since $\delta$ was arbitrary the result becomes a.s, and the induction goes through meaning that the whole thing does.

Consider the collection of sequences $\{(X_t^b)_{t>\tau} \ : \ b \in \Phi_{i+1}^-\}$ defined by $X_t^b := \theta_t(b) - \theta_{t-1}(b)$, and the sequence $(Y_t)_{t>\tau}$ defined by $Y_t := \sum_{a\in\mathcal{A}^*} \theta_t(a) - \theta_{t-1}(a)$. For each $b \in \Phi_{i+1}^-$, $(X_t^b)_{t>\tau}$ satisfies the requirements of Freedman's Inequality (Lemma A.3) with $B := \eta R$; also, $(Y_t)_{t>\tau}$ does with $B := K\eta R$. Therefore we can apply Freedman's Inequality with $\delta := \epsilon/K$ to all of these sequences simultaneously and take a union bound to conclude that, with probability $1 - \epsilon(|\Phi_{i+1}^-| + 1)/K \leq 1 - \epsilon$,

$$\sum_{a\in\mathcal{A}^*} \theta_t(a) - \theta_\tau(a) \geq \sum_{\tau<k\leq t} \mathbb{E}_{k-1}[Y_k^b] - 20\sqrt{1 + 4K^2\eta^2 R^2 + \sum_{\tau<k\leq t} \text{Var}_{k-1}[Y_k^b]} \log\left(\frac{2 + \sum_{\tau<k\leq t}\text{Var}_{k-1}[Y_k^b]}{\epsilon/K}\right), \tag{175}$$

$$\forall b \in \Phi_{i+1}^- \ : \ \theta_t(b) - \theta_\tau(b) \leq \sum_{\tau<k\leq t} \mathbb{E}_{k-1}[X_k^b] + 20\sqrt{1 + 4\eta^2 R^2 + \sum_{\tau<k\leq t} \text{Var}_{k-1}[X_k^b]} \log\left(\frac{2 + \sum_{\tau<k\leq t}\text{Var}_{k-1}[X_k^b]}{\epsilon/K}\right), \tag{176}$$

for all $t > \tau$. Let $\mathcal{E}$ denote the event that both Eqs. (175) and (176) hold at all such $t$. We will argue that, on $\mathcal{E}$,

$$\sum_{a\in\mathcal{A}^*} \theta_t(a) \geq K\big(\log(K + 4K/\Delta) + \max_{b\in\Phi_{i+1}^-} \theta_\tau(b) + D\big) \geq K\big(\log(K + 4K/\Delta) + \max_{b\in\Phi_{i+1}^-} \theta_t(b)\big) \tag{177}$$

for all $t \geq \tau$ by strong induction. Thus let $t \geq \tau$, and suppose that Eq. (177) holds with $k$ in place of $t$, for all $\tau \leq k < t$, noting that it holds for $k = \tau$ by the definition of $\tau$. Eqs. (84) and (91) together imply that

$$\text{Var}_{k-1}[Y_k] \leq \frac{\eta R^2}{\Delta}\mathbb{E}_{k-1}[Y_t] \leq \frac{4\eta R^2}{\Delta}\mathbb{E}_{k-1}[Y_t], \tag{178}$$

for $k > \tau$, so the assumption that event $\mathcal{E}$ holds implies

$$\sum_{a \in \mathcal{A}^*} \theta_t(a) \geq \sum_{a \in \mathcal{A}^*} \theta_\tau(a) - D \tag{179}$$

$$\geq K \left( \log(K + 4K/\Delta) + \max_{b \in \Phi_{i+1}^-} \theta_t(b) + D \right). \tag{180}$$

Now pick an arbitrary $b \in \Phi_{i+1}^-$. Without loss of generality, say $b \in \Phi_j$, where $j \in [i]$. For $\tau \leq k < t$ the inductive hypothesis implies that there exists $a \in \mathcal{A}^*$ such that $\theta_k(a) \geq \log(K + 4K/\Delta) + \max_{b \in \Phi_{i+1}^-} \theta_k(b)$. Thus

$$\pi_k(\Phi_j^+)/\pi_k(\Phi_j^-) \geq \pi_k(a)/\pi_k(\Phi_{i+1}^-) \qquad (a \in \mathcal{A}^* \subset \Phi_{i+1}^+) \tag{181}$$

$$\geq \frac{\exp(\theta_k(a))}{K \exp(\max_{b \in \Phi_{i+1}^-} \theta_k(b))} \tag{182}$$

$$\geq \exp(\log(K + 4K/\Delta))/K \tag{183}$$

$$\geq 1 + 4R/\Delta. \tag{184}$$

Defining the constant $\gamma := (\Delta/2 + r(b) + R)/(\Delta + r(b) + R)$, we have $\gamma \leq (\Delta + 4R)/(2\Delta + 4R)$, which implies $\gamma/(1 - \gamma) \leq 1 + 4R/\Delta \leq \pi_k(\Phi_j^+)/\pi_k(\Phi_j^-)$. Therefore

$$\pi_k(\Phi_j^+) \geq \frac{\gamma}{1 - \gamma} \pi_k(\Phi_j^-) \tag{185}$$

$$\Rightarrow \pi_k(\Phi_j^+) \geq \gamma(\pi_k(\Phi_j^+) + \pi_k(\Phi_j^-)) \tag{186}$$

$$\Rightarrow \pi_k(\Phi_j^+)(\Delta + r(b) + R) \geq (\Delta/2 + r(b) + R)(1 - \pi_k(\Phi_j)) \tag{187}$$

$$\Rightarrow \pi_k(\Phi_j^+)(\Delta + r(b)) - R(1 - \pi_k(\Phi_j) - \pi_k(\Phi_j^+)) \geq (\Delta/2 + r(b))(1 - \pi_k(\Phi_j)) \tag{188}$$

$$\Rightarrow -\Delta(1 - \pi_k(\Phi_j))/2 \geq (1 - \pi_k(\Phi_j))r(b) - \pi_k(\Phi_j^+)(\Delta + r(b)) + R\pi_k(\Phi_j^-). \tag{189}$$

Combining Eqs. (104) and (189) produces

$$\mathbb{E}_{k-1}[X_k^b] \leq -\frac{\eta\Delta}{2} \pi_k(b)(1 - \pi_k(\Phi_j)). \tag{190}$$

From Eq. (184) we have $\pi_k(a) \geq \pi_k(\Phi_{i+1}^-) \geq \pi_k(\Phi_j)$, so $1 - \pi_k(\Phi_j) \geq 1/2$. Thus Eqs. (105) and (190) together provide

$$\mathbb{E}_{k-1}[X_k^b] \leq -\frac{\eta\Delta}{4} \pi_k(b) \tag{191}$$

$$\leq -\frac{\Delta}{4\eta R^2} \mathrm{Var}_{k-1}[X_k^b]. \tag{192}$$

Eqs. (176) and (192) imply

$$\theta_t(b) = \theta_t(b) - \theta_\tau(b) + \theta_\tau(b) \tag{193}$$

$$\leq D + \max_{b \in \Phi_{i+1}^-} \theta_\tau(b), \tag{194}$$

and multiplying both sides of Eq. (194) by $K$ before adding $K \log(K + 4K/\Delta)$ implies the second inequality of Eq. (177) (since $b \in \Phi_{i+1}^-$ was arbitrary) This concludes the inductive argument over $t \geq \tau$. We have shown that, on $\mathcal{E}$, Eq. (177) holds. In fact, on event $\mathcal{E}$, we can also use Eqs. (176) and (192) together with the fact that $\sum_{t \geq 1} \mathrm{Var}_{t-1}[X_t^b] = \infty$ (using Eq. (105)) to conclude that $\lim_t \theta_t(b) = -\infty$ for an arbitrary $b \in \Phi_{i+1}^-$. This finishes off the inductive argument over $i \in [n - 1]$. $\qquad\square$

The above results are all that is needed for the proof of Theorem 4.2.

## B.2 Non-stationary Convergence

Here we show that, in general, the stochastic gradient bandit algorithm (Algorithm 1) only converges to "generalized one-hot policies", i.e. $\sum_{a \in \mathcal{A}^*} \pi_t(a) = 1$, and not "true one-hot policies", i.e. $\exists a \in \mathcal{A}^* : \pi_t(a) = 1$. Among the optimal arms, there will be permanent non-stationary behavior.

**Proposition 3.2** (Non-Stationary Convergence). *Consider a $K$-armed bandit with all arms being equally good, i.e. $\mathcal{A}^* = [K]$, and at least one arm has a nonzero probability of generating a nonzero reward. Running Algorithm 1 with any $\eta > 0$ leads to*

$$\liminf_t \pi_t(a) < 1 \ \text{a.s,} \tag{1}$$

*for all $a \in [K]$. Therefore $(\pi_t)_{t \geq 0}$ does not converge to any one-hot policy.*

*Proof.* We will first argue that $(\theta_t)_{t \geq 0}$ does not converge to a fixed vector. By way of contradiction, suppose $\lim_t \theta_t(a) = \varphi \in \mathbb{R}^K$. Let $a \in [K]$ be an arm such that $P(r_t \neq 0 | a_t = a) > 0$, which exists by assumption. Without loss of generality, there exist $\epsilon, \delta > 0$ such that $P(|r_t| > \epsilon | a_t = a) > \delta$. Since $\theta_t \to \varphi$, we have that $\max_{b \in [K]} |\theta_{t+1}(b) - \theta_t(b)| \to 0$. Also, convergence implies that $lim_t \pi_t(a) \in (0, 1)$, in particular there exists $\beta \in (0, 1)$ such that eventually $\pi_t(a) < \beta$. Since $a$ is selected i.o. and every time it is selected $|r_{t+1}| > \epsilon$ with fixed probability $\delta > 0$, we have that

$$|\theta_{t+1}(a) - \theta_t(a)| \geq \eta(1 - \pi_t(a))\epsilon \tag{195}$$
$$\geq \eta(1 - \beta)\epsilon \tag{196}$$

i.o. This constant lower bound on the step taken contradicts the assumption of convergence.

We will now argue that, for all $x \in \mathbb{R}$ and $a \in [K]$, $\theta_t(a) < x$ occur i.o. or else $\theta_t(a)$ will converge. By conservation of mass this implies the desired result. Since the mean reward is the same across all arms, $(\theta_t(a))_{t \geq 0}$ is a martingale. Given any (possibly random) time $\tau$, the stopped random process induced by running $(\theta_t(a) - \theta_\tau(a))_{t \geq \tau}$ until the time it hits the set $[x - \theta_\tau(a), -\infty)$ is thus a martingale bounded below, and Doob's martingale convergence theorem implies it will converge. Either the value converges without hitting the set, in which case $\theta_t(a)$ converges, or else $\theta_t(a)$ dips below $x$ i.o. □

## C  Nonstationary Bandit Setting

**Proposition C.1** (Infinite Optimal Parameters). *If $\mathcal{A}^* \neq [K]$ then $\lim_{t \to \infty} \sum_{a \in \mathcal{A}^*} \theta_t(a) = \infty$ almost surely.*

*Proof.* For $t \geq 0$, let $X_t := \sum_{a \in \mathcal{A}^*} \theta_{t+1}(a) - \theta_t(a)$, such that $\sum_{i=0}^t X_i = \sum_{a \in \mathcal{A}^*} \theta_{t+1}(a) - \theta_0(a)$. By the update rule of Algorithm 1, note also that

$$X_t = \eta \sum_{a \in \mathcal{A}^*} (\mathbb{I}[a_t = a] - \pi_t(a)) r_t . \tag{197}$$

The conditional expectation of $X_t$ given $\mathcal{F}_t$ can be lower bounded by

$$\mathbb{E}_t[X_t] = \sum_{a \in [K]} \mathbb{E}_t[\mathbb{I}[a_t = a]X_t] \qquad\qquad (\textstyle\sum_{a \in [K]} \mathbb{I}[a_t = a] = 1) \ (198)$$

$$= \sum_{a \in \mathcal{A}^*} \mathbb{E}_t[\mathbb{I}[a_t = a]\eta(1 - \pi_t(\mathcal{A}^*))r_t] \ + \sum_{b \in [K]\setminus\mathcal{A}^*} \mathbb{E}_t[\mathbb{I}[a_t = b]\eta(-\pi_t(\mathcal{A}^*))r_t]$$
$$\text{(Eq. (197))} \ (199)$$

$$= \eta(1 - \pi_t(\mathcal{A}^*)) \sum_{a \in \mathcal{A}^*} \mathbb{E}_t[\mathbb{I}[a_t = a]r_t] - \eta\pi_t(\mathcal{A}^*) \sum_{b \in [K]\setminus\mathcal{A}^*} \mathbb{E}_t[\mathbb{I}[a_t = b]r_t]$$
$$(\pi_t \text{ is } \mathcal{F}_t\text{-measurable}) \ (200)$$

$$= \eta(1 - \pi_t(\mathcal{A}^*)) \sum_{a \in \mathcal{A}^*} \pi_t(a)r^t(a) - \eta\pi_t(\mathcal{A}^*) \sum_{b \in [K]\setminus\mathcal{A}^*} \pi_t(b)r^t(b)$$
$$(\mathbb{E}_t[\mathbb{I}[a_t = \cdot]r_t] = \pi_t(\cdot)r^t(\cdot)) \ (201)$$

$$\geq \eta(1 - \pi_t(\mathcal{A}^*)) \sum_{a \in \mathcal{A}^*} \pi_t(a)(r(a) - \Delta/3) - \eta\pi_t(\mathcal{A}^*) \sum_{b \in [K]\setminus\mathcal{A}^*} \pi_t(b)(r(b) + \Delta/3)$$
$$(\forall a \in [K], \forall t \geq \tau, |r^t(a) - r(a)| \leq \Delta/3) \ (202)$$

$$\geq \eta(1 - \pi_t(\mathcal{A}^*)) \sum_{a \in \mathcal{A}^*} \pi_t(a)(r(\mathcal{A}^*) - \Delta/3) - \eta\pi_t(\mathcal{A}^*) \sum_{b \in [K]\setminus\mathcal{A}^*} \pi_t(b)(r(\mathcal{A}^*) - 2\Delta/3)$$
$$(r(a) = r(\mathcal{A}^*) \ , \ r(b) \leq r(\mathcal{A}^*) - \Delta) \ (203)$$

$$= \eta\pi_t(\mathcal{A}^*)(1 - \pi_t(\mathcal{A}^*))\big(r(\mathcal{A}^*) - \Delta/3 - (r(\mathcal{A}^*) - 2\Delta/3)\big)$$
$$(\textstyle\sum_{a \in \mathcal{A}^*} \pi_t(a) = \pi_t(\mathcal{A}^*), \ \sum_{b \in [K]\setminus\mathcal{A}^*} \pi_t(b) = 1 - \pi_t(\mathcal{A}^*)) \ (204)$$

$$= \eta\pi_t(\mathcal{A}^*)(1 - \pi_t(\mathcal{A}^*))\Delta/3 \,, \qquad\qquad\qquad (205)$$

and the conditional variance can be upper bounded by

$$\mathrm{Var}_t[X_t] \leq \mathbb{E}_t[X_t^2] \qquad\qquad\qquad (206)$$

$$= \sum_{a \in [K]} \mathbb{E}_t[\mathbb{I}[a_t = a]X_t^2] \qquad\qquad (\textstyle\sum_{a \in [K]} \mathbb{I}[a_t = a] = 1) \ (207)$$

$$= \sum_{a \in \mathcal{A}^*} \mathbb{E}_t[\mathbb{I}[a_t = a]\big(\eta(1 - \pi_t(\mathcal{A}^*))r_t\big)^2] + \sum_{b \in [K]\setminus\mathcal{A}^*} \mathbb{E}_t[\mathbb{I}[a_t = b]\big(-\eta\pi_t(\mathcal{A}^*)r_t\big)^2]$$
$$\text{(Eq. (77))} \ (208)$$

$$\leq \eta^2(1 - \pi_t(\mathcal{A}^*))^2 R^2 \sum_{a \in \mathcal{A}^*} \mathbb{E}_t[\mathbb{I}[a_t = a]] + \eta^2\pi_t(\mathcal{A}^*)^2 R^2 \sum_{b \in [K]\setminus\mathcal{A}^*} \mathbb{E}_t[\mathbb{I}[a_t = b]]$$
$$(r_t^2 \leq R^2) \ (209)$$

$$= \eta^2(1 - \pi_t(\mathcal{A}^*))^2 R^2 \sum_{a \in \mathcal{A}^*} \pi_t(a) + \eta^2\pi_t(\mathcal{A}^*)^2 R^2 \sum_{b \in [K]\setminus\mathcal{A}^*} \pi_t(b)$$
$$(\mathbb{E}_t[\mathbb{I}[a_t = \cdot]] = \pi_t(\cdot)) \ (210)$$

$$= \eta^2 R^2 \Big((1 - \pi_t(\mathcal{A}^*))^2 \pi_t(\mathcal{A}^*) + \pi_t(\mathcal{A}^*)^2(1 - \pi_t(\mathcal{A}^*))\Big)$$
$$(\textstyle\sum_{a \in \mathcal{A}^*} \pi_t(a) = \pi_t(\mathcal{A}^*), \ \sum_{b \in [K]\setminus\mathcal{A}^*} \pi_t(b) = 1 - \pi_t(\mathcal{A}^*)) \ (211)$$

$$= \eta^2 R^2 \pi_t(\mathcal{A}^*)(1 - \pi_t(\mathcal{A}^*)) \,. \qquad\qquad\qquad (212)$$

Thus for all $t \geq \tau$ we have $\mathrm{Var}_t[X_t] \leq \eta 3R^2\Delta^{-1}\mathbb{E}_t[X_t]$, $|X_t| \leq \eta R$, and $X_t$ is $\mathcal{F}_{t+1}$-measurable. Setting $b := \eta R$ and $c := \eta R^2 \Delta^{-1}$, we need only to prove that $\sum_{t \geq \tau} \mathbb{E}_t[X_t] = \infty$, at which point we can apply the Freedman Divergence Trick (Lemma A.5) to conclude

$$\lim_t \sum_{a \in \mathcal{A}^*} \theta_{t+1}(a) - \theta_0(a) = \sum_{t \geq \tau} X_t = \infty \ \text{a.s,} \qquad\qquad (213)$$

$$\implies \lim_t \sum_{a \in \mathcal{A}^*} \theta_t(a) = \infty \ \text{a.s.} \qquad (\textstyle\sum_{a \in \mathcal{A}^*} \theta_0(a) < \infty) \ (214)$$

Thus in the remainder of the proof we turn our attention to showing $\sum_{t \geq \tau} \mathbb{E}_t[X_t] = \infty$. Applying Eq. (205) and $\eta \Delta / 3 > 0$, we need only show that

$$\sum_{t \geq \tau} \pi_t(\mathcal{A}^*)(1 - \pi_t(\mathcal{A}^*)) = \infty. \tag{215}$$

Lemma 4.1 together with $\emptyset \neq \mathcal{A}^* \neq [K]$ implies that

$$\sum_{t \geq \tau} \mathbb{I}[a_t \in \mathcal{A}^*] = \sum_{t \geq \tau} \mathbb{I}[a_t \notin \mathcal{A}^*] = \infty \text{ a.s.} \tag{216}$$

Since $\mathbb{P}(a_t \in \mathcal{A}^* | \mathcal{F}_t) = \pi_t(\mathcal{A}^*)$ and $\mathbb{P}(a_t \notin \mathcal{A}^* | \mathcal{F}_t) = 1 - \pi_t(\mathcal{A}^*)$, the Extended Borel-Cantelli Lemma (Lemma A.1) applied to Eq. (95) furnishes $\sum_{t \geq \tau} \pi_t(\mathcal{A}^*) = \sum_{t \geq \tau} (1 - \pi_t(\mathcal{A}^*)) = \infty$ a.s. We now break into cases to show that Eq. (215) holds regardless of the behavior of $\pi_t(\mathcal{A}^*)$.

If $\pi_t(\mathcal{A}^*) \geq 1/2$ only finitely often then we can set $u := \max\{t \geq 0 : \pi_t(\mathcal{A}^*) \geq 1/2\}$ for

$$\sum_{t \geq \tau} \pi_t(\mathcal{A}^*)(1 - \pi_t(\mathcal{A}^*)) \geq \sum_{t > u} \frac{\pi_t(\mathcal{A}^*)}{2} = \infty. \tag{217}$$

Similarly, if $\pi_t(\mathcal{A}^*) < 1/2$ only finitely often then $u := \max\{t \geq 0 : \pi_t(\mathcal{A}^*) < 1/2\}$ gives us

$$\sum_{t \geq \tau} \pi_t(\mathcal{A}^*)(1 - \pi_t(\mathcal{A}^*)) \geq \sum_{t > u} \frac{1 - \pi_t(\mathcal{A}^*)}{2} = \infty. \tag{218}$$

We can narrow our focus to the case where $\pi_t(\mathcal{A}^*)$ is both above and below $1/2$ i.o. In particular, there must be infinitely many $t \geq 0$ such that $\pi_t(\mathcal{A}^*) < 1/2$ but $\pi_{\theta_{t+1}}(\mathcal{A}^*) \geq 1/2$, and for such $t$ we have

$$\pi_{\theta_{t+1}}(\mathcal{A}^*) = \frac{\sum_{a \in \mathcal{A}^*} \exp(\theta_{t+1}(a))}{\sum_{a \in \mathcal{A}^*} \exp(\theta_{t+1}(a)) + \sum_{b \in [K] \setminus \mathcal{A}^*} \exp(\theta_{t+1}(b))}. \tag{Eq. (2)) (219}$$

The above equation is of the form $x/(x + y)$, where $x := \sum_{a \in \mathcal{A}^*} \exp(\theta_{t+1}(a))$ and $y := \sum_{b \in [K] \setminus \mathcal{A}^*} \exp(\theta_{t+1}(b))$. Since $x/(x + y)$ is increasing in $x$ and decreasing in $y$ for $x, y > 0$, and $|\theta_{t+1}(c) - \theta_t(c)| \leq \eta R$ for all $c \in [K]$, we can maximize the right hand side for the upper bound

$$\pi_{\theta_{t+1}}(\mathcal{A}^*) \leq \frac{\sum_{a \in \mathcal{A}^*} \exp(\theta_t(a) + \eta R)}{\sum_{a \in \mathcal{A}^*} \exp(\theta_t(a) + \eta R) + \sum_{b \in [K] \setminus \mathcal{A}^*} \exp(\theta_t(b) - \eta R)}. \tag{220}$$

Also, $\pi_t(\mathcal{A}^*) < 1/2$ yields $\sum_{a \in \mathcal{A}^*} \exp(\theta_t(a)) < \sum_{b \in [K] \setminus \mathcal{A}^*} \exp(\theta_t(b))$, so

$$\frac{\sum_{a \in \mathcal{A}^*} \exp(\theta_t(a) + \eta R)}{\sum_{a \in \mathcal{A}^*} \exp(\theta_t(a) + \eta R) + \sum_{b \in [K] \setminus \mathcal{A}^*} \exp(\theta_t(b) - \eta R)} \tag{221}$$

$$= \frac{\exp(\eta R) \sum_{a \in \mathcal{A}^*} \exp(\theta_t(a))}{\exp(\eta R) \sum_{a \in \mathcal{A}^*} \exp(\theta_t(a)) + \exp(-\eta R) \sum_{b \in [K] \setminus \mathcal{A}^*} \exp(\theta_t(b))} \tag{222}$$

$$< \frac{\exp(\eta R) \sum_{a \in \mathcal{A}^*} \exp(\theta_t(a))}{(\exp(\eta R) + \exp(-\eta R)) \sum_{a \in \mathcal{A}^*} \exp(\theta_t(a))}$$
$$\left( \sum_{a \in \mathcal{A}^*} \exp(\theta_t(a)) < \sum_{b \in [K] \setminus \mathcal{A}^*} \exp(\theta_t(b)) \right) \tag{223}$$

$$= \frac{\exp(\eta R)}{\exp(\eta R) + \exp(-\eta R)} = \frac{\exp(2\eta R)}{\exp(2\eta R) + 1}. \tag{224}$$

Connecting the above displays, there are infinitely many $t \geq 0$ with $\pi_t(\mathcal{A}^*) < 1/2$ and $\pi_{\theta_{t+1}}(\mathcal{A}^*) \geq 1/2$, and for such $t$ we have $1 - \pi_{\theta_{t+1}}(\mathcal{A}^*) > 1 - \exp(2\eta R)/(\exp(2\eta R) + 1) = (\exp(2\eta R) + 1)^{-1}$. Therefore $\pi_{\theta_{t+1}}(\mathcal{A}^*)(1 - \pi_{\theta_{t+1}}(\mathcal{A}^*)) \geq (2 \exp(2\eta R) + 2)^{-1}$ i.o, establishing Eq. (215). □

**Proposition C.2** (Finite Suboptimal Parameters). *For every suboptimal arm $b \in [K] \setminus \mathcal{A}^*$, $\lim_{t \to \infty} \theta_t(b) = -\infty$ a.s.*

*Remark* C.3. The proof remains virtually unchanged from the proof of Proposition 4.4, and the necessary changes are identical to the ones made for the proof of Proposition C.1.

**Theorem C.4.** *In the non-stationary bandit setting described as above, Algorithm 1 with any $\eta \in \Theta(1)$ almost surely converges to playing optimal arms,*

$$\lim_{t \to \infty} \sum_{a \in \mathcal{A}^*} \pi_t(a) \to 1 \text{ a.s.} \tag{225}$$

# D Reinforcement Learning

Define the MDP $\mathcal{M} = (\mathcal{H}, \mathcal{S}, \mathcal{A}, \{r_h\}_{h=0}^{H-1}, \{P_h\}_{h=0}^{H-1}, \rho)$. Let $N_t(s,a) := \sum_{t \geq 0} \mathbb{I}\{s_t = s, a_t = a\}$ be the total number of visitations of state-action pair $(s,a)$ until episode $t$. We denote that $\mathbb{P}_t^{h+1}(s_{h+1} = s'|s_h = s)$ as the probability of visiting state $s'$ in the horizon $h+1$ from the state $s$ in the horizon $h$ during the episode $t$. First, we extend the bandit exploration lemma (Lemma 4.1) to obtain its counterpart in the RL setting.

**Lemma D.1** (RL exploration (Lemma 5.1)). *Using the REINFORCE algorithm with any $\eta \in \Theta(1)$ under the finite-horizon MDP $\mathcal{M}$ defined as above, for all $h \in \mathcal{H}$, for all reachable $s \in \mathcal{S}_h$ and for all $a \in \mathcal{A}_s$ we have, almost surely, that every reachable state action pair will be visited i.o, i.e $N_\infty(s,a) = \infty$.*

*Proof.* First, for all $h \in \mathcal{H}$, for a given reachable $s \in \mathcal{S}_h$ that is played infinite often, every action $a \in \mathcal{A}_s$ will be played i.o. by the bandit exploration result (Lemma 4.1) . In other words, for all $h \in \mathcal{H}$ , for a reachable state $s$ that is played i.o, we have, almost surely that,

$$N_\infty(s,a) = \infty \iff \sum_{t \geq 0} \pi_t^h(a|s) = \infty \quad \forall a \in \mathcal{A}_s \tag{226}$$

Next, for all $h \in \mathcal{H}$, we want to show that every reachable state $s \in \mathcal{S}_h$ will be visited i.o. by induction. Suppose for a given $h \in \mathcal{H}$, for some reachable $s \in \mathcal{S}_h$ and there exists an action $a \in \mathcal{A}_s$ such that $P_{h+1}(s_{h+1} = s'|s_h = s, a_h = a) > 0$ for some $s' \in \mathcal{S}_{h+1}$, if $s$ is visited i.o, $s'$ is also visited i.o. For the base case $h = 0$, for some reachable states $s \in \mathcal{S}_0$, i.e $\rho(s) > 0$ we have,

$$\sum_{t \geq 0} \rho(s) = \infty \tag{227}$$

since $\rho(s)$ is a constant for every episode. Therefore, every reachable state $s \in \mathcal{S}_0$ is visited i.o. For the inductive case, if any reachable states $s \in \mathcal{S}_h$ is visited i.o, then any reachable states $s' \in \mathcal{S}_{h+1}$ is also visited i.o. A state $s'$ is reachable if there exists an action $a \in \mathcal{A}_s$ from a reachable state $s \in \mathcal{S}_h$ such that $P_{h+1}(s'|s,a) > 0$. Denote $c := \min_{s \in \mathcal{S}_h} \min_{a \in \mathcal{A}_s} P_{h+1}(s'|s,a)$ be the minimum transition probability from the horizon $h$ to $h+1$ among states and actions. For reachable $s'$ from $s$, we have .

$$\sum_{t \geq 0} \mathbb{P}_t^{h+1}(s'|s) = \sum_{t \geq 0} \sum_{a \in \mathcal{A}_s} P_{h+1}(s'|s,a)\pi_t^h(a|s) \tag{228}$$

$$\geq \sum_{a \in \mathcal{A}_s} c \sum_{t \geq 0} \pi_t^h(a|s) \tag{229}$$

$$= \infty \qquad \text{(by Eq. (226))}$$

Therefore, if $s \in \mathcal{S}_h$ is reachable and visited i.o, then any reachable states $s' \in \mathcal{S}_{h+1}$ from $s$ will be visited i.o. Combined Eq. (226) and Eq. (228), for all $h \in \mathcal{H}$, we have that any reachable state-action $(s,a) \in \mathcal{S}_h \times \mathcal{A}_h$ pairs will be visited i.o. we know that every state-action pair will be visited i.o. $\quad\square$

Next, we obtain the convergence of REINFORCE in the finite-horizon setting.

**Theorem D.2** (RL convergence (Theorem 5.2)). *For the MDP defined as above, using the algorithm REINFORCE with constant learning rate $\eta \in \Theta(1)$, we have, almost surely, for all $s \in \mathcal{S}_0$, $V_0^{\pi_t}(s) \to V_0^*(s)$ as $t \to \infty$.*

*Proof.* We denote $\delta := \min_s \min_{a,b \in \mathcal{A}_s, a \neq b} |Q(s,a) - Q(s,b)| > 0$ to be the minimum non-zero gap between $Q$-values. Denote $\mathcal{A}_h^* = \{a|a = \arg\max_{a \in \mathcal{A}_s} r(s,a)\}$ is the set of optimal action at a given state $s$. We also denote $C := \max_s \max_a \min_b (Q(s,a) - Q(s,b))$. We want to prove by backward induction that for all reachable state $s_0 \in \mathcal{S}_0$, $\sum_{a \in \mathcal{A}_0^*} \pi_t^0(a|s) \to 1$ as $t \to \infty$. Suppose for all $h' \in \{h, \ldots, H-1\}$, for all reachable $s \in \mathcal{S}_{h'}$, we have $\sum_{a \in \mathcal{A}_{h'}^*} \pi_t^{h'}(a|s) \to 1$ as $t \to \infty$, we want to prove that for all reachable $s \in \mathcal{S}_{h-1}$, we have $\sum_{a \in \mathcal{A}_{h-1}^*} \pi_t^{h-1}(a|s) \to 1$ as $t \to \infty$. . In the base case $h = H-1$, the REINFORCE update rule (Algorithm 2) is reduced to,

$$\theta_{t+1}^{H-1}(s,a) = \theta_t^{H-1}(s,a) + \eta(\mathbb{I}[a_{H-1} = a] - \pi_t^{H-1}(a|s))r_h \tag{230}$$

This is the bandit update rule (Algorithm 1) for a given reachable state $s \in \mathcal{S}_{H-1}$. By Theorem 4.2, for a given reachable state $s \in \mathcal{S}_{H-1}$, using the stochastic gradient bandit algorithm with constant learning rate $\eta \in \Theta(1)$, we will have, almost surely that $\sum_{a^* \in \mathcal{A}_h^*} \pi_t^{H-1}(a^*|s) \to 1$ as $t \to \infty$. By Lemma D.1, any reachable states $s \in S_{H-1}$ will be sampled i.o. Hence, using REINFORCE with $\eta \in \Theta(1)$, that for all reachable $s \in \mathcal{S}_{H-1}$ that are played i.o, we have, almost surely, $\sum_{a \in \mathcal{A}_{H-1}^*} \pi_t^{H-1}(a|s) \to 1$ as $t \to \infty$. In other words, $V_{H-1}^{\pi_t}(s) \to V_{H-1}^*(s)$ as $t \to \infty$.

For inductive case, suppose for all $h' \in \{h, \ldots, H-1\}$, for all reachable $s \in \mathcal{S}_{h'}$, we have $\sum_{a \in \mathcal{A}_{h'}^*} \pi_t^{h'}(a|s) \to 1$ as $t \to \infty$, we want to prove that for all reachable $s \in \mathcal{S}_{h-1}$, we have $\sum_{a \in \mathcal{A}_{h-1}^*} \pi_t^{h-1}(a|s) \to 1$ as $t \to \infty$. By the induction hypothesis, for all $h \in \mathcal{H}$, for all reachable $s \in \mathcal{S}_h$, $V_h^{\pi_t}(s) \to V_h^*(s)$ and $Q_h^{\pi_t}(s,a) \to Q_h^*(s,a)$ for all $a \in \mathcal{A}_s$ as $t \to \infty$. First, we note that

$$V_{h'}^*(s) - V_{h'}^{\pi_t}(s) = \sum_{a'} \pi_t^{h'}(a'|s)(\max_a Q_{h'}^*(s,a) - Q_{h'}^{\pi_t}(s,a')) \tag{231}$$

$$= \sum_{a'} \pi_t^{h'}(a'|s)(\underbrace{\max_a Q_{h'}^*(s,a) - Q_{h'}^*(s,a')}_{C_1} + \underbrace{Q_{h'}^*(s,a') - Q_{h'}^{\pi_t}(s,a')}_{C_2}) \tag{232}$$

We denote that $\eta_{h'}(t) := \sum_{a' \notin \mathcal{A}_{h'}^*} \pi_t^{h'}(a'|s)$. For the first term $C_1$, we have

$$C_1 = \sum_{a'} \pi_t^{h'}(a'|s)(\max_{a \in \mathcal{A}_s} Q_{h'}^*(s,a) - Q_{h'}^*(s,a')) \tag{233}$$

$$= \sum_{a' \notin \mathcal{A}_{h'}^*} \pi_t^{h'}(a'|s)(\max_{a \in \mathcal{A}_s} Q_{h'}^*(s,a) - Q_{h'}^*(s,a')) \tag{234}$$

$$\leq C \gamma_{h'}(t) \tag{235}$$

since the horizon $H$ is fixed and $r_h \leq R$ for all $h \in \mathcal{H}$, then

$$\max_{a \in \mathcal{A}_s} Q_{h'}^*(s,a) - Q_{h'}^*(s,a') \leq C \tag{236}$$

By the induction hypothesis, we have, for all $h' \in \{h, \ldots, H-1\}$, we have that $\gamma_{h'}(t) \to 0$ as $t \to \infty$.

For the second term $C_2$, we have $Q_{h'}^*(s,a') - Q_{h'}^{\pi_t}(s,a') \leq \alpha_{h'}(t)$, where $\alpha_{h'}(t) \to 0$ as $t \to \infty$ by induction hypothesis. Therefore,

$$V_{h'}^*(s) - V_{h'}^{\pi_t}(s) \leq C \gamma_{h'}(t) + \alpha_{h'}(t) \tag{237}$$

Denote $\epsilon_h(t) := C\gamma_{h'}(t) + \alpha_{h'}(t)$ and $\epsilon_h(t) \to 0$ as $t \to \infty$. Hence, for sufficiently large timestep $\tau$, such that for all $t \geq \tau$, for all $h' = h, \ldots, H-1$, for all reachable $s \in \mathcal{S}_{h'}$, we have that

$$V_{h'}^*(s) - V_{h'}^{\pi_t}(s) \leq \frac{\delta}{3} \tag{238}$$

where $\delta$ is the minimum possible gap Q-value defined as above. The existence of $\tau$ is guaranteed since $\epsilon_h(t) \to 0$ as $t \to \infty$. First, for a given reachable $s \in \mathcal{S}_{h-1}$, such that for any actions $a \in \mathcal{A}_s$, we have,

$$Q_{h-1}^{\pi_t}(s,a) = r_{h-1}(s,a) + \mathbb{E}_{s' \sim P_h(.|s,a)}[V_h^{\pi_t}(s')] \tag{239}$$

$$\geq r_{h-1}(s,a) + \mathbb{E}_{s' \sim P_h(.|s,a)}[V_h^*(s')] - \epsilon \tag{240}$$

$$= Q_{h-1}^*(s,a) - \epsilon \tag{241}$$

Also, for any actions $a \in \mathcal{A}_s$, we have,

$$Q_{h-1}^{\pi_t}(s,a) \leq Q_{h-1}^*(s,a) \leq Q_{h-1}^*(s,a) + \epsilon \tag{242}$$

By definition of $\delta$, for a given reachable $s \in \mathcal{S}_{h-1}$, we know that $Q_{h-1}^*(s,a) - Q_{h-1}^*(s,b) \geq \delta$ for any $a, b \in \mathcal{A}_s$ such that $a \neq b$, then we have

$$Q_{h-1}^{\pi_t}(s,a) - Q_{h-1}^{\pi_t}(s,b) \geq \delta - 2\epsilon \geq \frac{\delta}{3} \tag{243}$$

Note that $\delta$ is a non-zero gap by definition. Note that the update rule of Algorithm 2,

$$\theta_{t+1}^{h-1}(s,a) = \theta_{h-1}^{h-1}(s,a) + \eta(1\{a_{h-1} = a\} - \pi_t^{h-1}(a|s)) \sum_{h'=h-1}^{H-1} r_{h'} \tag{244}$$

is equivalent to the update rule in the nonstationary bandit setting, by considering only the updates to the arms at state $s$ and the full trajectory's rewards as the observed rewards.

Since by definition

$$\mathbb{E}^{\pi_t}\left[\sum_{h'=h-1}^{H-1} r_{h'}|s_{h-1} = s, a_{h-1} = a\right] = Q_{h-1}^{\pi_t}(s,a)\,,$$

$\sum_{h'=h-1}^{H-1} r_{h'}$ is an unbiased estimate of $Q_{h-1}^{\pi_t}(s_{h-1}, a_{h-1})$, up to nonstationarity in $\pi_t$ which eventually diminishes below $\delta/3$. Also, note that

$$Q_{h-1}^{\pi_t}(s,a) = \mathbb{E}^{\pi_t}\left[\sum_{h'=h-1}^{H-1} r_{h'}|s_{h-1} = s, a_{h-1} = a\right] \leq R(H-h)\,, \tag{245}$$

since $r(s,a) \leq R, \forall s \in \mathcal{S}, a \in \mathcal{A}$. Since the sample return $\sum_{h'=h-1}^{H-1} r_{h'}$ is a bounded, unbiased estimator of $Q_{h-1}^{\pi_t}(s_{h-1}, a_{h-1})$, and there is a minimum gap of $\delta/3$ between Q values among different actions within the same state, we can apply the convergence result from the nonstationary bandit setting (Theorem C.4) to conclude that $\sum_{a \in \mathcal{A}_{h-1}^*(s)} \pi_t^{h-1}(a|s) \to 1$ as $t \to \infty$ for all reachable $s \in \mathcal{S}_{h-1}$. Therefore the induction hypothesis holds, and we conclude that, using REINFORCE with $\eta \in \Theta(1)$, $\sum_{a \in \mathcal{A}_h^*} \pi_t^h(a|s) \to 1$ as $t \to \infty$ for all $s \in \mathcal{S}_h$ (or $V_0^{\pi_t}(s) \to V_0^*(s)$ as $t \to \infty$ for all $s \in \mathcal{S}_0$). □

## E   Convergence rate

To obtain the convergence rate of the REINFORCE algorithm (Algorithm 2), we first generalize the convergence rate result from the bandit setting with the uniqueness assumption to the one without it. Then, we also obtain the convergence rate in the non-stationary bandit setting before showing the rate of the REINFORCE algorithm.

**Theorem E.1.** *In the bandit setting where multiple arms can have a same reward, for a large enough $\tau$, for all $T > \tau$, the average sub-optimality decreases at a rate $O(\frac{\log T}{T})$. Formally, for a constant $c$, we have*

$$\frac{1}{T}\sum_{s=\tau}^{T}\left(r(a^*) - \langle\pi_{\theta_s}, r\rangle\right) \leq \frac{c\log(T-\tau)}{(T-\tau)} \tag{246}$$

*Proof.* By Eq. (84), we have

$$\mathbb{E}_t[\theta_{t+1}(\mathcal{A}^*) - \theta_t(\mathcal{A}^*)] \geq \eta\Delta\pi_{\theta_t}(\mathcal{A}^*)(1 - \pi_{\theta_t}(\mathcal{A}^*)) \geq 0 \tag{247}$$

By Theorem 4.2, we have $\lim_t \pi_{\theta_t}(\mathcal{A}^*) = 1$ a.s. Therefore, for a large enough $t$, we have

$$\pi_{\theta_t}(\mathcal{A}^*) \geq \frac{1}{2} \tag{248}$$

By Lemma 4.1, we know that every action $a \in [K]$ will be played i.o. In other words, for all $a \in [K]$, $\sum_{t \geq 0} \pi_t(a) = \infty$. Therefore, we have

$$\sum_{t=0}^{\infty}(1 - \pi_{\theta_t}(\mathcal{A}^*)) = \infty \tag{249}$$

Therefore, we have

$$\sum_{t=0}^{\infty}\mathbb{E}_t[\theta_{t+1}(\mathcal{A}^*) - \theta_t(\mathcal{A}^*)] = \infty \quad \text{a.s} \tag{250}$$

By Eq. (91), we have

$$\mathbb{V}ar_t[\theta_{t+1}(\mathcal{A}^*) - \theta_t(\mathcal{A}^*)] \leq \eta^2 R^2 \pi_t(\mathcal{A}^*)(1 - \pi_t(\mathcal{A}^*)) \tag{251}$$

Since the conditional expecation and variance of the bound sequence $\{\theta_{t+1}(\mathcal{A}^*) - \theta_t(\mathcal{A}^*)\}_{t \geq 0}$ are proportional, we can use the Lemma A.5 to show that the expectation will dominate the variance eventually. Therefore, for all large enough $t \geq \tau$, for some constant $C > 0$

$$\frac{1}{|\mathcal{A}^*|}\theta_t(\mathcal{A}^*) \geq C \sum_{s=\tau}^{t} (1 - \pi_{\theta_s}(\mathcal{A}^*)) \tag{252}$$

It is easy to see that $\sup_t \theta_t(a) < \infty$ for all $a \in [K] \backslash A^*$. Therefore, for a large enough $t \geq \tau$, we have

$$\theta_t(a) - \frac{1}{|\mathcal{A}^*|}\theta_t(\mathcal{A}^*) \leq -C \sum_{s=\tau}^{t} (1 - \pi_{\theta_s}(\mathcal{A}^*)) \tag{253}$$

which implies that

$$\sum_{a \in [K] \backslash \mathcal{A}^*} \exp(\theta_t(a) - \frac{1}{|\mathcal{A}^*|}\theta_t(\mathcal{A}^*)) \leq (K - |\mathcal{A}^*|) \exp(-C \sum_{s=\tau}^{t} (1 - \pi_{\theta_s}(\mathcal{A}^*))) \tag{254}$$

Therefore, we have

$$1 - \pi_{\theta_t}(\mathcal{A}^*) \leq \frac{1 - \pi_{\theta_t}(\mathcal{A}^*)}{\pi_{\theta_t}(\mathcal{A}^*)} \tag{255}$$

$$= \sum_{a \in [K] \backslash \mathcal{A}^*} \frac{\pi_{\theta_t}(a)}{\pi_{\theta_t}(\mathcal{A}^*)} \tag{256}$$

$$= \sum_{a \in [K] \backslash \mathcal{A}^*} \frac{\exp(\theta_t(a))}{\sum_{a' \in \mathcal{A}^*} \exp(\theta_t(a'))} \tag{257}$$

$$\leq \sum_{a \in [K] \backslash \mathcal{A}^*} \frac{\exp(\theta_t(a))}{|\mathcal{A}^*| \exp(\frac{1}{|\mathcal{A}^*|}\theta_t(\mathcal{A}^*))} \quad \text{(Jensen's inequality)} \tag{258}$$

$$\leq (K - |\mathcal{A}^*|) \exp(-C \sum_{s=\tau}^{t} (1 - \pi_{\theta_s}(\mathcal{A}^*))) \quad \text{(Eq. (254))} \tag{259}$$

By (Mei et al., 2024a, Lemma 15) with $x_n = \sum_{s=\tau}^{t-1}(1 - \pi_{\theta_s}(\mathcal{A}^*)) > 0, x_{n+1} = \sum_{s=\tau}^{t}(1 - \pi_{\theta_s}(\mathcal{A}^*)) > 0, c = C > 0, B = (K - |\mathcal{A}^*|) \geq 1$, gives us for all $t \geq \tau$,

$$\sum_{s=\tau}^{t} (1 - \pi_{\theta_s}(\mathcal{A}^*)) \leq \frac{1}{C} \log(C(t - \tau) + \exp(CM)) + \frac{\pi^2}{12C} \tag{260}$$

where $M = \max\{B, \frac{1}{C}\log(C(K - |\mathcal{A}^*|)), 1 - \pi_{\theta_\tau}(\mathcal{A}^*)\}$ Finally, for all $s \geq \tau$ and $T > \tau$, we have

$$r(a^*) - \langle \pi_{\theta_s}, r \rangle = \sum_{a \in [K] \backslash \mathcal{A}^*} \pi_{\theta_s}(a)(r(a^*) - r(a)) \leq 2R(1 - \pi_{\theta_s}(\mathcal{A}^*)) \tag{261}$$

Summing from $\tau$ to $T$, we have

$$\frac{1}{T} \sum_{s=\tau}^{T} \left( r(a^*) - \langle \pi_{\theta_s}, r \rangle \right) \leq \frac{2R(\frac{1}{C}\log(C(T - \tau) + \exp(CM)) + \frac{\pi^2}{12C})}{T - \tau} \tag{262}$$

$\square$

**Theorem E.2.** *In the non-stationary bandit setting, for a large enough $\tau$, then for all $T > \tau$, the average sub-optimality decreases at a rate $O(\frac{\log T}{T})$. Formally, for a constant c, we have*

$$\frac{1}{T} \sum_{s=\tau}^{T} \left( r(a^*) - \langle \pi_{\theta_s}, r \rangle \right) \leq \frac{c \log(T - \tau)}{T - \tau} \tag{263}$$

*Proof.* Repeating the same analysis with Eq. (205), Eq. (212), Theorem C.4, we have, for all $t \geq \tau''$,

$$\sum_{s=\tau''}^{t} (1 - \pi_{\theta_s}(\mathcal{A}^*)) \leq \frac{1}{C} \log(C(t - \tau) + \exp(CM)) + \frac{\pi^2}{12C} \tag{264}$$

where $M = \max\{B, \frac{1}{C}\log(C(K - |\mathcal{A}^*|)), 1 - \pi_{\theta_{\tau''}}(\mathcal{A}^*)\}$. Also, from the non-stationary bandit setting, there exists $\tau'$ such that for all $t \geq \tau'$,

$$|r(a) - r^t(a)| \leq \frac{\Delta}{3} \tag{265}$$

for all $a \in [K]$. Therefore, for all $s \geq \max\{\tau', \tau''\}$ and $T > \max\{\tau', \tau''\}$, we have

$$r(a^*) - \langle \pi_{\theta_s}, r \rangle \leq 2R(1 - \pi_{\theta_s}(\mathcal{A}^*)) \tag{266}$$

Summing from $\tau := \max\{\tau', \tau''\}$ to T, we have

$$\frac{1}{T} \sum_{s=\tau}^{T} \left( r(a^*) - \langle \pi_{\theta_s}, r \rangle \right) \leq \frac{2R(\frac{1}{C}\log(C(T - \tau) + \exp(CM)) + \frac{\pi^2}{12C})}{T - \tau} \tag{267}$$

$\square$

**Theorem E.3.** *In the finite-horizon MDP setting, for a large enough $\tau$, for all $T > \tau$, for all $s \in \mathcal{S}_0$, the average sub-optimality decreases at a rate $O(\frac{\log T}{T})$. Formally, for a constant c, we have*

$$\frac{1}{T} \sum_{s=\tau}^{T} \left( V_0^*(s) - V_0^{\pi_s}(s) \right) \leq \frac{c \log T}{T} \tag{268}$$

*Proof.* Repeating the same analysis, we have, for each $h \in \{0, ..., H-1\}$, for all $s \in \mathcal{S}_h$, for all $t \geq \tau_h$,

$$\sum_{s=\tau_h}^{t} (1 - \pi_h^s(\mathcal{A}_s^*|s)) \leq \frac{1}{C} \log(C(t - \tau_h) + \exp(CM)) + \frac{\pi^2}{12C} \tag{269}$$

where $M_h = \max\{|\mathcal{A}_s| - |\mathcal{A}_s^*|, \frac{1}{C}\log(C(|\mathcal{A}_s| - |\mathcal{A}_s^*|)), 1 - \pi_{\theta_{\tau_h}}(\mathcal{A}_s^*|s)\}$. Also, there exists $\tau_h'$ such that for all $t \geq \tau_h'$,

$$\|Q_h^*(s, .) - Q_h^{\pi_t}(s, .)\|_\infty \leq \frac{\Delta}{3} \tag{270}$$

for all $a \in \mathcal{A}_s$. Therefore, for all horizon $h \in \{0, ..., H-1\}$, for all $t \geq \max\{\tau_h, \tau_h'\}$ and $T > \max\{\tau_h, \tau_h'\}$, we have

$$Q_h^*(s, a^*) - \sum_{a \in \mathcal{A}_s} \pi_h^t(a|s)Q_h^{\pi_t}(s, a) \leq Q_h^*(s, a^*) - \sum_{a \in \mathcal{A}_s} \pi_h^t(a|s)(Q_h^*(s, a) - \frac{\Delta}{3}) \tag{271}$$

$$= \sum_{a \in A_s \backslash A_s^*} \pi_h^t(a|s)(Q_h^*(s, a^*) - Q_h^*(s, a)) + \frac{\Delta}{3} \tag{272}$$

$$\leq 2(H - h)R(1 - \pi_h^t(\mathcal{A}_s^*|s)) + \frac{\Delta}{3} \tag{273}$$

which implies

$$\frac{1}{T} \sum_{s=\max\{\tau_h, \tau_h'\}}^{T} (V_h^*(s) - V_h^{\pi_t}(s)) \leq \frac{2R(H - h)(\frac{1}{C}\log(C(T - \tau_h) + \exp(CM)) + \frac{\pi^2}{12C}) + \frac{\Delta}{3}(T - \tau_h)}{T - \tau_h} \tag{274}$$

Since for all $h \in \{0, ..., H-1\}$, for all $s \in \mathcal{S}_h$, for all $a \in \mathcal{A}_s$, $\lim_t Q_h^{\pi_t}(s, a) = Q_h^*(s, a)$, we can take $\delta \to 0$. Therefore,

$$\frac{1}{T} \sum_{s=\max\{\tau_0, \tau_0'\}}^{T} (V_0^*(s) - V_0^{\pi_t}(s)) \leq \frac{2HR\frac{1}{C}\log(C(T - \tau_0) + \exp(CM)) + \frac{\pi^2}{12C})}{T - \tau_0} \tag{275}$$

$\square$

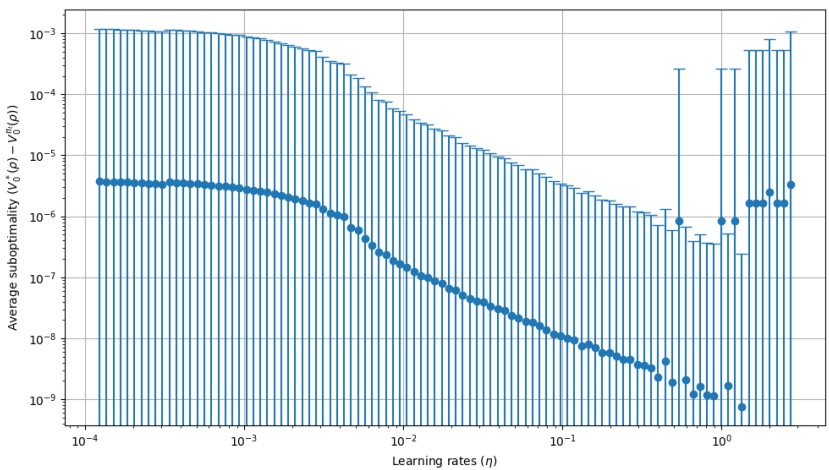

Figure 5: Average last-iterate suboptimality gap of ChainMDP size $4$

# F  Additional experiments

Specifically, we measured the average suboptimality in the last episodes over 30 of the algorithm in longer ChainMDP, DeepSea environment and CartPole environment. For ChainMDP, we extended the lengths of the environment to $H = \{4, 5, 6\}$ and measured the average suboptimality gap across 100 learning rates (from $\exp(-9)$ to $\exp(1)$). For each length, we observed a clear bowl-shaped curve. As complexity (chain length) increased, the specific thresholds of the bowl shape varied slightly, but the optimal learning rate remained consistently around $\eta \approx .95$. Next, we gradually increase the complexity of our evaluation by testing the REINFORCE algorithm (Algorithm 2) on the deep sea treasure environment. The agent operates in a square gridworld of a given depth $d = \{5, 6, 7\}$). It starts at the top left corner and its goal is to reach the bottom right corner and receive a reward of $1$. The agent has two action 1 and 2. While taking action 1 leads the agent downwards and receives no reward, taking the other leads the agent downwards and to the right and receives a reward of $-0.001$. Similar to the previous environment, for different depths, we measure the average suboptimality of the agent trained from $10^6$ episodes over 30 seeds using 100 different learning rates from $\exp(-9)$ to $\exp(7)$. We observed a similar "bowl" shape across the learning rates. However, the thresholds are different from the previous analysis. Specifically, the learning rate $\eta = 10$ has the lowest suboptimality. Finally, we evaluate the performance of the REINFORCE algorithm (Algorithm 2) in the Cartpole environment. Specifically, we measure the average return received by the agent from $10^5$ episodes over 5 seeds using $\eta = \{10^{-5}, 10^{-4}, 10^{-2}, 1\}$. Again, we observed a similar "bowl" shape across learning rates. The learning rate $\eta = 0.01$ achieves the highest average return (approximately 150), while the average return of the others stay around 25. Overall, we consistently find a "bowl-shaped" relationship between the learning rate and performance, and the specific shape and optimal point of this bowl vary significantly with the environment's structure.

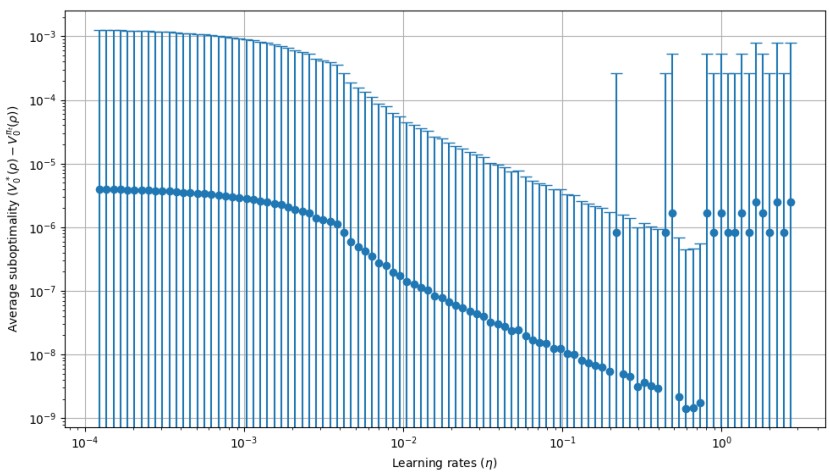

Figure 6: Average last-iterate suboptimality gap of ChainMDP size 5

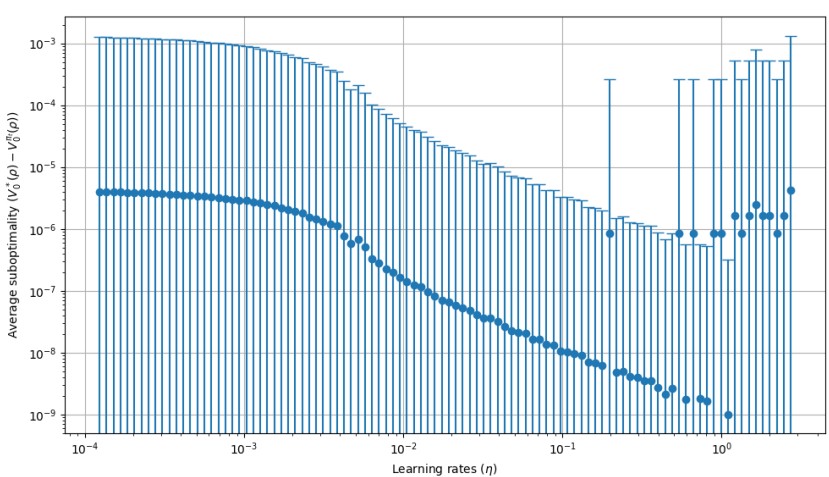

Figure 7: Average last-iterate suboptimality gap of ChainMDP size 6

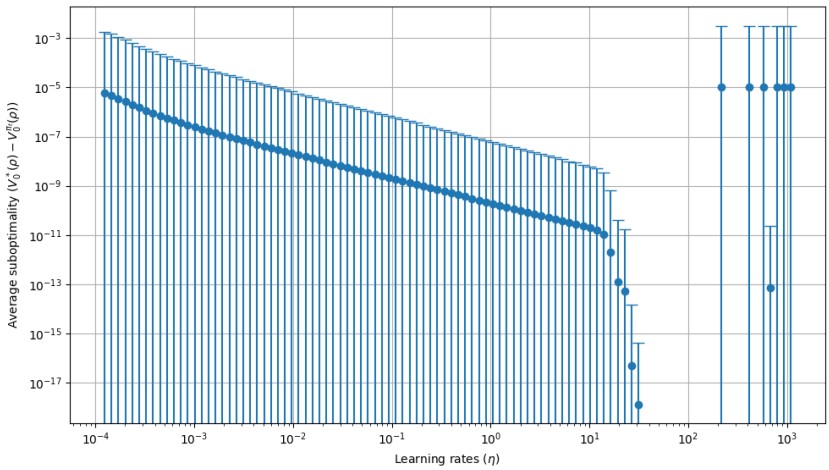

Figure 8: Average last-iterate suboptimality gap of DeepSea depth 5

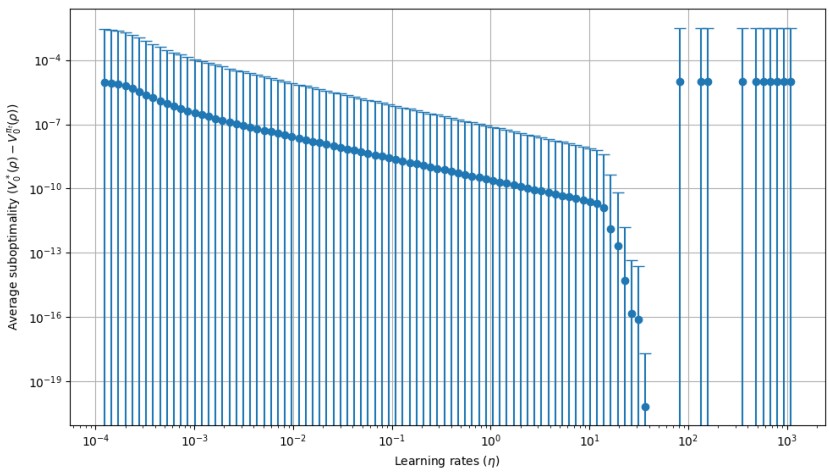

Figure 9: Average last-iterate suboptimality gap of DeepSea depth 6

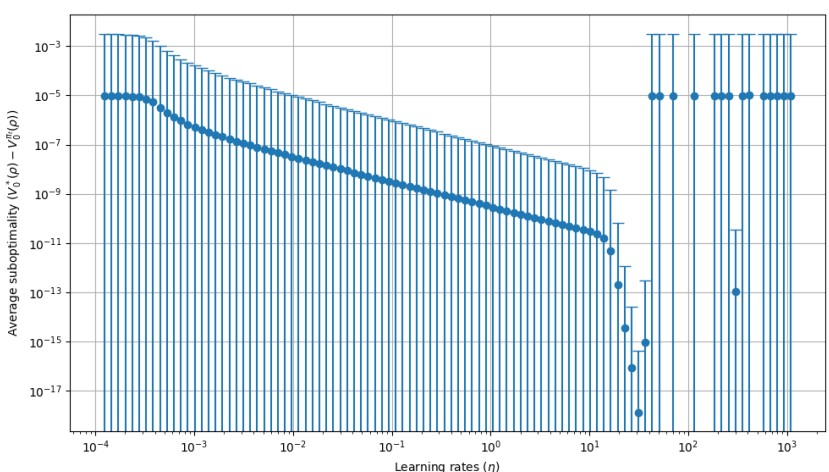

Figure 10: Average last-iterate suboptimality gap of DeepSea depth 7

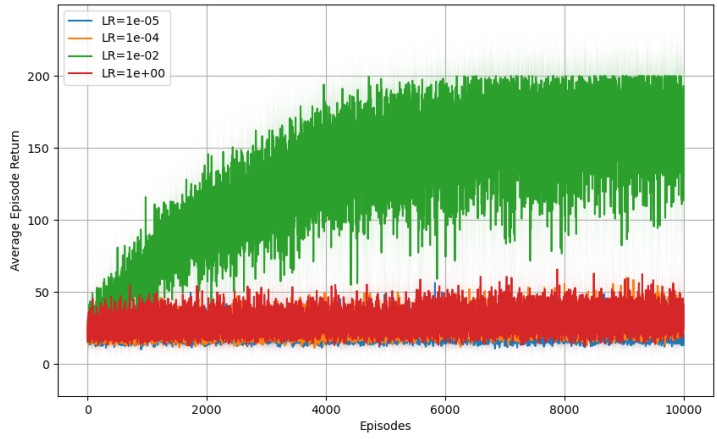

Figure 11: Average suboptimality gap of CartPole

