# OpenReview forum: "REINFORCE Converges to Optimal Policies with Any Learning Rate"
_NeurIPS.cc/2025/Conference — NeurIPS 2025 poster_

### Official Review · Reviewer_abhc · 2025-06-19

**Clarity:** 4
**Significance:** 4
**Originality:** 3
**Rating:** 5
**Confidence:** 2

**Summary:**

This paper studies the convergence of the stochastic policy gradient in the tabular MDP case with any $\Theta(1)$ learning rates. The technical challenges arise from the ties of policy values. The paper provides a new exploration lemma that enables the analysis.

**Questions:**

In the traditional Borel-Cantelli Lemma, the condition $\sum \Pr(E_n) < +\infty$ only implies that the probability of infinitely many events occurring is zero. Therefore, the first claim in Proof Sketch 2 cannot be justified using the classical version. However, upon checking the appendix, it becomes clear that an extended version of the Borel-Cantelli Lemma is actually employed. I suggest that the authors explicitly state this in Proof Sketch 2 and properly cite the relevant result.

Additionally, Lemma 3.1 is not proved in the appendix. Instead, it appears as Proposition B.2. For clarity and consistency, I recommend aligning the naming between the main text and the appendix.

**Ethical Concerns:**

["NO or VERY MINOR ethics concerns only"]

**Final Justification:**

This paper is technically sound. The author's rebuttal answers my questions.

**Limitations:**

The scope is restricted to tabular settings. I believe more general settings, including linear MDPs worth exploring.




Results are specific to the softmax policy class; generalization to other parameterizations (e.g., Gaussian in continuous action spaces) is not addressed.

**Quality:**

3

**Strengths And Weaknesses:**

Honestly, I am not an expert on the theoretical side of the policy gradient. However, as far as I see, this paper contributes fundamentally in this field.

### **Strengths**

  * The main contribution is proving global convergence of REINFORCE with any constant learning rate in finite-horizon tabular MDPs. This closes a longstanding theoretical gap between practice and theory.



*   The paper removes the assumption of unique optimal actions or rewards, which was previously required in the bandit setting (e.g., Mei et al. 2024ab). This is important for generalizing to MDPs.



*   The simulation section supports the theory and highlights the trade-off between fast learning with large learning rates and increased variance.

---

### **Weaknesses**




 *  The scope is restricted to tabular settings. I believe more general settings, including linear MDPs worth exploring.





   * Results are specific to the softmax policy class; generalization to other parameterizations (e.g., Gaussian in continuous action spaces) is not addressed.

---

> ### Author Rebuttal · Authors · 2025-07-30
>
> We thank the reviewer for their valuable suggestions, and we appreciate that the reviewer recognize the contribution of our work. We address the reviewer's questions as follows.
>
> # Weaknesses
>
> > 1. The scope is restricted to tabular settings. I believe more general settings, including linear MDPs worth exploring.
>
> We completely agree! There are a variety of very interesting extensions to more general classes of MDPs for the future work (e.g. various notions of infinite horizon, function approximation). We believe our results in the tabular setting will be useful in this direction. Some non-trivial efforts and techniques are also be required for these extensions, which is beyond the scope of one paper.
>
> For example, for linear function approximation, there has been recent progress in convergence results for policy gradient methods **in the bandit setting** [1]. The analysis uses results in [3] when using large learning rates, while being restricted to the assumption that there are no ties in the true mean reward (Assumption 2.1 in the paper). We take this as an evidence that our results can also be used in generalizing these findings to MDPs. Again this would require non-trivial efforts, which we are excited to potentially explore in the future.
>
> > 2. Results are specific to the softmax policy class; generalization to other parameterizations (e.g., Gaussian in continuous action spaces) is not addressed.
>
> Our proofs rely heavily on specific structures of the softmax parameterization (in short, it is the softmax derivative gives the specific update forms, which makes Lemma 3.1 hold). This strategy on its own should be considered as a strength of the study since (i) softmax is so widely used in modern machine learning models; (ii) otherwise using general analysis tools without considering special structures wouldn't obtain strong results in this paper.
>
> That being said, we agree with the reviewer that it is both interesting and possible to abstract out a set of generic conditions on the parameterization that will guarantee convergence. Our initial attempts indicate that a key condition is the parameterization is able to make its corresponding update not too aggressive (such that there is a chance to explore sufficiently, like Lemma 3.1). A more comprehensive study would require more time and non-trivial effort, which we leave it for future work.
>
> Regarding Gaussian parameterizations in continuous action spaces, our intuition is that our analysis cannot just apply to Gaussian, and more assumptions are needed for REINFORCE to work guarantee global convergence for Gaussian policies. In particular, the softmax parameterization can represent all policies in the simplex over a finite action set (except those that assign 0 probability to any action, but these can be approximated by the softmax to arbitrary precision). On the other hand, the Gaussian parameterization imposes significant restrictions on the representable policies, and cannot approximate arbitrary distributions over a continuous action space well (such as multi-model policies, and skewed non-Gaussian shapes). iIf the optimal policies cannot be represented by Gaussian, then convergence would not be guaranteed for such setups without further assumptions. One possible direction for looking for those assumptions is that the closure conditions explored by Bhandari and Russo, 2022 [2] would need to be met.
>
> # Questions
>
> > 1. In the traditional Borel-Cantelli Lemma, the condition $\sum\textup{Pr}(E\_n) < +\infty$ only implies that the probability of infinitely many events occurring is zero. Therefore, the first claim in Proof Sketch 2 cannot be justified using the classical version. However, upon checking the appendix, it becomes clear that an extended version of the Borel-Cantelli Lemma is actually employed. I suggest that the authors explicitly state this in Proof Sketch 2 and properly cite the relevant result.
>
> Good catch! While we do cite the extended result where we give its statement (Lemma A.1), it is definitely confusing for readers familiar with the standard version when we refer to "the Borel-Cantelli Lemma" in the main paper. We will include the citation at the first mention in the main text and carry around the "extended" qualification throughout the paper.
>
> > 2. Additionally, Lemma 3.1 is not proved in the appendix. Instead, it appears as Proposition B.2. For clarity and consistency, I recommend aligning the naming between the main text and the appendix.
>
> Agreed, in a future version of the manuscript we will implement matching names for results that are stated in the appendix and the main text.
>
> [1] M. Q. Lin, J. Mei, M. Aghaei, M. Lu, B. Dai, A. Agarwal, D. Schuurmans, Cs. Szepesvári, and S. Vaswani. Rethinking the Global Convergence of Softmax Policy Gradient with Linear Function Approximation, 2025.
>
> [2] J. Bhandari, and D. Russo. Global Optimality Guarantees For Policy Gradient Methods, 2022.
>
> [3] J. Mei, B. Dai, A. Agarwal, S. Vaswani, A. Raj, Cs. Szepesvari, and D. Schuurmans. Small steps no more: Global convergence of stochastic gradient bandits for arbitrary learning rates, 2024a.

---

> > ### Comment · Reviewer_abhc · 2025-08-05
> >
> > Thank you for your response. I will keep my score as acceptance."

---

### Official Review · Reviewer_2xdN · 2025-06-24

**Clarity:** 3
**Significance:** 3
**Originality:** 3
**Rating:** 4
**Confidence:** 3

**Summary:**

The main contribution of this paper is to show that the REINFORCE algorithm converges without any assumptions. The novelty mostly lies in the case where the MDP has more than two optimal policies. They developed a new type of analysis method, basically by showing that each action must be taken infinite times, to show that the REINFORCE algorithm converges asymptotically.

**Questions:**

I have the following questions regarding this paper:

1. This paper only proved convergence asymptotically, without providing the convergence rate. Is it possible to give a convergence rate guarantee for REINFORCE algorithm?
2. The numerical experiments showed that the performance of the REINFORCE algorithm deteriorate whenever the learning rate is too large or too small. Is there some suggestions on how to choose the optimal learning rate, either from theoretical or practical perspective?

**Ethical Concerns:**

["NO or VERY MINOR ethics concerns only"]

**Final Justification:**

This paper presents good results. However, due to absence of the convergence rate analysis, I think that this paper still has room for improvement. Hence I will maintain the score of this paper as weak acceptance.

**Limitations:**

Yes.

**Paper Formatting Concerns:**

No formatting concern appears in this paper.

**Quality:**

3

**Strengths And Weaknesses:**

Strengths:
1. This paper is well-written. The proofs are correct.
2. The results are very clean. They proved the convergence of the REINFORCE algorithm without any assumption and with any learning rate.

Weakness:
1. This paper only proves the convergence of the REINFORCE algorithm, they did not provide any convergence rate guarantee.
2. The numerical experiments are only based on examples constructed artificially by the authors. No experiment on real data or common RL tasks is conducted.

---

> ### Author Rebuttal · Authors · 2025-07-30
>
> We thank the reviewer for appreciating our contributions and their careful reading and valuable feedback. We address the main concerns as follows.
>
> # Weaknesses
> > 1. This paper only proves the convergence of the REINFORCE algorithm, they did not provide any convergence rate guarantee.
>
> We agree with the reviewer that a convergence rate is of great interest for improving the practicality of our analysis. To this end we have extended the $O(\log(T)/T)$ convergence rate of Mei et al. 2024a [1] to our bandit setting (without any assumptions about ties in the mean rewards of arms). This is the only known rate in the literature that is in a setting comparable to ours. We then establish the *exact same* asymptotic convergence rate in the RL setting.
>
> It remains an open question to see if this rate can be improved or extended to a nonasymptotic convergence rate; while there is still interesting work to be done in this direction, we believe that the inclusion of this rate increases the utility of our results significantly. Please find a high level description of the proof of the rates below (in the "Convergence Rate" section). It is&mdash;perhaps surprisingly&mdash;a much simpler argument compared to the proof of convergence. We will include these rates and proofs in an updated version of the manuscript.
>
> > 2. The numerical experiments are only based on examples constructed artificially by the authors. No experiment on real data or common RL tasks is conducted.
>
> While we believe that the major contributions of the paper are theoretical, and there is a large literature of empirical work (e.g. Duan et al. 2016 [2]) studying the behavior of policy gradient methods like REINFORCE, following the reviewer's suggestion, we have extended the experiments to include standard MDPs beyond those in the initial submission, such as gridworld and classical control tasks. While we are unfortunately not able to attach figures, these new experiments and their results are summarised in the "New Experiments" section of the rebuttal to Reviewer 2sF6 above (due to character limit of each rebuttal section). We will add visualizations and detailed descriptions of the experiments to a future version of the manuscript. We hope these extended experimental results are satisfactory.
>
> # Questions
>
> > 1. This paper only proved convergence asymptotically, without providing the convergence rate. Is it possible to give a convergence rate guarantee for REINFORCE algorithm?
>
> Yes, please see the "Convergence Rate" section below.
>
> > 2. The numerical experiments showed that the performance of the REINFORCE algorithm deteriorate whenever the learning rate is too large or too small. Is there some suggestions on how to choose the optimal learning rate, either from theoretical or practical perspective?
>
> At a first approximation, we believe the takeaway from our experiments is that small learning rates tend to converge with relatively stable progress, but simply adjust the weights too slowly. On the other hand, large learning rates are severely effected by noise in the reward signal, which causes them to overcommit to suboptimal actions. The process of unlearning these overcommitments is then very slow, because the real optimal actions are seldom sampled. Thus the optimal learning rate for a given time horizon seems to be related to the (problem dependent) optimal balance between exploration and exploitation. For more discussion see the "New Experiments" section mentioned above.
>
> We view the message of our convergence results as that practitioners are doing something valid when they tune a constant sized learning rate to get the best performance. Previous theoretical guarantees tell the story that the only constant learning rates which can be trusted to converge in the limit are impractically small. Furthermore, our results free practitioners from the concern that they might need complicated schedules or decaying learning rates to ensure the soundness of their algorithm.
>
> # Convergence Rate
> In bandits, we show that the average suboptimality converges to $0$ at an asymptotic rate of $O(\log(T)/T)$, matching Theorem 3 of Mei et al. 2024a [1], which is the only known convergence rate for the stochastic gradient setting with large learning rates. As with the rest of our results, we achieve this rate without assuming that there are no ties between the expected rewards of arms. We also extend the rate to the MDP setting, and show that REINFORCE achieves exactly the same guarantee.
>
> Below we sketch out the proof modifications necessary to extend Theorem 3 of Mei et al. 2024a [1] to our settings, which are fairly straightforward given our convergence result. We will freely use notation and reference lines from the submitted manuscript, including the appendices from the supplementary material.
>
> ### Bandit Convergence Rate
>
> We prove that, when running Algorithm 1 in the bandit setting, there exists a constant $C > 0$ such that
>
> \begin{align}
> \exists\tau>0\ :\ \forall T>\tau\ :\ \frac{\sum_{t=\tau}^T r(\mathcal{A}^*) - \pi_{\theta_t}^\top r}{T-\tau} \le \frac{C\log(T-\tau)}{T-\tau}\ .
> \end{align}
>
> We begin by controlling the "progress" of the sum of the optimal parameters, $\mathbb{E}_t[\sum\_{a\in \mathcal{A}^\*}\theta\_{t+1}(a) - \theta_t(a)]$.
> As shown in the proof of Proposition B.3, and in particular Eq. (56), this quantity is lower bounded by $\eta\Delta\pi\_{\theta\_t}(\mathcal{A}^\*) (1 - \pi\_{\theta_t}(\mathcal{A}^\*))$. Also, by Theorem 3.2, we have $\lim\_t\pi\_{\theta\_t}(\mathcal{A}^\*) = 1$ a.s, so we can select $\tau$ such that $\forall t \ge \tau\ :\ \pi\_{\theta\_t}(\mathcal{A}^\*) \ge 1/2$. Since $\mathcal{A}^\*\ne[K]$, Lemma 3.1 implies that an arm (every arm) in $[K]\setminus\mathcal{A}^\*$ is played i.o, and the extended Borel-Cantelli Lemma gives us that $\sum\_{t\ge0}(1 - \pi\_{\theta\_t}(\mathcal{A}^\*)) = \infty$. Therefore
> \begin{align}
> \sum\_{t\ge0}\mathbb{E}\_t\big[\sum\_{a\in \mathcal{A}^\*}\theta\_{t+1}(a) - \theta\_t(a)\big] = \infty \text{ a.s.}
> \end{align}
> We can also control the variance using Eq. (63):  Var$\_t[\sum\_{a\in \mathcal{A}^\*}\theta\_{t+1}(a) - \theta_t(a)] \le \eta^2R^2\pi\_{\theta\_t}(\mathcal{A}^\*) (1 - \pi\_{\theta_t}(\mathcal{A}^\*))$. Therefore the conditional variances and expectations of the sequence of bounded random variables $(\sum\_{a\in \mathcal{A}^\*}\theta\_{t+1}(a) - \theta_t(a))\_{t\ge0}$ are proportional, and since the conditional expectations are not summable we can apply the Freedman Divergence Trick (Lemma A.4). In fact, to extract the convergence rate we will need to use Eq. (21) from the proof of Lemma A.4, which tells us that, for arbitrary $\delta > 0$, $\sum\_{a\in\mathcal{A}^\*}\theta\_t(a)$ eventually grows at a rate of $\Omega(\sum\_{s=0}^t 1 - \pi\_s(\mathcal{A}^\*))$ with probability $1 - \delta$. Taking $\delta \to 0$, as $\delta$ does not appear in the event, makes this guarantee almost sure.
>
> Since we have shown in Proposition 3.4 that suboptimal arms have their parameters diverge to $-\infty$, we have the same $\Omega$ growth rate for $\sum\_{a\in\mathcal{A}^\*}\theta\_t(a) - \theta\_t(b)$ for any arm $b \in [K] \setminus \mathcal{A}^\*$. At this point we can use Lemma 15 of Mei et al. [1], replacing $\pi\_{\theta\_t}(a^\*)$ with $\pi\_{\theta\_t}(\mathcal{A}^\*)$, and then convert the result to a bound on the suboptimality gap. **Note** that this is where the difference between our result and the statement of Theorem 3 by Mei et al. [1] differ slightly: our result is mildly stronger by replacing $T$ with $T - \tau$ in two places; this does not result from novel analysis, but is a natural consequence of the arguments (in particular, Lemma 15) of Mei et al. [1]. We believe they made a typo here, resulting in a correct but slightly (non asymptotically) looser bound.
>
> ### MDP Convergence Rate
>
> Extending the above convergence rate from the bandit setting to MDPs, by way of the nonstationary bandit setting described in section 4.1, is straightforward. Intuitively, since the above rate kicks in after a $\tau$ such that $\forall t \ge \tau\ :\ \pi\_{\theta\_t}(\mathcal{A}^\*)\ge1/2$ (among other requirements), we will be able to get a similar result in MDPs after waiting sufficiently long for the learner to be guaranteed to play optimal actions with probability at least $1/2$, uniformly across all states. The formal statement of the convergence rate is as follows.
>
> We prove that, when running Algorithm 2 (REINFORCE) in the finite-horizon MDP setting, there exists a constant $C > 0$ such that
> \begin{align}
> \exists\tau>0\ :\ \forall T>\tau\ :\ \frac{\sum\_{t=\tau}^TV\_0^\*(\rho) - V\_0^{\pi_{\theta\_t}}(\rho)}{T-\tau} \le \frac{C\log(T-\tau)}{T-\tau}\ .
> \end{align}
> To prove this, first we repeat the bandit proof in the nonstationary bandit setting&mdash;making only minor notational changes suffices. Then we conclude from Theorem 4.2 that there exists a time $\tau$ such that at all $t\ge\tau$:
>
> *(a)* $\forall h \in \mathcal{H}\ :\ \forall s \in \mathcal{S}\_h\ :\ \lVert Q^\*\_h(s,\cdot) - Q^{\theta\_t}\_h(s,\cdot)\rVert\_\infty < \delta/3$ , and
> *(b)* $\forall s \in \mathcal{S}\ :\ \pi\_{\theta\_t}(\mathcal{A}^\*(s)) \ge 1/2$ .
>
> After $\tau$ we can apply the convergence rate uniformly across the states, take the maximum of the finite set of constant factors (the $C$s), and multiply this constant by the horizon $H$ (which is constant in $t$) to procure a constant $C$ which satisfies the desired inequality.
>
> [1] J. Mei, B. Dai, A. Agarwal, S. Vaswani, A. Raj, Cs. Szepesvari, and D. Schuurmans. Small steps no more: Global convergence of stochastic gradient bandits for arbitrary learning rates, 2024a.
>
> [2] Y. Duan, X. Chen, R Houthouft, J. Schulman, P. Abbeel. Benchmarking Deep Reinforcement Learning for Continuous Control, 2016.

---

> > ### Comment · Reviewer_2xdN · 2025-08-05
> >
> > Thank you very much for your response. I don't have further questions. And I will keep my score unchanged.

---

### Official Review · Reviewer_zR1q · 2025-06-27

**Clarity:** 3
**Significance:** 2
**Originality:** 2
**Rating:** 4
**Confidence:** 3

**Summary:**

This paper studies REINFORCE stochastic policy gradient (SPG) method and proves that it converges to globally optimal policies in both bandit and finite-horizon MDP settings without the assumption of unique mean reward values.

**Questions:**

1. Given that this work involves a re-analysis of existing algorithms, what key technical novelties have led to the improvements on convergence guarantees and the extension to finite-horizon MDPs?
2. Is it possible to establish similar results in infinite horizon MDPs or even with linear function approximation?

**Ethical Concerns:**

["NO or VERY MINOR ethics concerns only"]

**Final Justification:**

The rebuttal provides convergence rate guarantees and additional experimental results, which address my concerns and strengthen the original findings. I have therefore decided to raise my score.

**Limitations:**

From the experiments, an extremely large learning rate leads to significant training variance, while a small learning rate causes slow training progress. Notably, this work lacks theoretical analysis on how learning rates influence the convergence rate of algorithms, which undermines the practical significance of the theoretical findings in this work.

**Paper Formatting Concerns:**

No formatting issues were noticed.

**Quality:**

3

**Strengths And Weaknesses:**

**Strengths:**

This work eliminates the assumption of unique mean reward values, which is common in bandit setting and provides convergence guarantees for REINFORCE stochastic policy gradient (SPG) method. The results in this work also demonstrate that the online SPG method samples all actions infinitely often during exploration-exploitation, which contradicts the conjecture in [1].

**Weaknesses:**

Establishing a convergence rate theorem for different learning rates would significantly enhance the practical utility of this work. While the finding that REINFORCE SPG algorithms converge under any constant learning rate is theoretically valuable, its immediate practical significance remains limited without convergence rate guarantees. It would be valuable to explore whether novel learning rate designs could theoretically achieve improved convergence rates compared to [1].

Additionally, expanding experiments to larger-scale MDPs could further demonstrate how learning rates affect convergence speed, given that existing research is limited to a single MDP case.

[1] J. Mei, Z. Zhong, B. Dai, A. Agarwal, C. Szepesvari, and D. Schuurmans. Stochastic gradient succeeds for bandits, 2024b. URL https://arxiv.org/abs/2402.17235.

---

> ### Author Rebuttal · Authors · 2025-07-28
>
> We thank the reviewer for their careful reading and valuable feedback. The main concerns are addressed as follows, and hopefully the reviewer would reconsider the rating once the issue is clarified.
>
> # Weaknesses
>
> > 1. Establishing a convergence rate theorem for different learning rates would significantly enhance the practical utility of this work. While the finding that REINFORCE SPG algorithms converge under any constant learning rate is theoretically valuable, its immediate practical significance remains limited without convergence rate guarantees. It would be valuable to explore whether novel learning rate designs could theoretically achieve improved convergence rates compared to [1].
>
> A key contribution of our work, and a source of practical relevance is **simplicity**. While complicated learning rate designs can be powerful, they typically introduce additional hyperparameters, increasing the computational cost and human effort of implementation. Our analysis focuses on the fundamental constant step size setting and proves that this is sufficient for global convergence. One can begin experimenting without using a tapering learning rate or estimating a tiny problem-dependent constant, knowing that their algorithm is fundamentally sound. Specifically, in the chain MDP of length 4, our theory validates using a step size like $10^{−3}$—a choice that our experiments show accelerates convergence significantly compared to $10^{-5}$, which is suggested by prior theory [2, 3]. In summary, we show that practitioners can rely on a simpler, easier-to-tune optimizer and still be backed by a strong theoretical guarantee.
>
> That being said, we agree with the reviewer that a convergence rate is of great interest for improving the practicality of our analysis. To this end we have extended the $O(\log(T)/T)$ convergence rate of Mei et al. 2024a [1] to our bandit setting (without any assumptions about ties in the mean rewards of arms).  We then establish the *exact same* asymptotic convergence rate in the RL setting. Please find a high level description of the proofs in the "Convergence Rate" section of the rebuttal to Reviewer 2xdN below (due to the limit of characters in this rebuttal section). We will include these rates and proofs in subsequent versions of the manuscript. We hope that these rates significantly improve the impact of our work as the reviewer suggested.
>
> It is worthing noting that Mei et al. 2024a [1] have the only rate in the literature that is in a setting comparable to ours. We highlight that the rate of Mei et al. 2024b [2] is only available for impractically small learning rates (and also requires no ties in the true mean reward, prohibiting extensions to RL). The technical details of the $O(1/T)$ rate of Mei et al. 2024b [2] depend heavily on the miniscule learning rate, and are thus completely inapplicable to the large $\eta$ regime that we study.
>
> We thank the reviewer for pointing that a fine grained understanding of learning rate designs and their interaction with convergence rates would be incredibly valuable. This is a very interesting research program. Our preliminary attempts show some evidences that results in this direction will be problem dependent. For instance, in a trivial bandit problem with two arms, one always returning positive rewards and the other negative rewards, increasing the learning rate can only speed up convergence (since the optimal action will always increase its parameter while the sub-optimal action always decrease its parameter). On the other hand, in a more difficult problem instance, there is empirical evidence that large learning rates cause excessive variance, such as Fig 3.b in the paper.
>
> > 2. Additionally, expanding experiments to larger-scale MDPs could further demonstrate how learning rates affect convergence speed, given that existing research is limited to a single MDP case.
>
> While our contributions in this paper are primarily theoretical, as several reviewers suggested, we extend our experiments as described in the "New Experiments" section of the rebuttal to Reviewer 2sF6 (due to character limit). We have also summarized the results and conclusions in that section. Unfortunately it is not possible to attach figures, but we will visualize and explain in detail these experiments in subsequent versions.
>
> # Questions
>
> > 1. Given that this work involves a re-analysis of existing algorithms, what key technical novelties have led to the improvements on convergence guarantees and the extension to finite-horizon MDPs?
>
> When there is no isolated stationary points (w/o Assumption 2.1), our analysis is still able to argue for convergence (without relying on convergence toward stationary points). To achieve this, the following novelties are needed.
>
> ### Lemma 3.1 for exploration
> Previous work in the bandit setting [1] relied on the result that at least two actions are sampled infinitely often, and showed that the optimal action was one of the infinitely sampled ones. From there, the assumption of unique mean rewards was necessary to show that "progress" dominated "variance" in the updates. We significantly strengthened the exploration guarantee with a novel proof by contradiction, showing that *every action* is sampled infinitely often. This on its own could be used to simplify the proofs for previous results, but we combine it with the other technical contributions to remove Assumption 2.1:
>
> ### Removing Assumption 2.1
>
> A significant limitation in prior analyses is the reliance on Assumption 2.1. This is a strong condition, as in the RL setting it essentially requires the existence of a single, uniquely optimal trajectory. When this assumption is violated—for instance in any MDP where multiple actions at a state have the same $Q$-value—the analytical framework of previous work is no longer applicable (e.g. see Eq. 114, Theorem 2 of Mei et al. 2024a [1]). To overcome this, we have to show that $\sum\_{a\in\mathcal A^\*} \theta\_t(a) \to\infty$ as $t\to\infty$. This causes a surprising amount of difficulty: while showing that the sum of optimal parameters diverges to $\infty$ is relatively straightforward, showing that the sum of suboptimal parameters diverges to $-\infty$ is not enough for convergence. The pathological case where at least one $a \in \mathcal{A}^\*$ has $\lim\_t\theta\_t(a) = \infty$ but also some $b \notin \mathcal{A}\^*$ satisfies $\lim\_t\theta\_t(b) = \infty$ must be ruled out by arguing per-action for suboptimal actions. This pathology could induce nonconvergence if $\theta\_t(b)$ grows at an asymptotically faster rate! Arguing per-action in the presence of ties faces a technical hurdle: at a given timestep, if there are two or more equally good (suboptimal) arms and one is very likely to be selected, the expected update of any of them approaches 0. Thus the noise dominates. Handling this issue requires repeated and subtle use of concentration inequalities, and is the most involved proof in our work.
>
> ### Characterizing Convergence with the Set of Optimal Policies
>
> By removing the uniqueness assumption, we can provide a more precise characterization of the algorithm's limiting behavior. Our results show that the learned policy $\pi\_t$​ converges to a "generalized" one-hot distribution that concentrates its probability mass on the **set** of optimal actions, $\mathcal{A}^\*$. Specifically, we prove that $\lim\_t\sum\_{a\in\mathcal{A}^\*} \pi\_{\theta\_t}(a) = 1$, but in general $\lim\inf\_t\theta\_t(a) \ne \infty$ for all $a\in\mathcal A^\*$. This result provides a more nuanced understanding of how policy gradient methods behave when multiple optimal policies exist. We show that true one-hot convergence does not always hold when there are multiple optimal actions (in Proposition 2.2), contradicting a conjecture made in the literature [1]. This observation shows why prior proof techniques [1,2], which are only suitable for establishing one-hot convergence, cannot be easily extended to handle ties in the mean reward.
>
> > 2. Is it possible to establish similar results in infinite horizon MDPs or even with linear function approximation?
>
> Thank you for asking this interesting question! Our speculation is that it will be possible to establish similar results in an infinite horizon setting. The work by Zhang et al. [4] provides a promising starting point; they designed an algorithm using a randomized horizon to achieve unbiased gradient estimation. However, their analysis only guarantees convergence to a stationary point, not necessarily an optimal policy. They also require a decreasing learning rate which approaches 0 at a controlled rate. An interesting avenue for future work would be to build upon these ideas to establish the stronger global convergence guarantees that our paper provides for the finite-horizon case.
>
> Recent progress [5] shows the convergence of stochastic policy gradient with linear function approximation. However, the analysis is restricted to the bandit setting, and they rely on Assumption 2.1. Generalizing the findings to MDPs represents a non-trivial challenge that we are excited to potentially explore in the future.
>
> [1] J. Mei, B. Dai, A. Agarwal, S. Vaswani, A. Raj, Cs. Szepesvári, and D. Schuurmans. Small steps no more: Global convergence of stochastic gradient bandits for arbitrary learning rates, 2024a.
>
> [2] J. Mei, Z. Zhong, B. Dai, A. Agarwal, Cs. Szepesvári, and D. Schuurmans. Stochastic gradient succeeds for bandits, 2024b.
>
> [3] S. Klein, S. Weissmann, and L. Döring. Beyond stationarity: Convergence analysis of stochastic softmax policy gradient methods, 2023.
>
> [4] K. Zhang, A. Koppel, H. Zhu, and T. Basar. Global convergence of policy gradient methods to (almost) locally optimal policies, 2020.
>
> [5] M. Q. Lin, J. Mei, M. Aghaei, M. Lu, B. Dai, A. Agarwal, D. Schuurmans, Cs. Szepesvári, and S. Vaswani. Rethinking the Global Convergence of Softmax Policy Gradient with Linear Function Approximation, 2025.

---

> ### Comment · Reviewer_zR1q · 2025-08-05
>
> Thank you for the detailed rebuttal. It has addressed my concerns and clarified the issues I raised. I have increased my score accordingly.

---

### Official Review · Reviewer_2sF6 · 2025-06-30

**Clarity:** 3
**Significance:** 2
**Originality:** 3
**Rating:** 4
**Confidence:** 4

**Summary:**

This paper presents a novel theoretical analysis of the stochastic policy gradient method with a constant learning rate. The authors offer improved convergence guarantees in the presence of reward ties by extending the existing bandit exploration lemma. Furthermore, they generalize their results to the RL setting. Preliminary experimental results are also provided.

**Questions:**

1. Is it possible to extend the analysis to the case of inexact gradient updates for the REINFORCE algorithm? This could provide practical guidance for choosing an appropriate batch size.

2. The experimental results show that a large learning rate ( $\eta=2$) leads to non-convergence, yet Theorem 4.2 does not specify a clear upper bound on $\eta$. Could the authors provide an explanation for this apparent discrepancy?

**Ethical Concerns:**

["NO or VERY MINOR ethics concerns only"]

**Final Justification:**

I appreciate the author's rebuttal. I will keep my score.

**Limitations:**

yes

**Paper Formatting Concerns:**

No.

**Quality:**

3

**Strengths And Weaknesses:**

Strengths:

1. This paper extends the results of the previous work by Mai et al. (2024), removing the assumption that there is a unique optimal action. Technically, the authors strengthen the exploration property from "at least two actions are selected infinitely often" to "all actions are selected infinitely often." This refinement leads to an improved convergence result, showing that the SPG method ultimately assigns all probability mass to the set of optimal actions. The technical contribution is solid.

2. This paper also extends the results to the RL setting, which was not addressed in previous work. Although this extension may not be highly novel, it is important since the RL setting and the REINFORCE algorithm are more practical for real-world applications.

3. The paper is clearly written and easy to follow. The proof sketches effectively convey the high-level ideas, and the detailed proofs are rigorous.

Weaknesses:

1. The absence of convergence rate analysis is, in my view, a key weakness of this work. The paper provides only a convergence guarantee for a classical RL algorithm, which is not that useful in practice, as it offers no guidance for practitioners on how to select critical hyperparameters such as the learning rate or batch size.

2. The experiments are conducted only on a simple MDP. Including experiments on more diverse simulated environments and tasks would significantly strengthen the experimental section of the paper.


.

---

> ### Author Rebuttal · Authors · 2025-07-30
>
> We thank the reviewer for appreciating our contribution, and for their careful reading and valuable feedback. The main concerns are addressed as follows.
>
> # Weaknesses
>
> > 1. The absence of convergence rate analysis is, in my view, a key weakness of this work. The paper provides only a convergence guarantee for a classical RL algorithm, which is not that useful in practice, as it offers no guidance for practitioners on how to select critical hyperparameters such as the learning rate or batch size.
>
> We agree with the reviewer that a convergence rate is of great interest for improving the practicality of our analysis. To this end we have extended the $O(\log(T)/T)$ convergence rate of Mei et al. 2024a [1] to our bandit setting (without any assumptions about ties in the mean rewards of arms). This is the only known rate in the literature that is in a setting comparable to ours. We then establish the *exact same* asymptotic convergence rate in the RL setting. Please find a high level description of the proofs in the "Convergence Rate" section of the rebuttal to Reviewer 2xdN below (since there is no enough space in this rebuttal section due to character limit). We will include these rates and proofs in updated versions of the manuscript.
>
> As for the guidance our theory provides practitioners, there are takeaways regarding both the learning rate and the batch size. Existing theory provides guarantees for learning rates that are either impractically small or decreasing towards 0 on specific schedules; our work can be interpreted by practitioners as a license to tune a fixed learning rate without worrying about meeting scheduling conditions or being under a threshold. Similarly, our work (and other theoretical works in the stochastic gradient settings) do not rely on perfectly accurate estimates of the gradients. In other words, our results hold for a batch size of one or more&mdash;as opposed to the exact gradient setting, which is analogous to (but even stricter than) requiring a large batch for a highly accurate estimate of the gradient.
>
> > 2. The experiments are conducted only on a simple MDP. Including experiments on more diverse simulated environments and tasks would significantly strengthen the experimental section of the paper.
>
> While our contributions in this paper are primarily theoretical, following the reviewer's suggestions, we have extended our experiments as described in the "New Experiments" section below. Unfortunately it is not possible to attach figures, but we will add visualizations and detailed explanations of these experiments to subsequent versions.
>
> # Questions
>
> > 1. Is it possible to extend the analysis to the case of inexact gradient updates for the REINFORCE algorithm? This could provide practical guidance for choosing an appropriate batch size.
>
> We are unclear as to the precise meaning of this question, and will try to answer it with three interpretations. **First**, if by inexact gradients the reviewer means that the learner only receives noisy data from the environment, and has to estimate the gradients from these data, then our analysis already accounts for this; we are effectively studying the setting where the batch size is 1, i.e. the learner must act from a single observation (reward in bandits or trajectory in RL). Thus our analysis will work for any batch size.
>
> **Second**, if the reviewer is referring to biased gradient observations, then we would argue that this issue is not present in our settings: since the computation of unbiased gradients is trivial from single observations of the rewards, analysis of biased gradient computations is unnecessary.
>
> **Third**, if inexact gradient updates are from other structures, such as function approximations, our speculation is that combining our results with recent work on function approximations, such as [2] would be able to extend to that case of inexact gradient updates for the REINFORCE algorithm. However, [2] is restricted to the bandit setting, and they rely on Assumption 2.1. This mean non-trivial efforts are needed to extend our analysis to function approximation settings.
>
> > 2. The experimental results show that a large learning rate $(\eta=2)$ leads to non-convergence, yet Theorem 4.2 does not specify a clear upper bound on $\eta$. Could the authors provide an explanation for this apparent discrepancy?
>
> This seeming discrepancy highlights the important separation between finite time vs. asymptotic results. For excessively large learning rates there might be an arbitrarily large period before convergence (and even before the converge rate) kicks in. We think that the figure showing large learning rates failing to converge for an extended period is important to highlight the caveat of taking $\lim\_{t\to\infty}$. However, no finite-time experiment could actually conflict with our asymptotic convergence result.
>
> # New Experiments
> In order to strengthen the experimental section of our paper, we ran new experiments to validate our findings in a more diverse set of environments. Specifically, we have conducted new experiments with two primary questions in mind:
> 1. What is the relationship between the learner's convergence rate (as measured by finite-time performance, i.e. empirical suboptimality or return averaged across trials) and the magnitude of the constant learning rate?
> 2. Does this relationship depend on the complexity of the environment (e.g., number of states, exploration requirement, chain length) or vary significantly across different types of environments?
>
> Our experiments across three distinct groups of environments give preliminary empirical answers to these questions. We consistently find a "bowl-shaped" relationship between the learning rate and performance, i.e. there is a unique minimum. The specific shape of this bowl and the minimum (the optimal value of $\eta$ for the time ranges studied) vary significantly with the environment's structure, across many orders of magnitude. Since we cannot include the visualizations in the rebuttal, we provide a summary of our findings, which will be added in full (including plots) to the revised manuscript.
> ### 1. Chain MDPs (Extended)
> - **Method:** We extended our analysis to Chain MDPs of lengths $H=\{4, 5, 6\}$, measuring the *average suboptimality* of an agent trained for $10^6$ episodes over $30$ seeds across $100$ learning rates (from $10^{-9}$ to $2$).
> - **Result:** For each length, we observed a clear bowl-shaped curve. As complexity (chain length) increased, the specific thresholds of the bowl shape varied slightly, but the optimal learning rate remained consistently around $\eta ≈ 0.95$. We were surprised that the added length had such a muted effect, while switching the environment completely changed the optimal learning rate. We hypothesis that the size of the best learning rate (at a given budget of training timesteps) may be sensitive to the amount of exploration required to learn the environment's dymanics.
> ### 2. Deep Sea Environment
> - **Method:** To evaluate a tabular MDP with sparse and deceptive rewards, we tested on the Deep Sea environment for depths $d=\{5, 6, 7\}$, measuring the average suboptimality of an agent trained for $10^6$ episodes over 5 seeds with $\eta =\{10^{-8}, 10^{-5}, 10^{-3}, 0.1 , 0.5, 1, 2, 5, 10, 50, 100\}$.
> - **Result:** The bowl-shaped phenomenon was again present, but the scale of the bowl was dramatically different. The optimal learning rate shifted significantly to $\eta = 10$, demonstrating that the pattern varies substantially across different environment types.
> ### 3. CartPole Environment
> - **Method:** To evaluate a classic control task, we benchmarked on CartPole, which has a continuous observation space that we discretized. We highlight that this environment violates the assumption of previous work that optimal actions are unique: when the pole is nearly upright there are a variety of optimal policies which will prevent it from following down.
> We measured the *average return* of an agent trained from $10^5$ episodes over 5 seeds with $\eta = \{10^{-5}, 10^{-4}, 10^{-2}, 1\}$.
> - **Result:** The results again showed a clear bowl shape. The best average return was achieved at $\eta = 0.01$, a value orders of magnitude different from the optimal rates in the other environments.
> ### Conclusions
> In summary, our experiments lead to two preliminary conclusions about setting the learning rate in different environments.
> - First, there is indeed a consistent *bowl-shaped pattern* relating the constant learning rate to final performance across all tested environments, meaning that most problems have best constant learning rate ranges (too large or too small learning rates wouldn't perform well as is already observed in the simple simulations in the paper).
> - Second, the specific characteristics of this bowl—including the location of the optimal learning rate—*change with the environment's complexity and vary significantly across different environments*.
>
> However, we have to re-emphasize that these finite-time experiments do not contradict our asymptotic theory: we know that if the time axis was extended far enough, in the limit all of these fixed learning rates would achieve optimal performance.
>
> [1] J. Mei, B. Dai, A. Agarwal, S. Vaswani, A. Raj, Cs. Szepesvári, and D. Schuurmans. Small steps no more: Global convergence of stochastic gradient bandits for arbitrary learning rates, 2024a.
>
> [2] M. Q. Lin, J. Mei, M. Aghaei, M. Lu, B. Dai, A. Agarwal, D. Schuurmans, Cs. Szepesvári, and S. Vaswani. Rethinking the Global Convergence of Softmax Policy Gradient with Linear Function Approximation, 2025.

---

### Note · Authors · 2025-08-14

We thank the reviewers for their constructive feedback and responsiveness during the rebuttal phase. To summarize our discussion, we would like to highlight the following points.

First, we wish to emphasize that a key contribution of our work is its $\textbf{simplicity}$, which is also a primary source of its practical relevance. While more complex learning rate schedules can be powerful, they often introduce additional hyperparameters, increasing both the computational cost and the human effort required for implementation. Our analysis focuses on the fundamental constant step size setting and proves its sufficiency for achieving global convergence.

Second, our analysis is $\textbf{not a simple extension}$ of previous work from the bandit setting to the finite-horizon setting. Specifically, our theoretical contributions include: 1) removing the strong assumption of equally optimal mean rewards, 2) a novel exploration result that significantly strengthens previous exploration guarantees in our setting, and 3) characterizing the non-stationary convergence behavior that arises without a unique optimal policy.

Finally, in response to the initial reviews, we have expanded our empirical evaluation to more complex and realistic environments, including the Long Chain MDP, Deep Sea MDP, and the CartPole environment. Across all tested scenarios, we observed a consistent bowl-shaped relationship between the constant learning rate and final performance at a fixed number of timesteps. This suggests that an optimal range for the learning rate exists for most problems, though the performance curve and optimal value of the learning rate vary significantly with the environment's complexity. We have also provided an analysis of the asymptotic convergence rate, showing it to be $O(\log(T)/T)$, which matches the best known rate in the bandit setting.

We believe the additional experiments and the asymptotic convergence rate analysis, prompted by the reviewers' valuable suggestions, have significantly strengthened the contributions of our work.

---

### Decision · Program_Chairs · 2025-09-17

**Decision:**

Accept (poster)

**Comment:**

This paper presents a new theoretical analysis of the REINFORCE algorithm with constant learning rate. All the reviewers appreciated the work and commented that the analysis is novel and provides a natural extension of the previous work by Mei et al. (2024). During the discussion phase, the authors also provided definite convergence rates for the bandit setting and MDP setting and I request the authors to include the results in the final version of the paper.